# Consistency Trajectory Models: Learning Probability Flow ODE Trajectory of Diffusion

**Dongjun Kim**[*][†] **& Chieh-Hsin Lai**[*]
Sony AI
Tokyo, Japan
`dongjun@stanford.edu, chieh-hsin.lai@sony.com`

**Wei-Hsiang Liao & Naoki Murata & Yuhta Takida & Toshimitsu Uesaka**
Sony AI
Tokyo, Japan

**Yutong He**[†]
Carnegie Mellon University
PA, USA

**Yuki Mitsufuji**
Sony AI, Sony Group Corporation
Tokyo, Japan

**Stefano Ermon**
Stanford University
CA, USA

## Abstract

Consistency Models (CM) (Song et al., 2023) accelerate score-based diffusion model sampling at the cost of sample quality but lack a natural way to trade-off quality for speed. To address this limitation, we propose **C**onsistency **T**rajectory **M**odel (CTM), a generalization encompassing CM and score-based models as special cases. CTM trains a single neural network that can – in a single forward pass – output scores (i.e., gradients of log-density) and enables unrestricted traversal between any initial and final time along the Probability Flow Ordinary Differential Equation (ODE) in a diffusion process. CTM enables the efficient combination of adversarial training and denoising score matching loss to enhance performance and achieves new state-of-the-art FIDs for single-step diffusion model sampling on CIFAR-10 (FID $1.73$) and ImageNet at $64 \times 64$ resolution (FID $1.92$). CTM also enables a new family of sampling schemes, both deterministic and stochastic, involving long jumps along the ODE solution trajectories. It consistently improves sample quality as computational budgets increase, avoiding the degradation seen in CM. Furthermore, unlike CM, CTM's access to the score function can streamline the adoption of established controllable/conditional generation methods from the diffusion community. This access also enables the computation of likelihood. The code is available at `https://github.com/sony/ctm`.

## 1 Introduction

Deep generative models encounter distinct training and sampling challenges. Variational Autoencoder (VAE) (Kingma & Welling, 2013) can be trained easily but may suffer from posterior collapse, resulting in blurry samples, while Generative Adversarial Network (GAN) (Goodfellow et al., 2014) generates high-quality samples but faces training instability. Conversely, Diffusion Model (DM) (Sohl-Dickstein et al., 2015; Ho et al., 2020; Song et al., 2020b) addresses these issues by learning the score (i.e., gradient of log-density) (Song & Ermon, 2019), which can generate high quality samples. However, compared to VAE and GAN excelling at fast sampling, DM involves a gradual denoising process that slows down sampling, requiring numerous model evaluations.

Score-based diffusion models synthesize data by solving the reverse-time (stochastic or deterministic) process corresponding to a prescribed forward process that adds noise to the data (Song & Ermon, 2019; Song et al., 2020b). Although advanced numerical solvers (Lu et al., 2022b; Zhang & Chen,

---

[*]Equal contribution
[†]Work done during an internship at SONY AI

2022) of Stochastic Differential Equations (SDE) or Ordinary Differential Equations (ODE) substantially reduce the required Number of Function Evaluations (NFE), further improvements are challenging due to the intrinsic discretization error present in all differential equation solvers (De Bortoli et al., 2021). Recent developments in sample efficiency thus focus on *Distillation models* (Salimans & Ho, 2021) (Figure 1) that directly estimates the integral along the Probability Flow (PF) ODE sample trajectory, amortizing the computational cost of numerical solvers, exemplified by the Consistency Model (CM) (Song et al., 2023). However, their generation quality does not improve as NFE increases (the red curve of Figure 7). Theorem 1 (in this paper) explains this inherent absence of speed-quality trade-off in CM's multistep sampling by the overlapping time intervals between jumps. This persists as a fundamental issue when training jumps solely to zero-time as in CM.

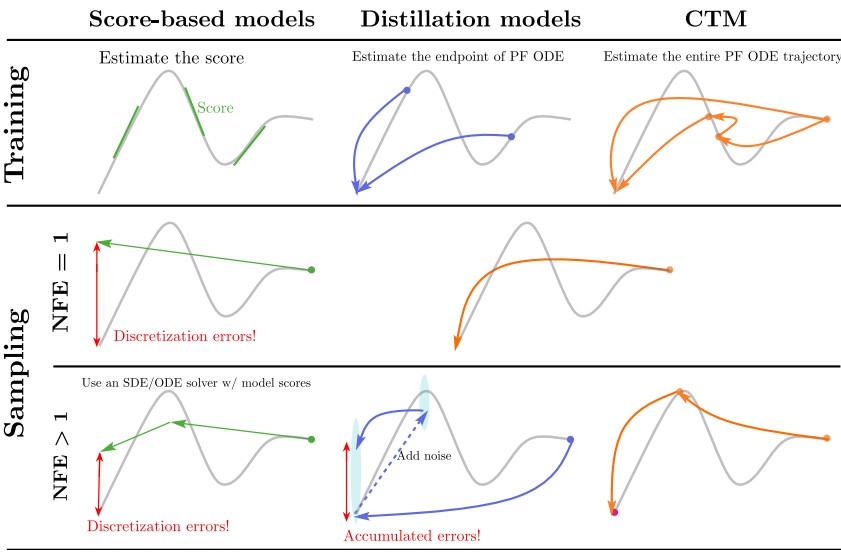

Figure 1: Training and sampling comparisons of score-based and distillation models with CTM. Score-based models exhibit discretization errors during SDE/ODE solving, while distillation models can accumulate errors in multistep sampling. CTM mitigates these issues with $\gamma$-sampling ($\gamma = 0$).

This paper introduces the ***Consistency Trajectory Model*** (CTM) as a unified framework simultaneously assessing both the integrand (score function) and the integral (jump) of the PF ODE, thus bridging score-based and distillation models. More specifically, CTM estimates anytime-to-anytime jump, ranging both infinitesimally small jumps (score function) and long jumps (integral over any time horizon) along the PF ODE, providing increased flexibility at inference time. Particularly, our unique feature enables a novel sampling method called $\gamma$-*sampling,* which alternates forward and backward jumps along the solution trajectory, with $\gamma$ governing the level of stochasticity.

CTM's anytime-to-anytime jump along the PF ODE greatly enhances its training flexibility as well. It allows the combination of the distillation loss and auxiliary losses, such as denoising score matching (DSM) and adversarial losses. These auxiliary losses measures statistical divergences[1] between the data distribution and the sample distribution, which provides student high-quality training signal for better jump learning. Notably, leveraging these statistical divergences to student training enables us to train the *student as good as teacher*, reaffirming the conventional belief established in the distillation community of classification tasks that auxiliary losses beyond distillation loss can enhance student performance. In experiments, we achieve the new State-Of-The-Art (SOTA) performance in both density estimation and image generation for CIFAR-10 (Krizhevsky, 2009) and ImageNet (Russakovsky et al., 2015) at a resolution of $64 \times 64$.

---

[1]The DSM loss is closely linked to the KL divergence (Song et al., 2021; Kim et al., 2022c). Also, the adversarial GAN loss is a proxy of $f$-divergence (Nowozin et al., 2016) or IPMs (Arjovsky et al., 2017).

## 2 PRELIMINARY

In DM (Sohl-Dickstein et al., 2015; Song et al., 2020b), the encoder structure is formulated using a set of continuous-time random variables defined by a fixed forward diffusion process[2], $d\mathbf{x}_t = \sqrt{2t}\,d\mathbf{w}_t$, initialized by the data variable, $\mathbf{x}_0 \sim p_{\text{data}}$. A reverse-time process (Anderson, 1982) from $T$ to $0$ is established as $d\mathbf{x}_t = -2t\nabla \log p_t(\mathbf{x}_t)dt + \sqrt{2t}\,d\bar{\mathbf{w}}_t$, where $\bar{\mathbf{w}}_t$ is the standard Wiener process in reverse-time, and $p_t(\mathbf{x})$ is the marginal density of $\mathbf{x}_t$ following the forward process. The solution of this reverse-time process aligns with that of the forward-time process marginally (in distribution) when the reverse-time process is initialized with $\mathbf{x}_T \sim p_T$. The deterministic counterpart of the reverse-time process, called the PF ODE (Song et al., 2020b), is given by

$$\frac{d\mathbf{x}_t}{dt} = -t\nabla \log p_t(\mathbf{x}_t) = \frac{\mathbf{x}_t - \mathbb{E}_{p_{t0}(\mathbf{x}|\mathbf{x}_t)}[\mathbf{x}|\mathbf{x}_t]}{t},$$

where $p_{t0}(\mathbf{x}|\mathbf{x}_t)$ is the probability distribution of the solution of the reverse-time stochastic process from time $t$ to zero, initiated from $\mathbf{x}_t$. Here, $\mathbb{E}_{p_{t0}(\mathbf{x}|\mathbf{x}_t)}[\mathbf{x}|\mathbf{x}_t] = \mathbf{x}_t + t\nabla \log p_t(\mathbf{x}_t)$ is the denoiser function (Efron, 2011), an alternative expression for the score function $\nabla \log p_t(\mathbf{x}_t)$. For notational simplicity, we omit $p_{t0}(\mathbf{x}|\mathbf{x}_t)$, a subscript in the expectation of the denoiser, throughout the paper.

In practice, the denoiser $\mathbb{E}[\mathbf{x}|\mathbf{x}_t]$ is approximated using a neural network $D_\phi$, obtained by minimizing the DSM (Vincent, 2011; Song et al., 2020b) loss $\mathbb{E}_{\mathbf{x}_0,t,p_{0t}(\mathbf{x}|\mathbf{x}_0)}[\|\mathbf{x}_0 - D_\phi(\mathbf{x},t)\|_2^2]$, where $p_{0t}(\mathbf{x}|\mathbf{x}_0)$ is the transition probability from time 0 to $t$, initiated with $\mathbf{x}_0$. With the approximated denoiser, the empirical PF ODE is given by

$$\frac{d\mathbf{x}_t}{dt} = \frac{\mathbf{x}_t - D_\phi(\mathbf{x}_t, t)}{t}. \tag{1}$$

Sampling from DM involves solving the PF ODE, equivalent to computing the integral

$$\int_T^0 \frac{d\mathbf{x}_t}{dt}\,dt = \int_T^0 \frac{\mathbf{x}_t - D_\phi(\mathbf{x}_t, t)}{t}\,dt \iff \mathbf{x}_0 = \mathbf{x}_T + \int_T^0 \frac{\mathbf{x}_t - D_\phi(\mathbf{x}_t, t)}{t}\,dt, \tag{2}$$

where $\mathbf{x}_T$ is sampled from a prior distribution $\pi$ approximating $p_T$. Decoding strategies of DM primarily fall into two categories: *score-based sampling* with time-discretized numerical integral solvers, and *distillation sampling* where a neural network directly estimates the integral.

**Score-based Sampling** Any off-the-shelf ODE solver, denoted as $\texttt{Solver}(\mathbf{x}_T, T, 0; \phi)$ (with an initial value of $\mathbf{x}_T$ at time $T$ and ending at time 0), can be directly applied to solve Eq. (2) (Song et al., 2020b). For instance, DDIM (Song et al., 2020a) corresponds to a 1st-order Euler solver, while EDM (Karras et al., 2022) introduces a 2nd-order Heun solver. Despite recent advancements in numerical solvers (Lu et al., 2022b; Zhang & Chen, 2022), further improvements may be challenging due to the inherent discretization error present in all solvers (De Bortoli et al., 2021), ultimately limiting the sample quality obtained with few NFEs.

**Distillation Sampling** Distillation models (Salimans & Ho, 2021; Meng et al., 2023) successfully amortize the sampling cost by directly estimating the integral of Eq. (2) with a single neural network evaluation. However, their multistep sampling approach (Song et al., 2023) exhibits degrading sample quality with increasing NFE, lacking a clear trade-off between computational budget (NFE) and sample fidelity. Furthermore, multistep sampling is not deterministic, leading to uncontrollable sample variance. We refer to Appendix A for a thorough literature review.

## 3 CTM: AN UNIFICATION OF SCORE-BASED AND DISTILLATION MODELS

To address the challenges in both score-based and distillation samplings, we introduce the Consistency Trajectory Model (CTM), which integrates both decoding strategies to sample from either SDE/ODE solving or direct anytime-to-anytime jump along the PF ODE trajectory.

### 3.1 DECODER REPRESENTATION OF CTM

---

[2]This paper can be extended to VPSDE encoding (Song et al., 2020b) with re-scaling (Kim et al., 2022a).

CTM predicts both infinitesimally small step jump and long step jump of the PF ODE trajectory. Specifically, we define $G(\mathbf{x}_t, t, s)$ as the solution of the PF ODE from initial time $t$ to final time $s \leq t$:

$$G(\mathbf{x}_t, t, s) := \mathbf{x}_t + \int_t^s \frac{\mathbf{x}_u - \mathbb{E}[\mathbf{x}|\mathbf{x}_u]}{u} \, \mathrm{d}u.$$

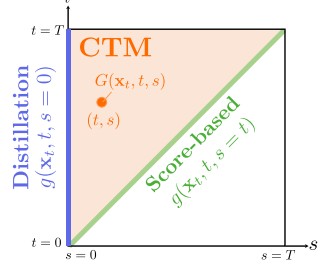

For stable training[3], we express $G$ as a mixture of $\mathbf{x}_t$ and a function $g$, (inspired from the Euler solver[4]):

$$G(\mathbf{x}_t, t, s) = \frac{s}{t}\mathbf{x}_t + \left(1 - \frac{s}{t}\right)g(\mathbf{x}_t, t, s),$$

where $g(\mathbf{x}_t, t, s) = \mathbf{x}_t + \frac{t}{t-s}\int_t^s \frac{\mathbf{x}_u - \mathbb{E}[\mathbf{x}|\mathbf{x}_u]}{u} \, \mathrm{d}u$. We predict

Figure 2: Learning objectives of Score-based ($t = s$ line), distillation ($s = 0$ line), and CTM (upper triangle).

$$G_{\boldsymbol{\theta}}(\mathbf{x}_t, t, s) = \frac{s}{t}\mathbf{x}_t + \left(1 - \frac{s}{t}\right)g_{\boldsymbol{\theta}}(\mathbf{x}_t, t, s)$$

as the neural jump, a combination of $\mathbf{x}_t$ and a neural output $g_{\boldsymbol{\theta}}$. This ensures the neural jump $G_{\boldsymbol{\theta}}$ satisfies the initial condition $G_{\boldsymbol{\theta}}(\mathbf{x}_t, t, t) = \mathbf{x}_t$ for free. It removes the necessity of enforcing the initial condition in neural network training, transforming the optimization problem from constrained to unconstrained. Figure 2 contrasts CTM's learning target with that of previous models.

A crucial characteristic of $g$ becomes evident when taking the limit as $s$ approaches $t$. From the definition, we obtain

$$\lim_{s \to t} g(\mathbf{x}_t, t, s) = \mathbf{x}_t + t \lim_{s \to t} \frac{1}{t-s} \int_t^s \frac{\mathbf{x}_u - \mathbb{E}[\mathbf{x}|\mathbf{x}_u]}{u} \, \mathrm{d}u = \mathbb{E}[\mathbf{x}|\mathbf{x}_t].$$

Therefore, estimating $g$ leads to the approximation of not only the $t$-to-$s$ jump but also the *infinitesimal* $t$-to-$t$ jump[5] (denoiser function). Indeed, from the Taylor expansion, we have

$$g(\mathbf{x}_t, t, s) = \mathbf{x}_t + \frac{t}{t-s} \int_t^s \frac{\mathbf{x}_u - \mathbb{E}[\mathbf{x}|\mathbf{x}_u]}{u} \, \mathrm{d}u = \mathbf{x}_t + \frac{t}{t-s}\left[(s-t)\frac{\mathbf{x}_t - \mathbb{E}[\mathbf{x}|\mathbf{x}_t]}{t} + \mathcal{O}((t-s)^2)\right]$$

$$= \mathbb{E}[\mathbf{x}|\mathbf{x}_t] + \mathcal{O}(|t-s|).$$

Therefore, $g(\mathbf{x}_t, t, s)$ (with general $s \leq t$) is interpreted as the denoiser function added with a residual term of the Taylor expansion.

## 3.2 DISTILLATION LOSS: SOFT CONSISTENCY LOSS

To achieve trajectory learning, CTM should match the neural jump $G_{\boldsymbol{\theta}}$ to the true jump $G$ by $G_{\boldsymbol{\theta}}(\mathbf{x}_t, t, s) \approx G(\mathbf{x}_t, t, s)$, for any $s \leq t$. We opt to train $G_{\boldsymbol{\theta}}$ by comparing with a solution of the numerical solver, $\texttt{Solver}(\mathbf{x}_t, t, s; \boldsymbol{\phi})$, of the pre-trained PF ODE in Eq. (1):

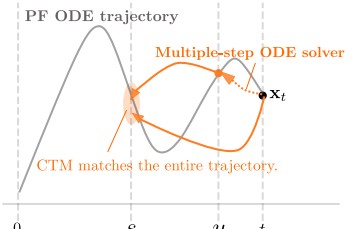

$$G_{\boldsymbol{\theta}}(\mathbf{x}_t, t, s) \approx \texttt{Solver}(\mathbf{x}_t, t, s; \boldsymbol{\phi}) \approx G(\mathbf{x}_t, t, s).$$

With a perfect teacher $\boldsymbol{\phi}$, $\texttt{Solver}$ accurately reconstructs $G(\mathbf{x}_t, t, s)$, and the optimal $G_{\boldsymbol{\theta}^*}(\mathbf{x}_t, t, s)$ coincides with the ground truth $G(\mathbf{x}_t, t, s)$, given sufficient student network flexibility.

Figure 3: An illustration of CTM's two predictions at time $s$ with an initial value $\mathbf{x}_t$.

For a more precise estimation of the entire solution trajectory, we introduce *soft consistency matching*. As illustrated in Figure 3, soft consistency compares two $s$-predictions: one from the teacher and the other from the student. More precisely, the target prediction is a mixture of teacher and student,

---

[3]Directly learning $\mathbf{x}_t + \int_t^s \frac{\mathbf{x}_u - \mathbb{E}[\mathbf{x}|\mathbf{x}_u]}{u}$ or $\int_t^s \frac{\mathbf{x}_u - \mathbb{E}[\mathbf{x}|\mathbf{x}_u]}{u}$ with a neural network can easily lead to divergence.

[4]Solving the PF ODE from $t$ to $s$ with a single-step Euler solver gives $\mathbf{x}_s^{\text{Euler}} = \mathbf{x}_t - (t-s)\frac{\mathbf{x}_t - \mathbb{E}[\mathbf{x}|\mathbf{x}_t]}{t} = \frac{s}{t}\mathbf{x}_t + (1 - \frac{s}{t})\mathbb{E}[\mathbf{x}|\mathbf{x}_t]$. Our scale choices of $\frac{s}{t}$ and $1 - \frac{s}{t}$, thus, are naturally derived from the Euler representation by replacing $\mathbb{E}[\mathbf{x}|\mathbf{x}_t]$ with $g$. We refer to Appendix C.1 for detailed analysis with this respect.

[5]In contrast, $G$ is unaware of this infinitesimal jump as it collapses to the identity function when $s \to t$, and estimating the denoiser with finite differences along the time derivative is imprecise due to numerical issues.

where we solve the teacher PF ODE on the $(u, t)$-interval and jump to $s$ using the stop-gradient student. In summary, soft consistency compares

$$G_{\boldsymbol{\theta}}(\mathbf{x}_t, t, s) \approx G_{\text{sg}(\boldsymbol{\theta})}\big(\text{Solver}(\mathbf{x}_t, t, u; \boldsymbol{\phi}), u, s\big), \tag{3}$$

where a random $u \in [s, t)$ determines the amount of teacher information to distill, and sg is exponential moving average stop-gradient $\text{sg}(\boldsymbol{\theta}) \leftarrow \text{stopgrad}(\mu\text{sg}(\boldsymbol{\theta}) + (1 - \mu)\boldsymbol{\theta})$.
By the choice of $u$, this soft matching spans two frameworks:

- At $u = s$, Eq. (3) enforces *global consistency*, i.e., the student distills the teacher information on the entire interval $(s, t)$.

- At $u = t - \Delta t$, Eq. (3) is *local consistency*, i.e., the student only distills the teacher information on a single-step interval $(t - \Delta t, t)$. Moreover, it becomes CM's distillation target when $s = 0$.

To quantify the dissimilarity between the student prediction $G_{\boldsymbol{\theta}}(\mathbf{x}_t, t, s)$ and the teacher prediction $G_{\text{sg}(\boldsymbol{\theta})}(\text{Solver}(\mathbf{x}_t, t, u; \boldsymbol{\phi}), u, s)$, we use a feature distance $d$ in clean data space by transporting two $s$-time predictions to 0-time using a stop-gradient student $G_{\text{sg}(\boldsymbol{\theta})}(\cdot, s, 0)$. More specifically, transported predictions become $\mathbf{x}_{\text{est}}(\mathbf{x}_t, t, s) := G_{\text{sg}(\boldsymbol{\theta})}(G_{\boldsymbol{\theta}}(\mathbf{x}_t, t, s), s, 0)$ and $\mathbf{x}_{\text{target}}(\mathbf{x}_t, t, u, s) := G_{\text{sg}(\boldsymbol{\theta})}(G_{\text{sg}(\boldsymbol{\theta})}(\text{Solver}(\mathbf{x}_t, t, u; \boldsymbol{\phi}), u, s), s, 0)$. Summing altogether, the CTM loss is defined as

$$\mathcal{L}_{\text{CTM}}(\boldsymbol{\theta}; \boldsymbol{\phi}) := \mathbb{E}_{t \in [0, T]}\mathbb{E}_{s \in [0, t]}\mathbb{E}_{u \in [s, t)}\mathbb{E}_{\mathbf{x}_0}\mathbb{E}_{\mathbf{x}_t | \mathbf{x}_0}\Big[d\big(\mathbf{x}_{\text{target}}(\mathbf{x}_t, t, u, s), \mathbf{x}_{\text{est}}(\mathbf{x}_t, t, s)\big)\Big],$$

which leads to the neural jump, at optimum, to match with the jump provided by solving the empirical PF ODE of Eq. (1), see Appendix B (Propositions 3 and 5) for details.

## 3.3 AUXILIARY LOSSES FOR BETTER TRAINING OF STUDENT

In knowledge distillation for classification problems, it is widely acknowledged that the student classifier often performs as well as, or even outperforms, the teacher classifier. A crucial factor contributing to this success is the direct training signal derived from the data label. More precisely, the student loss $\mathcal{L}_{\text{distill}}(\text{teacher}, \text{student}) + \mathcal{L}_{\text{cls}}(\text{data}, \text{student})$ combines a distillation loss $\mathcal{L}_{\text{distill}}$ and a classifier loss $\mathcal{L}_{\text{cls}}$, which provides a high-quality signal to the student with the data label.

However, in the context of generation tasks, distillation models tend to exhibit inferior sample quality compared to the teacher. This is primarily because model optimization relies solely on the distillation loss. In our approach, we extend the principles of classification distillation to our model by introducing direct signals from both DSM and adversarial losses to facilitate student learning.

First, we guide the student training with the DSM loss, given by

$$\mathcal{L}_{\text{DSM}}(\boldsymbol{\theta}) = \mathbb{E}_{t, \mathbf{x}_0}\mathbb{E}_{\mathbf{x}_t | \mathbf{x}_0}[\|\mathbf{x}_0 - g_{\boldsymbol{\theta}}(\mathbf{x}_t, t, t)\|_2^2].$$

The optimal $g_{\boldsymbol{\theta}*}$ obtained from the DSM loss becomes $g_{\boldsymbol{\theta}*}(\mathbf{x}_t, t, t) = \mathbb{E}[\mathbf{x}|\mathbf{x}_t] = g(\mathbf{x}_t, t, t)$[6]. Therefore, the DSM loss improves jump precision when $s \approx t$ by acting as a regularizer. We remark that the DSM loss mitigates the vanishing gradient problem of $g$ learning when $s \to t$ (because the scale factor $1 - \frac{s}{t} \to 0$) and significantly improves the accuracy of small neural jumps.

Second, we employ adversarial training for enhanced student learning, utilizing the GAN loss

$$\mathcal{L}_{\text{GAN}}(\boldsymbol{\theta}, \boldsymbol{\eta}) = \mathbb{E}_{\mathbf{x}_0}[\log d_{\boldsymbol{\eta}}(\mathbf{x}_0)] + \mathbb{E}_{t \in [0, T]}\mathbb{E}_{s \in [0, t]}\mathbb{E}_{\mathbf{x}_0}\mathbb{E}_{\mathbf{x}_t | \mathbf{x}_0}\big[\log\big(1 - d_{\boldsymbol{\eta}}(\mathbf{x}_{\text{est}}(\mathbf{x}_t, t, s))\big)\big],$$

where $d_{\boldsymbol{\eta}}$ is a discriminator. This adversarial loss is motivated by VQGAN (Esser et al., 2021), which shows that a combination of reconstruction and adversarial losses is beneficial for generation quality.

In summary, CTM incorporates distillation, DSM and GAN losses as

$$\mathcal{L}(\boldsymbol{\theta}, \boldsymbol{\eta}) := \mathcal{L}_{\text{CTM}}(\boldsymbol{\theta}; \boldsymbol{\phi}) + \lambda_{\text{DSM}}\mathcal{L}_{\text{DSM}}(\boldsymbol{\theta}) + \lambda_{\text{GAN}}\mathcal{L}_{\text{GAN}}(\boldsymbol{\theta}, \boldsymbol{\eta}),$$

---

[6]We opt to use the conventional DSM loss instead of the score matching loss of $\mathbb{E}[\|D_{\boldsymbol{\phi}}(\mathbf{x}_t, t) - g_{\boldsymbol{\theta}}(\mathbf{x}_t, t, t)\|_2^2]$ to ensure the exact denoiser estimation at the optimality.

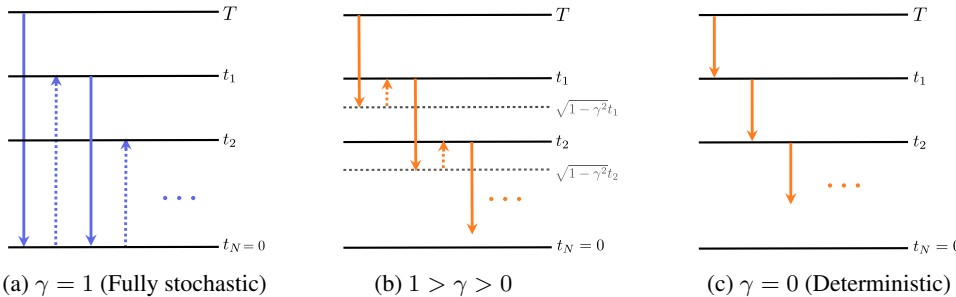

(a) $\gamma = 1$ (Fully stochastic)  (b) $1 > \gamma > 0$  (c) $\gamma = 0$ (Deterministic)

Figure 5: Illustration of $\gamma$-sampling with varying $\gamma$ value. It denoises with the network evaluation and iteratively diffuses the sample in reverse by $(t_n \xrightarrow{\text{Denoise}} \sqrt{1-\gamma^2}t_{n+1} \xrightarrow{\text{Noisify}} t_{n+1})_{n=0}^{N-1}$.

into a single and unified training framework[7]; and CTM solves the mini-max problem $\min_{\boldsymbol{\theta}} \max_{\boldsymbol{\eta}} \mathcal{L}(\boldsymbol{\theta}, \boldsymbol{\eta})$. Following VQGAN, we employ adaptive weighting with $\lambda_{\text{DSM}} = \frac{\|\nabla_{\boldsymbol{\theta}_L} \mathcal{L}_{\text{CTM}}(\boldsymbol{\theta};\phi)\|}{\|\nabla_{\boldsymbol{\theta}_L} \mathcal{L}_{\text{DSM}}(\boldsymbol{\theta})\|}$ and $\lambda_{\text{GAN}} = \frac{\|\nabla_{\boldsymbol{\theta}_L} \mathcal{L}_{\text{CTM}}(\boldsymbol{\theta};\phi)\|}{\|\nabla_{\boldsymbol{\theta}_L} \mathcal{L}_{\text{GAN}}(\boldsymbol{\theta};\boldsymbol{\eta})\|}$, where $\theta_L$ is the last layer of the UNet output block. This adaptive weighting significantly stabilizes the training by balancing the gradient scale of each term. Algorithm 4 summarizes CTM's training algorithm.

# 4 SAMPLING WITH CTM

CTM enables exact score evaluation through $g_{\boldsymbol{\theta}}(\boldsymbol{x}_t, t, t)$, supporting standard score-based sampling with ODE/SDE solvers. In high-dimensional image synthesis, as shown in Figure 4's left two columns, CTM performs comparably to EDM using Heun's method as a PF ODE solver.

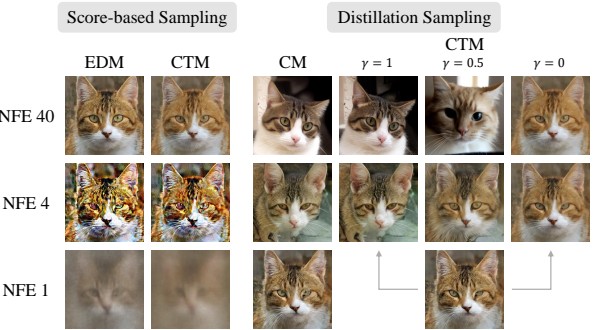

Figure 4: Comparison of score-based models (EDM), distillation models (CM), and CTM with various sampling methods and NFE trained on AFHQ-cat (Choi et al., 2020) $256 \times 256$.

CTM additionally enables time traversal along the solution trajectory, allowing for the newly introduced $\gamma$-sampling method, refer to Algorithm 2 and Figure 5. Suppose the sampling timesteps are $T = t_0 > \cdots > t_N = 0$. With $\mathbf{x}_{t_0} \sim \pi$, where $\pi$ is the prior distribution, $\gamma$-sampling denoises $\mathbf{x}_{t_0}$ to time $\sqrt{1-\gamma^2}t_1$ with $G_{\boldsymbol{\theta}}(\mathbf{x}_{t_0}, t_0, \sqrt{1-\gamma^2}t_1)$, and perturb this neural sample with forward diffusion to the noise level at time $t_1$. The $\gamma$-sampling iterates this back-and-forth traversal until reaching to time $t_N = 0$.

Our $\gamma$-sampling is a new distillation sampler that unifies previously proposed sampling techniques, including distillation sampling and score-based sampling.

- Figure 5-(a): When $\gamma = 1$, it coincides to the multistep sampling introduced in CM, which is fully stochastic and results in semantic variation when NFE changes, e.g., compare samples of NFE 4 and 40 with the deterministic sample of NFE 1 in the third column of Figure 4. With a fixed $\mathbf{x}_T$, CTM reproduces CM's samples in the fourth column of Figure 4.

- Figure 5-(c): When $\gamma = 0$, it becomes the deterministic distillation sampling that estimates the solution of the PF ODE. A key distinction between the $\gamma$-sampling and score-based sampling is that CTM avoids the discretization errors, e.g., compare (score-based) samples in the leftmost column and ($\gamma = 0$ distillation) samples in the rightmost column of Figure 4.

---

[7]The DSM loss is closely linked to the KL divergence (Song et al., 2021; Kim et al., 2022b) and the GAN loss is a proxy of $f$-divergences (Nowozin et al., 2016) or IPMs (Arjovsky et al., 2017). Therefore, our comprehensive loss can be interpreted as $\mathcal{L}_{\text{distill}}(\text{teacher, student}) + D_{\text{KL}}(\text{data}\|\text{student}) + D_f(\text{data, student})$, which combines the distillation loss (between teacher/student) with statistical divergences (between data/student).

Reference  $\gamma = 1$  $\gamma = 0$

Figure 6: $\gamma$ controls sample variance in stroke-based generation (see Appendix C.5).

- Figure 5-(b): When $0 < \gamma < 1$, it generalizes the EDM's stochastic sampler (Algorithm 1). Appendix B.4 shows that $\gamma$-sampling's sample variances scale proportionally with $\gamma^2$.

The optimal choice of $\gamma$ depends on practical usage and empirical configuration (Karras et al., 2022; Xu et al., 2023). Figure 6 demonstrates $\gamma$-sampling in stroke-based generation (Meng et al., 2021), revealing that the sampler with $\gamma = 1$ leads to significant semantic deviations from the reference stroke, while smaller $\gamma$ values yield closer semantic alignment and maintain high fidelity. Moreover, Figure 7 showcases $\gamma$'s impact on generation performance. In Figure 7-(a), $\gamma$ has less influence with small NFE, but the setup with $\gamma \approx 0$ is the only one that resembles the performance of the Heun's solver as NFE increases. Additionally, CM's multistep sampler ($\gamma = 1$) significantly degrades sample quality as NFE increases. This quality deterioration concerning $\gamma$ becomes more pronounced with higher NFEs, shown in Figure 7-(b), potentially attributed to the error accumulation during the iterative neural jump overlap to zero-time. We explain this phenomenon using a 2-steps $\gamma$-sampling example in the following theorem, see Theorem 8 for a generalized result for $N$ steps.

**Theorem 1** ((Informal) 2-steps $\gamma$-sampling). *Let $t \in (0, T)$ and $\gamma \in [0, 1]$. Let $p_{\boldsymbol{\theta}^*, 2}$ denote as the density obtained from the $\gamma$-sampler with the optimal CTM, following the transition sequence*

$$T \rightarrow \sqrt{1 - \gamma^2} t \rightarrow t \rightarrow 0, \text{ starting from } p_T. \text{ Then } D_{TV} \left( p_{data}, p_{\boldsymbol{\theta}^*, 2} \right)^8 = \mathcal{O}\left( \sqrt{T - \sqrt{1 - \gamma^2} t + t} \right).$$

When it becomes $N$ steps, the $\gamma$-sampling with $\gamma = 1$ iteratively conducts long jumps from $t_n$ to 0 for each step $n$, which aggregates the error to $\mathcal{O}(\sqrt{T + t_1 + \cdots + t_N})$. In contrast, such time overlap between jumps does not occur with $\gamma = 0$, eliminating the error accumulation, resulting in only $\mathcal{O}(\sqrt{T})$ error, see Appendix C.2. In summary, CTM addresses challenges associated with large NFE in distillation models with $\gamma = 0$ and removes the discretization error in score-based models.

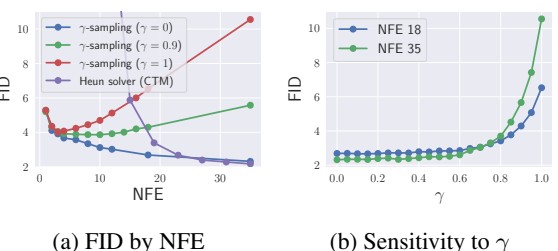

(a) FID by NFE  (b) Sensitivity to $\gamma$

Figure 7: (a) CTM enables score-based sampling and distillation $\gamma$-sampling on CIFAR-10. (b) The FID degrade highlights the importance of trajectory learning.

## 5 EXPERIMENTS

### 5.1 STUDENT (CTM) BEATS TEACHER (DM) – QUANTITATIVE ANALYSIS

We evaluate CTM on CIFAR-10 and ImageNet $64 \times 64$, using the pre-trained diffusion checkpoints from EDM (CIFAR-10) and CM (ImageNet) as the teacher models. We adopt EDM's training configuration for $\mathcal{L}_{\text{DSM}}(\boldsymbol{\theta})$ and employ StyleGAN-XL's (Sauer et al., 2022) discriminator for $\mathcal{L}_{\text{GAN}}(\boldsymbol{\theta}, \boldsymbol{\eta})$. For student, we use EDM's DDPM++ implementation on CIFAR-10; and CM's ADM implementation on ImageNet. In addition to these default architectures, we incorporate $s$-information via auxiliary temporal embedding with positional embedding (Vaswani et al., 2017), and add this embedding to the $t$-embedding. We minimally change the CM's design to comply the previous implementation, and we list important modifications in Table 4 in Appendix D.1: 1) We find that a large $\mu$ for stop-gradient EMA significantly stabilizes the adversarial training; 2) We evaluate the model performance with student EMA rate of 0.999; 3) we reuse the skip connection and output scaling of the pre-trained diffusion model to our neural output modeling: $g_{\boldsymbol{\theta}}(\mathbf{x}_t, t, s) = c_{\text{skip}}(t)\mathbf{x}_t + c_{\text{out}}(t)\text{NN}_{\boldsymbol{\theta}}(\mathbf{x}_t, t, s)$, where $\text{NN}_{\boldsymbol{\theta}}$ is a neural network. The selection of $c_{\text{skip}}$ and $c_{\text{out}}$ ensures that the initialized $g_{\boldsymbol{\theta}}(\mathbf{x}_t, t, t)$ closely

---

[8]The total variation of two densities $p$ and $q$ is defined as $D_{TV}(p, q) := \frac{1}{2} \int |p(\mathbf{x}) - q(\mathbf{x})| \, d\mathbf{x}$.

[9]Bold text indicates the best performance, while underlined text means the second best performance.

Table 1: Performance comparisons on CIFAR-10[9].

| Model | NFE | Unconditional | | Conditional |
|---|---|---|---|---|
| | | FID↓ | NLL↓ | FID↓ |
| **GAN Models** | | | | |
| BigGAN (Brock et al., 2018) | 1 | 8.51 | ✗ | - |
| StyleGAN-Ada (Karras et al., 2020) | 1 | 2.92 | ✗ | 2.42 |
| StyleGAN-D2D (Kang et al., 2021) | 1 | - | ✗ | 2.26 |
| StyleGAN-XL (Sauer et al., 2022) | 1 | - | ✗ | 1.85 |
| **Diffusion Models – Score-based Sampling** | | | | |
| DDPM (Ho et al., 2020) | 1000 | 3.17 | 3.75 | - |
| DDIM (Song et al., 2020a) | 100 | 4.16 | - | - |
| | 10 | 13.36 | - | - |
| Score SDE (Song et al., 2020a) | 2000 | 2.20 | 3.45 | - |
| VDM (Kingma et al., 2021) | 1000 | 7.41 | 2.49 | - |
| LSGM (Vahdat et al., 2021) | 138 | 2.10 | 3.43 | - |
| EDM (Karras et al., 2022) | 35 | 2.01 | 2.56 | 1.82 |
| **Diffusion Models – Distillation Sampling** | | | | |
| KD (Luhman & Luhman, 2021) | 1 | 9.36 | ✗ | - |
| DFNO (Zheng et al., 2023) | 1 | 3.78 | ✗ | - |
| 2-Rectified Flow (Liu et al., 2022) | 1 | 4.85 | ✗ | - |
| PD (Salimans & Ho, 2021) | 1 | 9.12 | ✗ | - |
| CD (official report) (Song et al., 2023) | 1 | 3.55 | ✗ | - |
| CD (retrained) | 1 | 10.53 | ✗ | - |
| CD + GAN (Lu et al., 2023) | 1 | 2.65 | ✗ | - |
| CTM (ours) | 1 | 1.98 | 2.43 | 1.73 |
| PD (Salimans & Ho, 2021) | 2 | 4.51 | - | - |
| CD (Song et al., 2023) | 2 | 2.93 | - | - |
| CTM (ours) | 2 | **1.87** | 2.43 | 1.63 |
| **Models without Pre-trained DM – Direct Generation** | | | | |
| CT | 1 | 8.70 | ✗ | - |
| CTM (ours) | 1 | 2.39 | - | - |

Table 2: Performance comparisons on ImageNet $64 \times 64$.

| Model | NFE | FID↓ | IS↑ | Rec↑ |
|---|---|---|---|---|
| Validation Data | | 1.41 | 64.10 | 0.67 |
| ADM (Dhariwal & Nichol, 2021) | 250 | 2.07 | - | 0.63 |
| EDM (Karras et al., 2022) | 79 | 2.44 | 48.88 | **0.67** |
| BigGAN-deep (Brock et al., 2018) | 1 | 4.06 | - | 0.48 |
| StyleGAN-XL (Sauer et al., 2022) | 1 | 2.09 | **82.35** | 0.52 |
| **Diffusion Models – Distillation Sampling** | | | | |
| PD (Salimans & Ho, 2021) | 1 | 15.39 | - | 0.62 |
| BOOT (Gu et al., 2023) | 1 | 16.3 | - | 0.36 |
| CD (Song et al., 2023) | 1 | 6.20 | 40.08 | 0.63 |
| CTM (ours) | 1 | 1.92 | 70.38 | 0.57 |
| PD (Salimans & Ho, 2021) | 2 | 8.95 | - | 0.65 |
| CD (Song et al., 2023) | 2 | 4.70 | - | 0.64 |
| CTM (ours) | 2 | **1.73** | 64.29 | 0.57 |

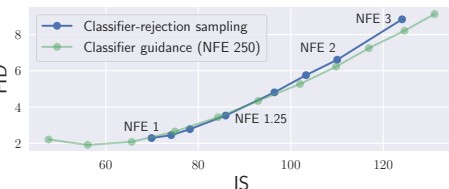

Figure 8: FID-IS curve on ImageNet.

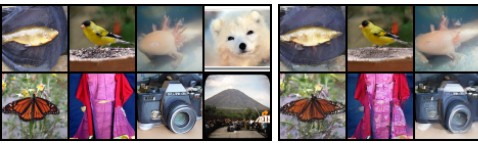
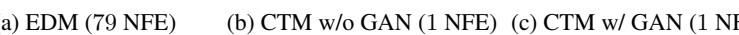

(a) EDM (79 NFE)    (b) CTM w/o GAN (1 NFE)    (c) CTM w/ GAN (1 NFE)

Figure 9: Samples generated by (a) EDM, (b) CTM without GAN ($\lambda_{\text{GAN}} = 0$), and (c) CTM with GAN (adaptive $\lambda_{\text{GAN}}$). More generated samples are demonstrated in Appendix E.

aligns with the pre-trained denoiser, with slight random noise introduced from the $s$-embedding. Reusing $c_{\text{skip}}$ and $c_{\text{out}}$ directs the student network to focus on training long jumps while preserving the accuracy of small jumps (via $\mathcal{L}_{\text{DSM}}$) from the initial training phase. Consequently, achieving good performance requires only 100K iterations (10x faster) for CIFAR-10 and 30K iterations (20x faster) for ImageNet, compared to corresponding baselines.

**CIFAR-10** CTM's NFE 1 FID (1.98) excels not only CM (3.55) on unconditional generation, but CTM (1.73) outperforms the SOTA models, such as EDM (1.82 with 35 NFE) and StyleGAN-XL (1.85) on conditional generation. In addition, CTM achieves the SOTA FID (1.63) with 2 NFEs, surpassing all previous generative models. These results on CIFAR-10 are obtained upon the official PyTorch code of CM, where retraining CM with their code yields FID of 10.53 (unconditional), significantly worse than the reported FID of 3.55. Additionally, CTM's ability to approximate scores using $g_{\boldsymbol{\theta}}(\mathbf{x}_t, t, t)$ enables evaluating Negative Log-Likelihood (NLL) (Song et al., 2021; Kim et al., 2022b), establishing a new SOTA NLL. This improvement can be attributed, in part, to CTM's reconstruction loss when $u = s$, and improved alignment with the oracle process (Lai et al., 2023a).

**ImageNet** CTM surpasses any previous non-guided generative models in FID. Also, CTM most closely resembles the IS of validation data, which implies that StyleGAN-XL tends to generate samples with a higher likelihood of being classified for a specific class, even surpassing the probabilities of real-world validation data, whereas CTM's generation is statistically consistent in terms of the classifier likelihood. In sample diversity, CTM reports an intermediate level of recall, but the random samples in Figure 16 exhibits the actual samples are comparably diverse to those of EDM or CM. Furthermore, the high likelihood of CTM on CIFAR-10 indirectly indicates that CTM has no issue on mode collapse. Lastly, we emphasize that all results in Tables 1 and 2 are achieved within 30K training iterations, requiring only 5% of the iterations needed to train CM and EDM.

**Classifier-Rejection Sampling** CTM's fast sampling enables classifier-rejection sampling. In the evaluation, for each class, we select the top 50 samples out of $\frac{50}{1-r}$ samples based on predicted class probability, where $r$ is the rejection ratio. This sampler, combined with NFE 1 sampling, consumes an average of NFE $\frac{1}{1-r}$. In Figure 8, CTM shows a FID-IS trade-off comparable to classifier-guided results (Ho & Salimans, 2021) achieved with high NFEs of 250 (see Figure 17 for samples).

## 5.2 QUALITATIVE ANALYSIS

**CTM Loss** Figure 10 highlights that soft consistency outperforms local consistency and performs comparable to global consistency. Specifically, local consistency distills only 1-step teacher, so the teacher of time interval $[0, T - \Delta t]$ is not used to train the neural jump starting from $\mathbf{x}_T$. Rather, teacher on $[t - \Delta t, t]$ with $t \in [0, T - \Delta t]$ is distilled to student

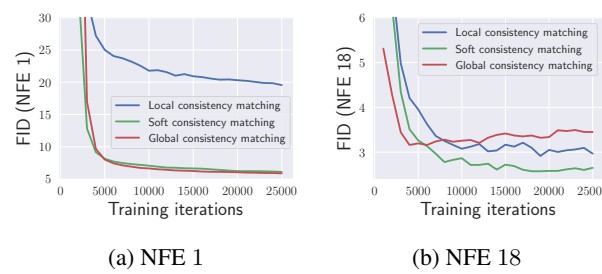

(a) NFE 1  (b) NFE 18

Figure 10: Comparison of local, global, and the proposed soft consistency matching.

from neural jump starting from $\mathbf{x}_t$, not $\mathbf{x}_T$. The student, thus, has to extrapolate the learnt but scattered teacher across time intervals to estimate the jump from $\mathbf{x}_T$, which could potentially lead to imprecise estimation. In contrast, the amount of teacher to be distilled in soft consistency is determined by a random $u$, where $u = 0$ represents distilling teacher on the entire interval $[0, T]$, see Appendix C.3. Hence, soft matching serves as a computationally efficient and high-performing loss.

**DSM Loss** Figure 11 illustrates two benefits of incorporating $\mathcal{L}_{\text{DSM}}$ with $\mathcal{L}_{\text{CTM}}$. It preserves sample quality for small NFE unless DSM scale outweighs CTM. For large NFE sampling, it significantly improves sample quality due to accurate score estimation. Throughout the paper, we maintain $\lambda_{\text{DSM}}$ to be the adaptive weight (the case of CTM +1.0DSM), based on insights from Figure 11.

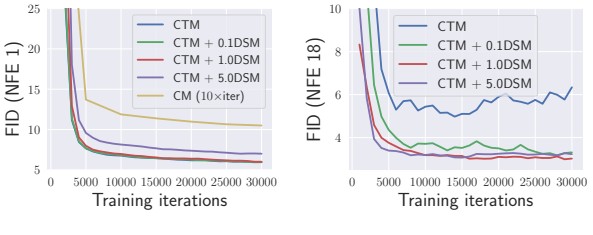

(a) NFE 1  (b) NFE 18

Figure 11: The effect of DSM loss.

**GAN Loss** Figure 12 highlights the benefits of integrating the GAN loss for both small- and large-NFE sample quality. In Figure 9, CTM shows superior sample production compared to the teacher, with GAN refining local details. Throughout the paper, we implement a GAN warm-up strategy: deactivating GAN training with $\lambda_{\text{GAN}} = 0$ during warm-up iterations

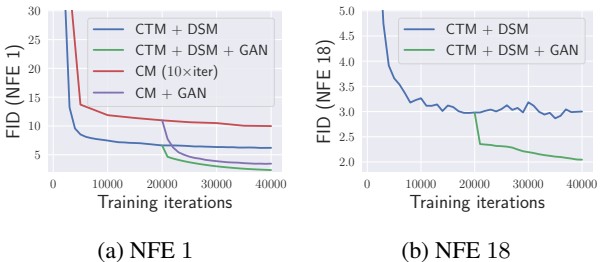

(a) NFE 1  (b) NFE 18

Figure 12: The effect of GAN loss.

and subsequently activating GAN training with an adaptive $\lambda_{\text{GAN}}$, following the VQGAN's approach. Additional insights into the effects of GAN on generated samples are discussed in Appendix C.4.

**Training Without Pre-trained DM** Leveraging our score learning capability, we replace the pre-trained score approximation, $D_\phi(\mathbf{x}_t, t)$, with CTM's approximation, $g_\theta(\mathbf{x}_t, t, t)$, allowing us to obtain the corresponding empirical PF ODE $d\mathbf{x}_t = \frac{\mathbf{x}_t - g_\theta(\mathbf{x}_t, t, t)}{t}$. Consequently, we can construct a pretrained-free target, $\hat{\mathbf{x}}_{\text{target}} := G_{\text{sg}(\theta)}\big(G_{\text{sg}(\theta)}(\texttt{Solver}(\mathbf{x}_t, t, u; \text{sg}(\theta))), u, s\big), s, 0)$, to replace $\mathbf{x}_{\text{target}}$ in computing the CTM loss $\mathcal{L}_{\text{CTM}}$. When incorporated with DSM and GAN losses, it achieves a NFE 1 FID of 2.39 on unconditional CIFAR-10, a performance on par with pre-trained DMs. Contrastive to CM, our CTM uses the identical form of loss from its score approximation capability.

## 6 CONCLUSION

CTM, a novel generative model, addresses issues in established models. With a unique training approach accessing intermediate PF ODE solutions, it enables unrestricted time traversal and seamless integration with prior models' training advantages. A universal framework for Consistency and Diffusion Models, CTM excels in both training and sampling. Remarkably, it surpasses its teacher model, achieving SOTA results in FID and likelihood for few-steps diffusion model sampling on CIFAR-10 and ImageNet $64 \times 64$, highlighting its versatility and process.

ETHICS STATEMENT

CTM poses a risk for generating harmful or inappropriate content, including deepfake images, graphic violence, or offensive material. Mitigating these risks involves the implementation of strong content filtering and moderation mechanisms to prevent the creation of unethical or harmful media content.

REPRODUCIBILITY STATEMENT

The code is available at https://github.com/sony/ctm. Moreover, we outline our training and sampling procedures in Algorithms 4 and 2, and detailed implementation instructions for result reproducibility can be found in Appendix D.

ACKNOWLEDGEMENT

We sincerely acknowledge the support of everyone who made this research possible. Our heartfelt thanks go to Koichi Saito, Woosung Choi, Kin Wai Cheuk, and Yukara Ikemiya for their assistance. Computational resource of AI Bridging Cloud Infrastructure (ABCI) provided by National Institute of Advanced Industrial Science and Technology (AIST) was used.

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

## CONTENTS

## A    RELATED WORKS

**Diffusion Models**   DMs excel in high-fidelity synthetic image and audio generation (Dhariwal & Nichol, 2021; Saharia et al., 2022; Rombach et al., 2022), as well as in applications like media editing, restoration (Meng et al., 2021; Cheuk et al., 2023; Kawar et al., 2022; Saito et al., 2023; Hernandez-Olivan et al., 2023; Murata et al., 2023). Recent research aims to enhance DMs in sample quality (Kim et al., 2022b;a), density estimation (Song et al., 2021; Lu et al., 2022a), and especially, sampling speed (Song et al., 2020a).

**Fast Sampling of DMs**   The SDE framework underlying DMs (Song et al., 2020b) has driven research into various numerical methods for accelerating DM sampling, exemplified by works such as (Song et al., 2020a; Zhang & Chen, 2022; Lu et al., 2022b). Notably, (Lu et al., 2022b) reduced the ODE solver steps to as few as 10-15. Other approaches involve learning the solution operator of ODEs (Zheng et al., 2023), discovering optimal transport paths for sampling (Liu et al., 2022), or employing distillation techniques (Luhman & Luhman, 2021; Salimans & Ho, 2021; Berthelot et al., 2023; Shao et al., 2023). However, previous distillation models may experience slow convergence or extended runtime. Gu et al. (2023) introduced a bootstrapping approach for data-free distillation. Furthermore, Song et al. (2023) introduced CM which extracts DMs' PF ODE to establish a direct mapping from noise to clean predictions, achieving one-step sampling while maintaining good sample quality. CM has been adapted to enhance the training stability of GANs, as (Lu et al., 2023). However, it's important to note that their focus does not revolve around achieving sampling acceleration for DMs, nor are the results restricted to simple datasets.

**Consistency of DMs**   Score-based generative models rely on a differential equation framework, employing neural networks trained on data to model the conversion between data and noise. These networks must satisfy specific consistency requirements due to the mathematical nature of the underlying equation. Early investigations, such as (Kim et al., 2022c), identified discrepancies between learned scores and ground truth scores. Recent developments have introduced various consistency concepts, showing their ability to enhance sample quality (Daras et al., 2023; Li et al., 2023), accelerate sampling speed (Song et al., 2023), and improve density estimation in diffusion modeling (Lai et al., 2023a). Notably, Lai et al. (2023b) established the theoretical equivalence of these consistency concepts, suggesting the potential for a unified framework that can empirically leverage their advantages. CTM can be viewed as the first framework which achieves all the desired properties.

## B    THEORETICAL INSIGHTS ON CTM

In this section, we explore several theoretical aspects of CTM, encompassing convergence analysis (Section B.2), properties of well-trained CTM, variance bounds for $\gamma$-sampling, and a more general form of accumulated errors induced by $\gamma$-sampling (cf. Theorem 1).

We first introduce and review some notions. Starting at time $t$ with an initial value of $\mathbf{x}_t$ and ending at time $s$, recall that $G(\mathbf{x}_t, t, s)$ represents the true solution of the PF ODE, and $G(\mathbf{x}_t, t, s; \boldsymbol{\phi})$ is the solution function of the following empirical PF ODE.

$$\frac{\mathrm{d}\mathbf{x}_u}{\mathrm{d}u} = \frac{\mathbf{x}_u - D_{\boldsymbol{\phi}}(\mathbf{x}_u, u)}{u}, \quad u \in [0, T]. \tag{4}$$

Here $\boldsymbol{\phi}$ denotes the teacher model's weights learned from DSM. Thus, $G(\mathbf{x}_t, t, s; \boldsymbol{\phi})$ can be expressed as

$$G(\mathbf{x}_t, t, s; \boldsymbol{\phi}) = \frac{s}{t}\mathbf{x}_t + (1 - \frac{s}{t})g(\mathbf{x}_t, t, s; \boldsymbol{\phi}),$$

where $g(\mathbf{x}_t, t, s; \boldsymbol{\phi}) = \mathbf{x}_t + \frac{t}{t-s}\int_t^s \frac{\mathbf{x}_u - D_{\boldsymbol{\phi}}(\mathbf{x}_u, u)}{u}\,\mathrm{d}u$. (Delbracio & Milanfar, 2023) also derived a similar formulation, albeit for different purposes.

### B.1    UNIFICATION OF SCORE-BASED AND DISTILLATION MODELS

The following lemma summarizes our dedicated $G$-expression using an auxiliary function $g$, allowing convenient access to both the integral via $G$ and the integrand via $g$, visualized in Figure 13.

**Lemma 2** (Unification of score-based and distillation models). *Suppose that the score satisfies* $\sup_{\mathbf{x}} \int_0^T \|\nabla \log p_u(\mathbf{x})\|_2 \, \mathrm{d}u < \infty$. *The solution $G(\mathbf{x}_t, t, s)$ can be expressed as:*

$$G(\mathbf{x}_t, t, s) = \frac{s}{t}\mathbf{x}_t + \left(1 - \frac{s}{t}\right)g(\mathbf{x}_t, t, s) \ \ \text{with} \ \ g(\mathbf{x}_t, t, s) = \mathbf{x}_t + \frac{t}{t-s}\int_t^s \frac{\mathbf{x}_u - \mathbb{E}[\mathbf{x}|\mathbf{x}_u]}{u} \, \mathrm{d}u.$$

*Here, g satisfies:*

- *When $s = 0$, $G(\mathbf{x}_t, t, 0) = g(\mathbf{x}_t, t, 0)$ is the solution of PF ODE at $s = 0$, initialized at $\mathbf{x}_t$.*

- *As $s \to t$, $g(\mathbf{x}_t, t, s) \to \mathbb{E}[\mathbf{x}|\mathbf{x}_t]$. Hence, g can be defined at $s = t$ by its limit: $g(\mathbf{x}_t, t, t) := \mathbb{E}[\mathbf{x}|\mathbf{x}_t]$.*

As outlined in Section 3.1, the $g$-expression for $G$ is inherently associated with the Taylor approximation to the integral:

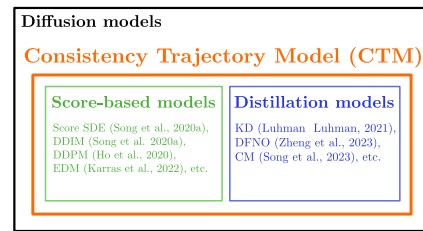

$$G(\mathbf{x}_t, t, s) = \mathbf{x}_t + \left[(s-t)\frac{\mathbf{x}_t - \mathbb{E}[\mathbf{x}|\mathbf{x}_t]}{t} + \mathcal{O}\left(|t-s|^2\right)\right]$$

$$= \frac{s}{t}\mathbf{x}_t + \left(1 - \frac{s}{t}\right)\underbrace{\left[\mathbb{E}[\mathbf{x}|\mathbf{x}_t] + \mathcal{O}\left(|t-s|\right)\right]}_{=g(\mathbf{x}_t, t, s)},$$

Figure 13: Schematic illustration of our CTM.

To further elucidate why Taylor's expansion is the primary cause of discretization errors, we will provide an explanation using the DDIM sampler with the oracle score function. The denoised sample with DDIM from $t$ to $t - \Delta t$ is $\mathbf{x}_{t-\Delta t}^{\text{DDIM}} = \left(1 - \frac{\Delta t}{t}\right)\mathbf{x}_t + \frac{\Delta t}{t}\mathbb{E}[\mathbf{x}|\mathbf{x}_t]$. However, the Taylor expansion of the integration yields that the true trajectory sample is $\mathbf{x}_{t-\Delta t}^{\text{true}} = \left(1 - \frac{\Delta t}{t}\right)\mathbf{x}_t + \frac{\Delta t}{t}\left(\mathbb{E}[\mathbf{x}|\mathbf{x}_t] + \mathcal{O}(\Delta t)\right)$. Therefore, the DDIM trajectory differs from the true trajectory by $\frac{\Delta t}{t}\mathcal{O}(\Delta t)$, which exactly represents the residual term of the Taylor expansion beyond the 2nd order. Consequently, the discretization error originates from the failure to estimate the residual term of the Taylor expansion.

### B.2 Convergence Analysis – Distillation from Teacher Models

**Convergence along Trajectory in a Time Discretization Setup.** CTM's practical implementation follows CM's one, utilizing discrete timesteps $t_0 = 0 < t_1 < \cdots < t_N = T$ for training. Initially, we assume local consistency matching for simplicity, but this can be extended to soft matching. This transforms the continuous time CTM loss to the discrete time counterpart:

$$\mathcal{L}_{\text{CTM}}^N(\boldsymbol{\theta}; \boldsymbol{\phi}) := \mathbb{E}_{n \in [\![1,N]\!]}\mathbb{E}_{m \in [\![0,n]\!]}\mathbb{E}_{\mathbf{x}_0, p_{0t_n}(\mathbf{x}|\mathbf{x}_0)}\left[d\left(\mathbf{x}_{\text{target}}(\mathbf{x}_{t_n}, t_n, t_m), \mathbf{x}_{\text{est}}(\mathbf{x}_{t_n}, t_n, t_m)\right)\right],$$

where $d(\cdot, \cdot)$ is a metric, and

$$\mathbf{x}_{\text{est}}(\mathbf{x}_{t_n}, t_n, t_m) := G_{\boldsymbol{\theta}}\left(G_{\boldsymbol{\theta}}(\mathbf{x}_{t_n}, t_n, t_m), t_m, 0\right)$$

$$\mathbf{x}_{\text{target}}(\mathbf{x}_{t_n}, t_n, t_{n-1}, t_m) := G_{\boldsymbol{\theta}}\left(G_{\boldsymbol{\theta}}\left(\texttt{Solver}(\mathbf{x}_{t_n}, t_n, t_{n-1}; \boldsymbol{\phi}), t_{n-1}, t_m\right), t_m, 0\right).$$

In the following theorem, we demonstrate that irrespective of the initial time $t_n$ and end time $t_m$, CTM $G_{\boldsymbol{\theta}}(\cdot, t_n, t_m; \boldsymbol{\phi})$, will eventually converge to its teacher model, $G(\cdot, t_n, t_m; \boldsymbol{\phi})$.

**Proposition 3.** *Define $\Delta_N t := \max_{n \in [\![1,N]\!]} \{|t_{n+1} - t_n|\}$. Assume that $G_{\boldsymbol{\theta}}$ is uniform Lipschitz in $\mathbf{x}$ and that the ODE solver admits local truncation error bounded uniformly by $\mathcal{O}((\Delta_N t)^{p+1})$ with $p \geq 1$. If there is a $\boldsymbol{\theta}_N$ so that $\mathcal{L}_{CTM}^N(\boldsymbol{\theta}_N; \boldsymbol{\phi}) = 0$, then for any $n \in [\![1, N]\!]$ and $m \in [\![1, n]\!]$*

$$\sup_{\mathbf{x} \in \mathbb{R}^D} d\left(G_{\boldsymbol{\theta}_N}(G_{\boldsymbol{\theta}_N}(\mathbf{x}, t_n, t_m), t_m, 0), G_{\boldsymbol{\theta}_N}(G(\mathbf{x}, t_n, t_m; \boldsymbol{\phi}), t_m, 0)\right) = \mathcal{O}((\Delta_N t)^p)(t_n - t_m).$$

Similar argument applies, confirming convergence along the PF ODE trajectory, ensuring the local consistency with $\boldsymbol{\theta}$ replacing $\texttt{sg}(\boldsymbol{\theta})$:

$$G_{\boldsymbol{\theta}}(\mathbf{x}_t, t, s) \approx G_{\boldsymbol{\theta}}(\texttt{Solver}(\mathbf{x}_t, t, t - \Delta t; \boldsymbol{\phi}), t - \Delta t, s)$$

by enforcing the following loss

$$\tilde{\mathcal{L}}_{\text{CTM}}^N(\boldsymbol{\theta}; \boldsymbol{\phi}) := \mathbb{E}_{n \in [\![1,N]\!]} \mathbb{E}_{m \in [\![0,n]\!]} \mathbb{E}_{\mathbf{x}_0, p_{0t_n}(\mathbf{x}|\mathbf{x}_0)} \Big[ d\big( \tilde{\mathbf{x}}_{\text{target}}(\mathbf{x}_{t_n}, t_n, t_m), \tilde{\mathbf{x}}_{\text{est}}(\mathbf{x}_{t_n}, t_n, t_m) \big) \Big],$$

where

$$\tilde{\mathbf{x}}_{\text{est}}(\mathbf{x}_{t_n}, t_n, t_m) := G_{\boldsymbol{\theta}}(\mathbf{x}_{t_n}, t_n, t_m)$$

$$\tilde{\mathbf{x}}_{\text{target}}(\mathbf{x}_{t_n}, t_n, t_{n-1}, t_m) := G_{\boldsymbol{\theta}}\big( \texttt{Solver}(\mathbf{x}_{t_n}, t_n, t_{n-1}; \boldsymbol{\phi}), t_{n-1}, t_m \big).$$

**Proposition 4.** *If there is a $\boldsymbol{\theta}_N$ so that $\tilde{\mathcal{L}}_{CTM}^N(\boldsymbol{\theta}_N; \boldsymbol{\phi}) = 0$, then for any $n \in [\![1,N]\!]$ and $m \in [\![1,n]\!]$*

$$\sup_{\mathbf{x} \in \mathbb{R}^D} d\big( G_{\boldsymbol{\theta}_N}(\mathbf{x}, t_n, t_m), G(\mathbf{x}, t_n, t_m; \boldsymbol{\phi}) \big) = \mathcal{O}((\Delta_N t)^p)(t_n - t_m).$$

**Convergence of Densities.** In Proposition 3, we demonstrated point-wise trajectory convergence, from which we infer that CTM may converge to its training target in terms of density. More precisely, in Proposition 5, we establish that if CTM's target $\mathbf{x}_{\text{target}}$ is derived from the teacher model (as defined above), then the data density induced by CTM will converge to that of the teacher model. Specifically, if the target $\mathbf{x}_{\text{target}}$ perfectly approximates the true $G$-function:

$$\mathbf{x}_{\text{target}}(\mathbf{x}_{t_n}, t_n, t_{n-1}, t_m) \equiv G(\mathbf{x}_{t_n}, t_n, t_m), \quad \text{for all } n \in [\![1,N]\!], m \in [\![0,n]\!], N \in \mathbb{N}. \quad (5)$$

Then the data density generated by CTM will ultimately learn the data distribution $p_{\text{data}}$.

Simplifying, we use the $\ell_2$ for the distance metric $d$ and consider the prior distribution $\pi$ to be $p_T$, which is the marginal distribution at time $t = T$ defined by the diffusion process:

$$\mathrm{d}\mathbf{x}_t = \sqrt{2t}\,\mathrm{d}\mathbf{w}_t, \quad (6)$$

**Proposition 5.** *Suppose that*

(i) *The uniform Lipschitzness of $G_{\boldsymbol{\theta}}$ (and $G$),*

$$\sup_{\boldsymbol{\theta}} \|G_{\boldsymbol{\theta}}(\mathbf{x}, t, s) - G_{\boldsymbol{\theta}}(\mathbf{x}', t, s)\|_2 \leq L \|\mathbf{x} - \mathbf{x}'\|_2, \quad \text{for all } \mathbf{x}, \mathbf{x}' \in \mathbb{R}^D, t, s \in [0, T],$$

(ii) *The uniform boundedness in $\boldsymbol{\theta}$ of $G_{\boldsymbol{\theta}}$: there is a $L(\mathbf{x}) \geq 0$ so that*

$$\sup_{\boldsymbol{\theta}} \|G_{\boldsymbol{\theta}}(\mathbf{x}, t, s)\|_2 \leq L(\mathbf{x}) < \infty, \quad \text{for all } \mathbf{x} \in \mathbb{R}^D, t, s \in [0, T]$$

*If for any $N$, there is a $\boldsymbol{\theta}_N$ such that $\mathcal{L}_{CTM}^N(\boldsymbol{\theta}_N; \boldsymbol{\phi}) = 0$. Let $p_{\boldsymbol{\theta}_N}(\cdot)$ denote the pushforward distribution of $p_T$ induced by $G_{\boldsymbol{\theta}_N}(\cdot, T, 0)$. Then, as $N \to \infty$, $\|p_{\boldsymbol{\theta}_N}(\cdot) - p_{\boldsymbol{\phi}}(\cdot)\|_\infty \to 0$. Particularly, if the condition in Eq. (5) is satisfied, then $\|p_{\boldsymbol{\theta}_N}(\cdot) - p_{data}(\cdot)\|_\infty \to 0$ as $N \to \infty$.*

### B.3 Non-Intersecting Trajectory of the Optimal CTM

CTM learns distinct trajectories originating from various initial points $\mathbf{x}t$ and times $t$. In the following proposition, we demonstrate that the distinct trajectories derived by the optimal CTM, which effectively distills information from its teacher model ($G_{\boldsymbol{\theta}^*}(\cdot, t, s) \equiv G(\cdot, t, s; \boldsymbol{\phi})$ for any $t, s \in [0, T]$), do not intersect.

**Proposition 6.** *Suppose that a well-trained $\boldsymbol{\theta}^*$ such that $G_{\boldsymbol{\theta}^*}(\cdot, t, s) \equiv G(\cdot, t, s; \boldsymbol{\phi})$ for any $t, s \in [0, T]$, and that $D_{\boldsymbol{\phi}}(\cdot, t)$ is Lipschitz, i.e., there is a constant $L_{\boldsymbol{\phi}} > 0$ so that for any $\mathbf{x}, \mathbf{y} \in \mathbb{R}^D$ and $t \in [0, T]$*

$$\|D_{\boldsymbol{\phi}}(\mathbf{x}, t) - D_{\boldsymbol{\phi}}(\mathbf{y}, t)\|_2 \leq L_{\boldsymbol{\phi}} \|\mathbf{x} - \mathbf{y}\|_2.$$

*Then for any $s \in [0, t]$, the mapping $G_{\boldsymbol{\theta}^*}(\cdot, t, s) \colon \mathbb{R}^D \to \mathbb{R}^D$ is bi-Lipschitz. Namely, for any $\mathbf{x}_t, \mathbf{y}_t \in \mathbb{R}^D$*

$$e^{-L_{\boldsymbol{\phi}}(t-s)} \|\mathbf{x}_t - \mathbf{y}_t\|_2 \leq \|G_{\boldsymbol{\theta}^*}(\mathbf{x}_t, t, s) - G_{\boldsymbol{\theta}^*}(\mathbf{y}_t, t, s)\|_2 \leq e^{L_{\boldsymbol{\phi}}(t-s)} \|\mathbf{x}_t - \mathbf{y}_t\|_2. \quad (7)$$

*This implies that $\mathbf{x}_t \neq \mathbf{y}_t$, $G_{\boldsymbol{\theta}^*}(\mathbf{x}_t; t, s) \neq G_{\boldsymbol{\theta}^*}(\mathbf{y}_t; t, s)$ for all $s \in [0, t]$.*

Specifically, the mapping from an initial value to its corresponding solution trajectory, denoted as $\mathbf{x}_t \mapsto G_{\boldsymbol{\theta}^*}(\mathbf{x}_t, t, \cdot)$, is injective. Conceptually, this ensures that if we use guidance at intermediate times to shift a point to another guided-target trajectory, the guidance will continue to affect the outcome at $t = 0$.

### B.4 Variance Bounds of $\gamma$-sampling

Suppose the sampling timesteps are $T = t_0 > t_1 > \cdots > t_N = 0$. In Proposition 7, we analyze the variance of

$$X_{n+1} := G_{\boldsymbol{\theta}}(X_n, t_n, \sqrt{1-\gamma^2}t_{n+1}) + Z_n,$$

resulting from $n$-step $\gamma$-sampling, initiated at

$$X_1 := G_{\boldsymbol{\theta}}(\mathbf{x}_{t_0}, t_0, \sqrt{1-\gamma^2}t_1) + \gamma Z_0, \quad \text{where } Z_n \overset{\text{iid}}{\sim} \mathcal{N}(\mathbf{0}, \gamma^2 t_{n+1}^2)\mathbf{I}).$$

Here, we assume an optimal CTM which precisely distills information from the teacher model $G_{\boldsymbol{\theta}^*}(\cdot) = G(\cdot, t, s; \boldsymbol{\phi})$ for all $t, s \in [0, T]$, for simplicity.

**Proposition 7.** *We have*

$$\zeta^{-1}(t_n, t_{n+1}, \gamma)\text{Var}(X_n) + \gamma^2 t_{n+1}^2 \leq \text{Var}(X_{n+1}) \leq \zeta(t_n, t_{n+1}, \gamma)\text{Var}(X_n) + \gamma^2 t_{n+1}^2,$$

*where $\zeta(t_n, t_{n+1}, \gamma) = \exp\left(2L_{\boldsymbol{\phi}}(t_n - \sqrt{1-\gamma^2}t_{n+1})\right)$ and $L_{\boldsymbol{\phi}}$ is a Lipschitz constant of $D_{\boldsymbol{\phi}}(\cdot, t)$.*

In line with our intuition, CM's multistep sampling ($\gamma = 1$) yields a broader range of $\text{Var}(X_{n+1})$ compared to $\gamma = 0$, resulting in diverging semantic meaning with increasing sampling NFE.

### B.5 Accumulated Errors in the General Form of $\gamma$-sampling.

We can extend Theorem 1 for two steps $\gamma$-sampling for the case of multisteps.

We begin by clarifying the concept of "density transition by a function". For a measurable mapping $\mathcal{T} : \Omega \to \Omega$ and a measure $\nu$ on the measurable space $\Omega$, the notation $\mathcal{T}\sharp\nu$ denotes the pushforward measure, indicating that if a random vector $X$ follows the distribution $\nu$, then $\mathcal{T}(X)$ follows the distribution $\mathcal{T}\sharp\nu$.

Given a sampling timestep $T = t_0 > t_1 > \cdots > t_N = 0$. Let $p_{\boldsymbol{\theta}^*, N}$ represent the density resulting from N-steps of $\gamma$-sampling initiated at $p_T$. That is,

$$p_{\boldsymbol{\theta}^*, N} := \bigcirc_{n=0}^{N-1}\left(\mathcal{T}_{\sqrt{1-\gamma^2}t_{n+1} \to t_{n+1}}^{\boldsymbol{\theta}^*} \circ \mathcal{T}_{t_n \to \sqrt{1-\gamma^2}t_{n+1}}^{\boldsymbol{\theta}^*}\right)\sharp p_T.$$

Here, $\bigcirc_{n=0}^{N-1}$ denotes the sequential composition. We assume an optimal CTM which precisely distills information from the teacher model $G_{\boldsymbol{\theta}^*}(\cdot) = G(\cdot, t, s; \boldsymbol{\phi})$ for all $t, s \in [0, T]$.

**Theorem 8** (Accumulated errors of N-steps $\gamma$-sampling). *Let $\gamma \in [0, 1]$.*

$$D_{TV}(p_{data}, p_{\boldsymbol{\theta}^*, N}) = \mathcal{O}\left(\sum_{n=0}^{N-1}\sqrt{t_n - \sqrt{1-\gamma^2}t_{n+1}}\right).$$

*Here, $\mathcal{T}_{t \to s} : \mathbb{R}^D \to \mathbb{R}^D$ denotes the oracle transition mapping from $t$ to $s$, determined by Eq. (6). The pushforward density via $\mathcal{T}_{t \to s}$ is denoted as $\mathcal{T}_{t \to s}\sharp p_t$, with similar notation applied to $\mathcal{T}_{t \to s}^{\boldsymbol{\theta}^*}\sharp p_t$, where $\mathcal{T}_{t \to s}^{\boldsymbol{\theta}^*}$ denotes the transition mapping associated with the optimal CTM trained with $\mathcal{L}_{CTM}$.*

### B.6 Transition Densities with the Optimal CTM

In this section, for simplicity, we assume the optimal CTM, $G_{\boldsymbol{\theta}^*} \equiv G$ with a well-learned $\boldsymbol{\theta}^*$, which recovers the true $G$-function. We establish that the density propagated by this optimal CTM from any time $t$ to a subsequent time $s$ aligns with the predefined density determined by the fixed forward process.

We now present the proposition ensuring alignment of the transited density.

**Proposition 9.** *Let $\{p_t\}_{t=0}^T$ be densities defined by the diffusion process Eq. (6), where $p_0 := p_{data}$. Denote $\mathcal{T}_{t \to s}(\cdot) := G(\cdot, t, s) : \mathbb{R}^D \to \mathbb{R}^D$ for any $t \geq s$. Suppose that the score $\nabla \log p_t$ satisfies that there is a function $L(t) \geq 0$ so that $\int_0^T |L(t)|\, dt < \infty$ and*

(i) *Linear growth:* $\|\nabla \log p_t(\mathbf{x})\|_2 \leq L(t)(1 + \|\mathbf{x}\|_2)$, *for all* $\mathbf{x} \in \mathbb{R}^D$

(ii) *Lipschitz:* $\|\nabla \log p_t(\mathbf{x}) - \nabla \log p_t(\mathbf{y})\|_2 \leq L(t) \|\mathbf{x} - \mathbf{y}\|_2$, *for all* $\mathbf{x}, \mathbf{y} \in \mathbb{R}^D$.

*Then for any* $t \in [0, T]$ *and* $s \in [0, t]$, $p_s = \mathcal{T}_{t \to s} \sharp p_t$.

This theorem guarantees that by learning the optimal CTM, which possesses complete trajectory information, we can retrieve all true densities at any time using CTM.

## C  ALGORITHMIC DETAILS

### C.1  MOTIVATION OF PARAMETRIZATION

Our parametrization of $G_\theta$ is affected from the discretized ODE solvers. For instance, the one-step Euler solver has the solution of

$$\mathbf{x}_s^{\text{Euler}} = \mathbf{x}_t - (t - s)\frac{\mathbf{x}_t - \mathbb{E}[\mathbf{x}|\mathbf{x}_t]}{t} = \frac{s}{t}\mathbf{x}_t + \left(1 - \frac{s}{t}\right)\mathbb{E}[\mathbf{x}|\mathbf{x}_t].$$

The one-step Heun solver is

$$\begin{aligned}
\mathbf{x}_s^{\text{Heun}} &= \mathbf{x}_t - \frac{t - s}{2}\left(\frac{\mathbf{x}_t - \mathbb{E}[\mathbf{x}|\mathbf{x}_t]}{t} + \frac{\mathbf{x}_s^{\text{Euler}} - \mathbb{E}[\mathbf{x}|\mathbf{x}_s^{\text{Euler}}]}{s}\right) \\
&= \mathbf{x}_t - \frac{t - s}{2}\left(\frac{\mathbf{x}_t}{t} + \frac{\mathbf{x}_s^{\text{Euler}}}{s}\right) + \frac{t - s}{2}\left(\frac{\mathbb{E}[\mathbf{x}|\mathbf{x}_t]}{t} + \frac{\mathbb{E}[\mathbf{x}|\mathbf{x}_s^{\text{Euler}}]}{s}\right) \\
&= \frac{s}{t}\mathbf{x}_t + \left(1 - \frac{s}{t}\right)\left(\left(1 - \frac{t}{2s}\right)\mathbb{E}[\mathbf{x}|\mathbf{x}_t] + \frac{t}{2s}\mathbb{E}[\mathbf{x}|\mathbf{x}_s^{\text{Euler}}]\right).
\end{aligned}$$

Again, the solver scales $\mathbf{x}_t$ with $\frac{s}{t}$ and multiply $1 - \frac{s}{t}$ to the second term. Therefore, our $G(\mathbf{x}_t, t, s) = \frac{s}{t}\mathbf{x}_t + (1 - \frac{s}{t})g(\mathbf{x}_t, t, s)$ is a natural way to represent the ODE solution.

For future research, we establish conditions enabling access to both integral and integrand expressions. Consider a continuous real-valued function $a(t, s)$. We aim to identify necessary conditions on $a(t, s)$ for the expression of $G$ as:

$$G(\mathbf{x}_t, t, s) = a(t, s)\mathbf{x}_t + (1 - a(t, s))h(\mathbf{x}_t, t, s),$$

for a vector-valued function $h(\mathbf{x}_t, t, s)$ and that $h$ satisfies:

- $\lim_{s \to t} h(\mathbf{x}_t, t, s)$ exists;
- it can be expressed algebraically with $\mathbb{E}[\mathbf{x}|\mathbf{x}_t]$.

Starting with the definition of $G$, we can obtain

$$\begin{aligned}
G(\mathbf{x}_t, t, s) &= \mathbf{x}_t + \int_t^s \frac{\mathbf{x}_u - \mathbb{E}[\mathbf{x}|\mathbf{x}_u]}{u}\,\mathrm{d}u \\
&= a(t, s)\mathbf{x}_t + (1 - a(t, s))\underbrace{\left[\mathbf{x}_t + \frac{1}{1 - a(t, s)}\int_t^s \frac{\mathbf{x}_u - \mathbb{E}[\mathbf{x}|\mathbf{x}_u]}{u}\,\mathrm{d}u\right]}_{h(\mathbf{x}_t, t, s)}.
\end{aligned}$$

Suppose that there is a continuous function $c(t)$ so that

$$\lim_{s \to t} \frac{s - t}{1 - a(t, s)} = c(t),$$

then

$$\begin{aligned}
\lim_{s \to t} h(\mathbf{x}_t, t, s) &= \mathbf{x}_t + \lim_{s \to t}\left[\frac{1}{1 - a(t, s)}\int_t^s \frac{\mathbf{x}_u - \mathbb{E}[\mathbf{x}|\mathbf{x}_u]}{u}\,\mathrm{d}u\right] \\
&= \mathbf{x}_t + \lim_{s \to t}\left[\frac{s - t}{1 - a(t, s)}\frac{\mathbf{x}_{t^*} - \mathbb{E}[\mathbf{x}|\mathbf{x}_{t^*}]}{t^*}\right], \quad \text{for some } t^* \in [s, t]
\end{aligned}$$

$$= \mathbf{x}_t + c(t)\Big(\frac{\mathbf{x}_t - \mathbb{E}[\mathbf{x}|\mathbf{x}_t]}{t}\Big)$$

$$= \Big(\frac{1 + c(t)}{t}\Big)\mathbf{x}_t - \frac{c(t)}{t}\mathbb{E}[\mathbf{x}|\mathbf{x}_t].$$

The second equality follows from the mean value theorem (We omit the continuity argument details for Markov filtrations). Therefore, we obtain the desired property 2). We summarize the necessary conditions on $a(s,t)$ as:

$$\text{There is some continuous function } c(t) \text{ in } t \text{ so that } \lim_{s \to t} \frac{s - t}{1 - a(t, s)} = c(t). \tag{8}$$

We now explain the above observation with an example by considering EDM-type parametrization. Consider $c_{\text{skip}}(t, s) := \sqrt{\frac{(s - \sigma_{\min})^2 + \sigma_{\text{data}}^2}{(t - \sigma_{\min})^2 + \sigma_{\text{data}}^2}}$ and $c_{\text{out}}(t, s) := (1 - \frac{s}{t})$. Then $G(\mathbf{x}_t, t, s)$ can be expressed as

$$G(\mathbf{x}_t, t, s) = c_{\text{skip}}(t, s)\mathbf{x}_t + c_{\text{out}}(t, s)h(\mathbf{x}_t, t, s),$$

where $h$ is defined as

$$h(\mathbf{x}_t, t, s) = \frac{1}{c_{\text{out}}}\Big[(1 - c_{\text{skip}})\mathbf{x}_t + \int_t^s \frac{\mathbf{x}_u - \mathbb{E}[\mathbf{x}|\mathbf{x}_u]}{u}\,\mathrm{d}u\Big].$$

Then, we can verify that $c_{\text{skip}}(t, s)$ satisfies the condition in Eq. (8) and that

$$\mathbb{E}[\mathbf{x}|\mathbf{x}_t] = g(\mathbf{x}_t, t, t) + \frac{\sigma_{\min}^2 + \sigma_{\text{data}}^2 - \sigma_{\min}t}{(t - \sigma_{\min})^2 + \sigma_{\text{data}}^2}\mathbf{x}_t.$$

The DSM loss with this $c_{\text{skip}}(t, s)$ becomes

$$\mathcal{L}_{\text{DM}}(\boldsymbol{\theta}) = \mathbb{E}\Bigg[\Big\|\mathbf{x}_0 - \Big(g_{\boldsymbol{\theta}}(\mathbf{x}_t, t, t) + \frac{\sigma_{\min}^2 + \sigma_{\text{data}}^2 - \sigma_{\min}t}{(t - \sigma_{\min})^2 + \sigma_{\text{data}}^2}\mathbf{x}_t\Big)\Big\|_2^2\Bigg]$$

However, empirically, we find that the parametrization of $c_{\text{skip}}(t, s)$ and $c_{\text{out}}(t, s)$ other than the ODE solver-oriented one, i.e., $c_{\text{skip}}(t, s) = \frac{s}{t}$ and $c_{\text{skip}}(t, s) = 1 - \frac{s}{t}$, faces training instability. Therefore, we set $G(\mathbf{x}_t, t, s) = \frac{s}{t}\mathbf{x}_t + (1 - \frac{s}{t})g(\mathbf{x}_t, t, s)$ as our default design and estimate $g$-function with the neural network.

### C.2 CHARACTERISTICS OF $\gamma$-SAMPLING

**Connection with SDE** When $G_{\boldsymbol{\theta}} = G$, a single step of $\gamma$-sampling is expressed as:

$$\mathbf{x}_{t_{n+1}}^{\gamma} = \mathbf{x}_{t_n} + G(\mathbf{x}_{t_n}, t_n, \sqrt{1 - \gamma^2}t_{n+1}) + \gamma t_{n+1}\boldsymbol{\epsilon}$$

$$= \mathbf{x}_{t_n} - \Bigg(\underbrace{\int_{t_n}^{t_{n+1}} u\nabla \log p_u(\mathbf{x}_u)\,\mathrm{d}u}_{\text{past information}} + \underbrace{\int_{t_{n+1}}^{\sqrt{1-\gamma^2}t_{n+1}} u\nabla \log p_u(\mathbf{x}_u)\,\mathrm{d}u}_{\text{future information}}\Bigg) + \gamma t_{n+1}\boldsymbol{\epsilon},$$

where $\boldsymbol{\epsilon} \sim \mathcal{N}(0, \mathbf{I})$. This formulation cannot be interpreted as a differential form (Øksendal, 2003) because it look-ahead future information (from $t_{n+1}$ to $\sqrt{1 - \gamma^2}t_{n+1}$) to generate the sample $\mathbf{x}_{t_{n+1}}^{\gamma}$ at time $t_{n+1}$. This suggests that there is no Itô's SDE that corresponds to our $\gamma$-sampler pathwisely, opening up new possibilities for the development of a new family of diffusion samplers.

**Connection with EDM's stochastic sampler** We conduct a direct comparison between EDM's stochastic sampler and CTM's $\gamma$-sampling. We denote `Heun`$(\mathbf{x}_t, t, s)$ as Heun's solver initiated at time $t$ and point $\mathbf{x}_t$ and ending at time $s$. It's worth noting that EDM's sampler inherently experiences discretization errors stemming from the use of Heun's solver, while CTM is immune to such errors.

| **Algorithm 1** EDM's sampler |
| --- |
| 1: Start from $\mathbf{x}_{t_0} \sim \pi$ |
| 2: **for** $n = 0$ to $N - 1$ **do** |
| 3:     $\hat{t}_n \leftarrow (1 + \gamma)t_n$ |
| 4:     Diffuse $\mathbf{x}_{\hat{t}_n} \leftarrow \mathbf{x}_{t_n} + \sqrt{\hat{t}_n^2 - t_n^2}\boldsymbol{\epsilon}$ |
| 5:     Denoise $\mathbf{x}_{t_{n+1}} \leftarrow \texttt{Heun}(\mathbf{x}_{\hat{t}_n}, \hat{t}_n, t_{n+1})$ |
| 6: **end for** |
| 7: **Return** $\mathbf{x}_{t_N}$ |

| **Algorithm 2** CTM's $\gamma$-sampling |
| --- |
| 1: Start from $\mathbf{x}_{t_0} \sim \pi$ |
| 2: **for** $n = 0$ to $N - 1$ **do** |
| 3:     $\tilde{t}_{n+1} \leftarrow \sqrt{1 - \gamma^2}t_{n+1}$ |
| 4:     Denoise $\mathbf{x}_{\tilde{t}_{n+1}} \leftarrow G_{\boldsymbol{\theta}}(\mathbf{x}_{t_n}, t_n, \tilde{t}_{n+1})$ |
| 5:     Diffuse $\mathbf{x}_{t_{n+1}} \leftarrow \mathbf{x}_{\hat{t}_{n+1}} + \gamma t_{n+1}\boldsymbol{\epsilon}$ |
| 6: **end for** |
| 7: **Return** $\mathbf{x}_{t_N}$ |

The primary distinction between EDM's stochastic sampling in Algorithm 1 and CTM's $\gamma$-sampling in Algorithm 2 is the order of the forward (diffuse) and backward (denoise) steps. However, through the iterative process of forward-backward time traveling, these two distinct samplers become indistinguishable. Aside from the order of forward-backward steps, the two algorithms essentially align if we opt to synchronize the CTM's time $(t_n^{\text{CTM}}, \tilde{t}_n^{\text{CTM}})$ to with the EDM's time $(\hat{t}_n^{\text{EDM}}, t_{n+1}^{\text{EDM}})$, respectively, and their $\gamma$s accordingly.

## C.3   ALGORITHMIC COMPARISON OF LOCAL CONSISTENCY AND SOFT CONSISTENCY

In this subsection, we explain the algorithmic difference between the local consistency loss and the soft consistency loss focusing on how the neural jump $G_{\boldsymbol{\theta}}(\mathbf{x}_T, T, 0)$ is trained.

**Local Consistency (Implicit Information from Teacher)** Let us assume that at some training iteration the maximum time $T$ is sampled as a random time $t$. Then CM matches the long jump provided by $G_{\boldsymbol{\theta}}(\mathbf{x}_T, T, 0)$ and $G_{\text{sg}(\boldsymbol{\theta})}(\texttt{Solver}(\mathbf{x}_T, T, T - \Delta t), T - \Delta t, 0)$. Hence, the neural jump $G_{\boldsymbol{\theta}}(\mathbf{x}_T, T, 0)$ distills on the teacher information within the interval $[T - \Delta t, T]$ and may lack precision for the trajectory within $[0, T - \Delta t]$. The transfer of teacher information for the interval $[0, T - \Delta t]$ may occur in another iteration with a random time $t \leq T - \Delta t$. In this case, the student model $G_{\boldsymbol{\theta}}(\mathbf{x}_t, t, 0)$ distills the teacher information solely within the interval $[t - \Delta t, t]$, where the network is trained with $\mathbf{x}_t$ as the input.

However, for 1-step generation, $g_{\boldsymbol{\theta}}(\mathbf{x}_T, T, 0)$ still lacks perfect knowledge of the teacher information within $[t - \Delta t, t]$. This is because, when the student network input is $\mathbf{x}_T$, the teacher information for the interval $[t - \Delta t, t]$ with $t \leq T - \Delta t$ has not been explicitly provided, as the student was trained with the input $\mathbf{x}_t$ to distill information within $[t - \Delta t, t]$. Given the non-overlapping intervals with distilled information from local consistency, the student neural network must extrapolate and attempt to connect the scattered teacher information. Consequently, this implicit signal provided by teacher results in slow convergence and inferior performance.

**Soft Consistency (Explicit Information from Teacher)** At the opposite end of local consistency, there is glocal consistency, where the teacher prediction is constructed solely with an ODE solver to cover the entire interval $[0, T]$ (or $[0, t]$ for a random time $t$). In this case, the student model can explicitly extract information from the teacher. However, this approach is resource-intensive (3x slower than local consistency on CIFAR-10) during training due to the ODE solving calls on the entire interval at each iteration.

In contrast, our innovative loss, soft matching, constructs the teacher prediction by using an ODE solver spanning from $T$ to a random $u$. Importantly, $u$ is not limited to $T - \Delta t$, but can take any value in the range of $[0, T]$. As a result, the teacher information has the opportunity to be distilled and transmitted over a broader range of $[u, T]$. More precisely, if a random $u$ is sampled with $u \leq t - \Delta t$, the range $[u, T]$ contains the interval $[t - \Delta t, t]$, and the student network directly distills teacher information of the interval $[t - \Delta t, t]$. As $u$ is arbitrary, the student of CTM with input $\mathbf{x}_T$ will ultimately receive the explicit information from the teacher for any intermediate timesteps. This renders CTM superior to CM for 1-step generation, as evidenced in Figure 10, while maintaining training efficiency (2x faster than global consistency on CIFAR-10).

## C.4   COMPARISON OF GAN EFFECTS IN GENERATION

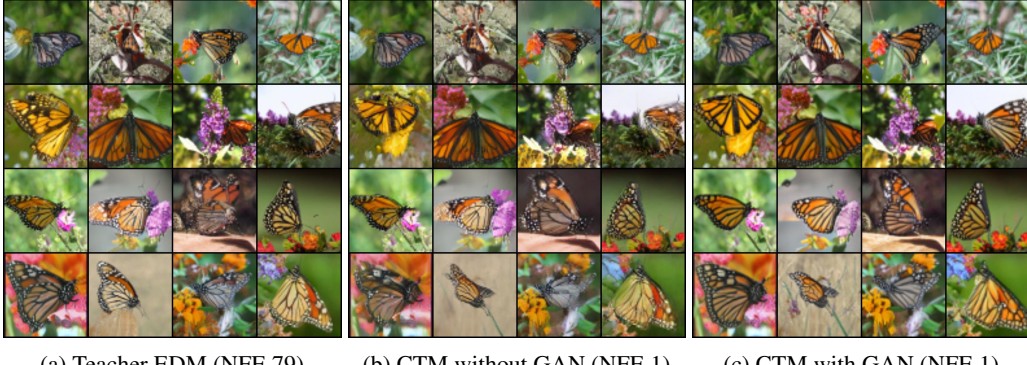

(a) Teacher EDM (NFE 79)    (b) CTM without GAN (NFE 1)    (c) CTM with GAN (NFE 1)

Figure 14: Uncurated samples from (a) teacher, (b) CTM without GAN, and (c) CTM with GAN. For visualization purpose, we upsample $64 \times 64$ samples to $224 \times 224$ resolution with bilinear upsampling technique. Best viewed with zoom-in.

This section investigates the effect of adversarial training with generated samples and its statistics. In Figure 14, we compare the samples of (a) the teacher diffusion model, (b) CTM (NFE 1) without GAN, and (c) CTM (NFE 1) with GAN. It shows that the samples with auxiliary GAN loss exhibit enhanced fine details, effectively addressing high-frequency alisasing artifacts. Moreover, improvements in overall shapes (butterfly/background) and features (brightness/contrast/saturation) are evident in these samples. Although existing literature (Kynkäänniemi et al., 2023) discusses the possibility that FID improvement may not necessarily correlate with an actual enhancement in human perceptual judgement, our observations in Figure 14 indicates that, in the case of CTM, the improvement achieved through GAN is indeed perceptually discernible in human judgement.

Table 3: Effect of GAN Loss on CIFAR-10. We use identical hyperparameters except the GAN loss for fair comparison.

| Model | NFE | FID |
|---|---|---|
| CTM w/o GAN | 1 | 5.19 |
| CTM w/ GAN | | 2.28 |
| CTM w/o GAN | 18 | 3.00 |
| CTM w/ GAN | | 2.23 |

We compare FIDs of CTM trained with/without GAN in Table 3. Consistent to the findings in previous research (Song & Ermon, 2020), the improved fine-details of GAN-augmented samples in Figure 14-(c) results in better FID than CTM without GAN, as indicated in the table. Moreover, Table 3 demonstrates that the use of adversarial loss is also beneficial on the generation of large-NFE samples.

## C.5    TRAJECTORY CONTROL WITH GUIDANCE

We could apply $\gamma$-sampling for application tasks, such as image inpainting or colorization, using the (straightforwardly) generalized algorithm suggested in CM. In this section, however, we propose a loss-based trajectory optimization algorithm in Algorithm 3 for potential application downstream tasks.

---

**Algorithm 3** Loss-based Trajectory Optimization

1: $\mathbf{x}_{ref}$ is given
2: Diffuse $\mathbf{x}_{t_0} \leftarrow \mathbf{x}_{ref} + t_0\boldsymbol{\epsilon}$
3: **for** $n = 1$ to $N$ **do**
4:     $\tilde{t}_n \leftarrow \sqrt{1 - \gamma^2}t_n$
5:     Denoise $\mathbf{x}_{\tilde{t}_n} \leftarrow G_{\boldsymbol{\theta}}(\mathbf{x}_{t_{n-1}}, t_{n-1}, \tilde{t}_n)$
6:     **for** $m = 1$ to $M$ **do**
7:         Sample $\boldsymbol{\epsilon}, \boldsymbol{\epsilon}' \sim \mathcal{N}(0, \mathbf{I})$
8:         Apply corrector $\mathbf{x}_{\tilde{t}_n} \leftarrow \mathbf{x}_{\tilde{t}_n} + \frac{\zeta}{2}\left(\nabla \log p_{\tilde{t}_n}(\mathbf{x}_{\tilde{t}_n}) - c_{\tilde{t}_n}\nabla_{\mathbf{x}_{\tilde{t}_n}}L(\mathbf{x}_{\tilde{t}_n}, \mathbf{x}_{ref} + \tilde{t}_n\boldsymbol{\epsilon})\right) + \sqrt{\zeta}\boldsymbol{\epsilon}'$
9:     **end for**
10:    Sample $\boldsymbol{\epsilon} \sim \mathcal{N}(0, \mathbf{I})$
11:    Diffuse $\mathbf{x}_{t_n} \leftarrow \mathbf{x}_{\hat{t}_n} + \gamma t_n\boldsymbol{\epsilon}$
12: **end for**

---

Algorithm 3 uses the time traversal from $t_{n-1}$ to $\tilde{t}_n$, and apply the loss-embedded corrector (Song et al., 2020b) algorithm to explore $\tilde{t}_n$-manifold. For instance, the loss could be a feature loss between $\mathbf{x}_{\tilde{t}_n}$ and $\mathbf{x}_{ref} + \tilde{t}_n\boldsymbol{\epsilon}$. With this corrector-based guidance, we could control the sample variance. This loss-embedded corrector could also be interpreted as sampling from a posterior distribution. For Figure 6, we choose $N = 2$ with $(t_0, t_1) = \left((\sigma_{\max}^{1/\rho} + (\sigma_{\min}^{1/\rho} - \sigma_{\max}^{1/\rho})0.45)^\rho, (\sigma_{\max}^{1/\rho} + (\sigma_{\min}^{1/\rho} - \sigma_{\max}^{1/\rho})0.35)^\rho\right)$, $c_{\tilde{t}_n} \equiv 1$, and $M = 10$.

## D   IMPLEMENTATION DETAILS

### D.1   TRAINING DETAILS

Following Karras et al. (2022), we utilize the EDM's skip connection $c_{\text{skip}}(t) = \frac{\sigma_{\text{data}}^2}{t^2 + \sigma_{\text{data}}^2}$ and output scale $c_{\text{out}}(t) = \frac{t\sigma_{\text{data}}}{\sqrt{t^2 + \sigma_{\text{data}}^2}}$ for $g_{\boldsymbol{\theta}}$ modeling as

$$g_{\boldsymbol{\theta}}(\mathbf{x}_t, t, s) = c_{\text{skip}}(t)\mathbf{x}_t + c_{\text{out}}(t)\text{NN}_{\boldsymbol{\theta}}(\mathbf{x}_t, t, s),$$

where $\text{NN}_{\boldsymbol{\theta}}$ refers to the actual neural network output. The advantage of this EDM-style skip and output scalings are that if we copy the teacher model's parameters to the student model's param-

---

**Algorithm 4** CTM Training

1: **repeat**
2:     Sample $\mathbf{x}_0$ from data distribution
3:     Sample $\boldsymbol{\epsilon} \sim \mathcal{N}(0, I)$
4:     Sample $t \in [0, T], s \in [0, t], u \in [s, t]$
5:     Calculate $\mathbf{x}_t = \mathbf{x}_0 + t\boldsymbol{\epsilon}$
6:     Calculate $\texttt{Solver}(\mathbf{x}_t, t, u; \boldsymbol{\phi})$
7:     Update $\boldsymbol{\theta} \leftarrow \boldsymbol{\theta} - \frac{\partial}{\partial\boldsymbol{\theta}}\mathcal{L}(\boldsymbol{\theta}, \boldsymbol{\eta})$
8:     Update $\boldsymbol{\eta} \leftarrow \boldsymbol{\eta} + \frac{\partial}{\partial\boldsymbol{\eta}}\mathcal{L}_{\text{GAN}}(\boldsymbol{\theta}, \boldsymbol{\eta})$
9: **until** converged

---

eters, except student model's $s$-embedding structure, $g_{\boldsymbol{\theta}}(\mathbf{x}_t, t, t)$ initialized with $\boldsymbol{\phi}$ would be close to the teacher denoiser $D_{\boldsymbol{\phi}}(\mathbf{x}_t, t)$. This good initialization partially explains the fast convergence speed.

We use 4×V100 (16G) GPUs for CIFAR-10 experiments and 8×A100 (40G) GPUs for ImageNet experiments. We use the warm-up for $\lambda_{\text{GAN}}$ hyperparameter. On CIFAR-10, we deactivate GAN training with $\lambda_{\text{GAN}} = 0$ until $50K$ training iterations ($200K$ for training without pre-trained DM) and activate the generator training with the adversarial loss (added to CTM and DSM losses) by setting $\lambda_{\text{GAN}}$ to be the adaptive weight. The minibatch per GPU is 16 in the CTM+DSM training phase, and 11 in the CTM+DSM+GAN training phase. On ImageNet, due to the excessive training budget, we deactivate GAN only for 10k iterations and activate GAN training afterwards. We fix the minibatch to be 11 throughout the CTM+DSM or the CTM+DSM+GAN training in ImageNet.

We follow the training configuration mainly from CM, but for the discriminator training, we follow that of StyleGAN-XL (Sauer et al., 2022). For $\mathcal{L}_{\text{CTM}}$ calculation, we use LPIPS (Zhang et al., 2018) as a feature extractor. We choose $t$ and $s$ from the $N$-discretized timesteps to calculate $\mathcal{L}_{\text{CTM}}$, following CM. Across the training, we choose the maximum number of ODE steps to prevent a single iteration takes too long time. For CIFAR-10, we choose $N = 18$ and the maximum number of ODE steps to be 17, i.e., we do nothing for CIFAR-10 training. For ImageNet, we choose $N = 40$ and the maximum number of ODE steps to be 20. We find the tendency that the training performance is improved by the number of ODE steps, so one could possibly improve our ImageNet result by choosing larger maximum ODE steps.

Table 4: Experimental details on hyperparameters.

| Hyperparameter | CIFAR-10 | | | ImageNet 64x64 |
|---|---|---|---|---|
| | Unconditional | | Conditional | Conditional |
| | Training with $\phi$ | Training from Scratch | Training with $\phi$ | Training with $\phi$ |
| Learning rate | 0.0004 | 0.0004 | 0.0004 | 0.000008 |
| Discriminator learning rate | 0.002 | 0.002 | 0.002 | 0.002 |
| Student's stop-grad EMA parameter $\mu$ | 0.9999 | 0.999 | 0.999 | 0.999 |
| $N$ | 18 | 18 | 18 | 40 |
| ODE solver | Heun | Heun | Heun | Heun |
| Teacher | $D_\phi(\mathbf{x}_t, t)$ | $g_\theta(\mathbf{x}_t, t, t)$ | $D_\phi(\mathbf{x}_t, t)$ | $D_\phi(\mathbf{x}_t, t)$ |
| Max. ODE steps | 17 | 17 | 17 | 20 |
| EMA decay rate | 0.999 | 0.999 | 0.999 | 0.999 |
| Training iterations | 100K | 300K | 100K | 30K |
| Mixed-Precision (FP16) | True | True | True | True |
| Batch size | 256 | 128 | 512 | 2048 |
| Number of GPUs | 4 | 4 | 4 | 8 |

For $\mathcal{L}_{\text{DSM}}$ calculation, we select $50\%$ of time sampling from EDM's original scheme of $t \sim \mathcal{N}(-1.2, 1.2^2)$. For the other half time, we first draw sample from $\xi \sim [0, 0.7]$ and transform it using $(\sigma_{\max}^{1/\rho} + \xi(\sigma_{\min}^{1/\rho} - \sigma_{\max}^{1/\rho}))^\rho$. This specific time sampling blocks the neural network to forget the denoiser information for large time. For $\mathcal{L}_{\text{GAN}}$ calculation, we use two feature extractors to transform GAN input to the feature space: the EfficientNet (Tan & Le, 2019) and DeiT-base (Touvron et al., 2021). Before obtaining an input's feature, we upscale the image to 224x224 resolution with bilinear interpolation. After transforming to the feature space, we apply the cross-channel mixing and cross-scale mixing to represent the input with abundant and non-overlapping features. The output of the cross-scale mixing is a feature pyramid consisting of four feature maps at different resolutions (Sauer et al., 2022). In total, we use eight discriminators (four for EfficientNet features and the other four for DeiT-base features) for GAN training.

Following CM, we apply Exponential Moving Average (EMA) to update $\text{sg}(\boldsymbol{\theta})$ by

$$\text{sg}(\boldsymbol{\theta}) \leftarrow \text{stopgrad}(\mu\text{sg}(\boldsymbol{\theta}) + (1 - \mu)\boldsymbol{\theta}).$$

However, unlike CM, we find that our model bestly works with $\mu = 0.999$ or $\mu = 0.9999$, which largely remedy the subtle instability arise from GAN training. Except for the unconditional CIFAR-10 training with $\phi$, we set $\mu$ to be 0.999 as default. Throughout the experiments, we use $\sigma_{\min} = 0.002$, $\sigma_{\max} = 80$, $\rho = 7$, and $\sigma_{\text{data}} = 0.5$.

## D.2 EVALUATION DETAILS

For likelihood evaluation, we solve the PF ODE, following the practice suggested in Kim et al. (2022b) with the RK45 (Dormand & Prince, 1980) ODE solver of $\text{tol} = 1e - 3$ and $t_{\min} = 0.002$.

Throughout the paper, we choose $\gamma = 0$ otherwise stated. In particular, for Tables 1 and 2, we report the sample quality metrics based on either the one-step sampling of CM or the $\gamma = 0$ sampling for NFE 2 case. For CIFAR-10, we calculate the FID score based on Karras et al. (2022) statistics, and Figure 15 summarizes the result. For ImageNet, we compute the metrics following Dhariwal &

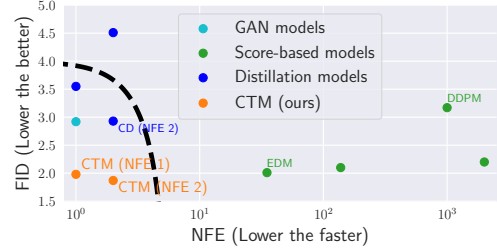

Figure 15: SOTA on CIFAR-10. Closeness to the origin indicates better performance.

Nichol (2021) and their pre-calculated statistics. For the StyleGAN-XL ImageNet result, we recalculated the metrics based on the statistics released by Dhariwal & Nichol (2021), using StyleGAN-XL's official checkpoint.

For large-NFE sampling, we follow the EDM's time discretization. Namely, if we draw $n$-NFE samples, we equi-divide $[0, 1]$ with $n$ points and transform it (say $\xi$) to the time scale by $(\sigma_{\max}^{1/\rho} + (\sigma_{\min}^{1/\rho} - \sigma_{\max}^{1/\rho})\xi)^\rho$. However, we emphasize the time discretization for both training and sampling is a modeler's choice.

## E ADDITIONAL GENERATED SAMPLES

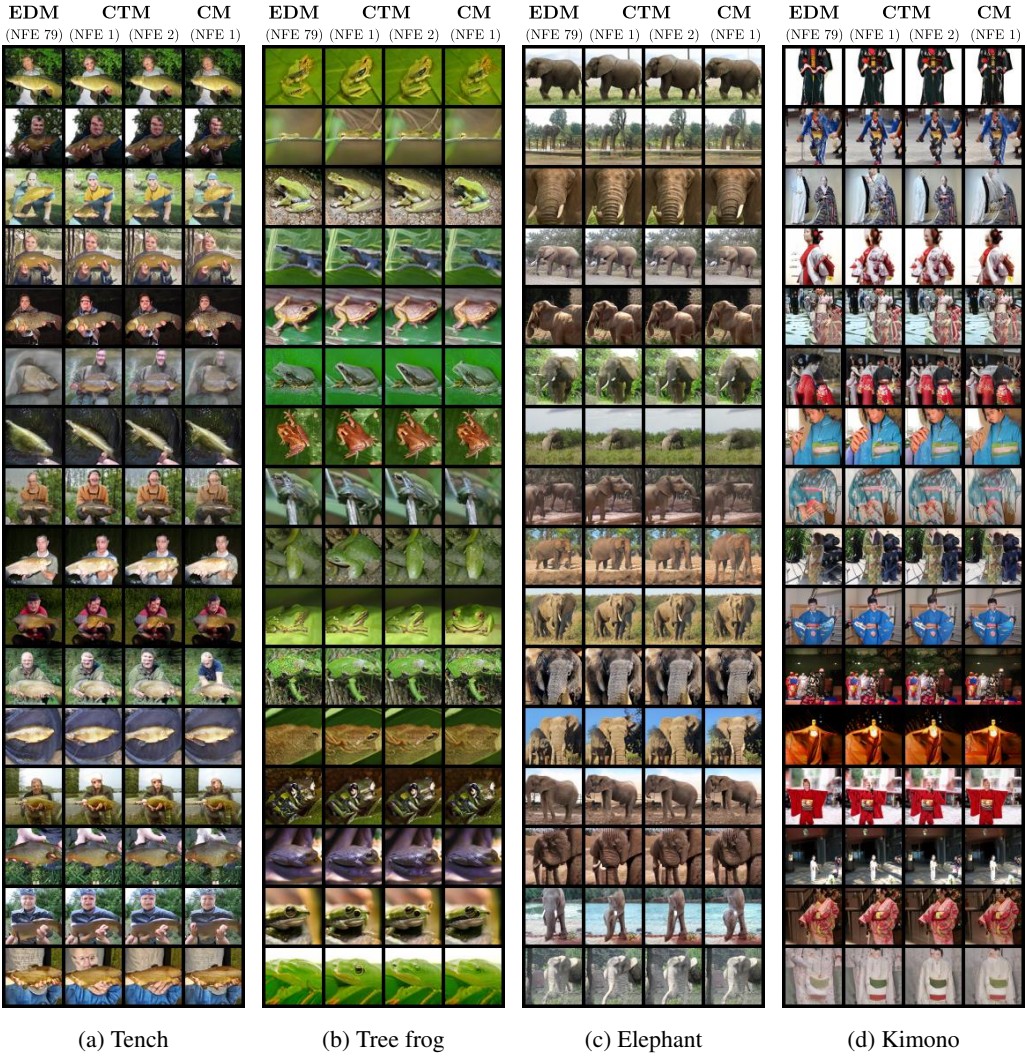

|   (a) Tench   |   (b) Tree frog   |   (c) Elephant   |   (d) Kimono   |

Figure 16: Uncurated sample comparisons with identical starting points, generated by EDM (FID 2.44) with NFE 79, CTM (FID 2.19) with NFE 1, CTM (FID 1.90) with NFE 2, and CM (FID 6.20) with NFE 1, on (a) tench (class id: 0), (b) tree frog (class id: 31), (c) elephant (class id: 386), and (d) kimono (class id: 614).

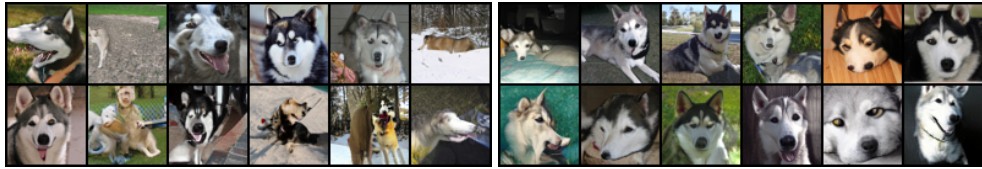

(a) W/o classifier-rejection sampling (NFE 1)     (b) W/ classifier-rejection sampling (avg. NFE 2)

Figure 17: Random samples (Siberian Husky) (d) with and (e) without classifier-free sampling.

# F  THEORETICAL SUPPORTS AND PROOFS

## F.1  PROOF OF LEMMA 2

***Proof of Lemma 2.***  The reverse-time SDE is

$$d\mathbf{x}_{T-\tau} = g^2(T-\tau)\nabla \log p_{T-\tau}(\mathbf{x}_{T-\tau})dt + g(T-\tau)d\mathbf{w}_\tau,$$

where the forward-time SDE is given by

$$d\mathbf{x}_\tau = g(\tau)d\mathbf{w}_\tau.$$

The reverse-time PF-ODE thus becomes

$$d\mathbf{x}_{T-\tau} = \frac{1}{2}g^2(T-\tau)\nabla \log p_{T-\tau}(\mathbf{x}_{T-\tau})d\tau.$$

Therefore, by integrating from $T-t$ to $T-s$ ($s < t$), we obtain

$$\mathbf{x}_s - \mathbf{x}_t = \int_{T-t}^{T-s} \frac{1}{2}g^2(T-\tau)\nabla \log p_{T-\tau}(\mathbf{x}_{T-\tau})d\tau.$$

By change-of-variable with $u = T - \tau$, the equation is derived to be

$$\mathbf{x}_s = \mathbf{x}_t - \int_t^s \frac{1}{2}g^2(u)\nabla \log p_u(\mathbf{x}_u)du.$$

With $g(u) = \sqrt{2u}$ and the Tweedie's formula $\nabla \log p_u(\mathbf{x}_u) = \frac{\mathbb{E}[\mathbf{x}|\mathbf{x}_u]-\mathbf{x}_u}{u^2}$, we derive Eq. (2) in our paper:

$$\mathbf{x}_s = G(\mathbf{x}_t, t, s) = \mathbf{x}_t + \int_t^s \frac{\mathbf{x}_u - \mathbb{E}[\mathbf{x}|\mathbf{x}_u]}{u}du.$$

Now, we derive the following equations:

$$\begin{aligned}
G(\mathbf{x}_t, t, s) &= \frac{s}{t}\mathbf{x}_t + (1-\frac{s}{t})\mathbf{x}_t + \int_t^s \frac{\mathbf{x}_u - \mathbb{E}[\mathbf{x}|\mathbf{x}_u]}{u}du \\
&= \frac{s}{t}\mathbf{x}_t + (1-\frac{s}{t})[\mathbf{x}_t + \frac{1}{(1-\frac{s}{t})}\int_t^s \frac{\mathbf{x}_u - \mathbb{E}[\mathbf{x}|\mathbf{x}_u]}{u}du] \\
&= \frac{s}{t}\mathbf{x}_t + (1-\frac{s}{t})[\mathbf{x}_t + \frac{t}{t-s}\int_t^s \frac{\mathbf{x}_u - \mathbb{E}[\mathbf{x}|\mathbf{x}_u]}{u}du] \\
&= \frac{s}{t}\mathbf{x}_t + (1-\frac{s}{t})g(\mathbf{x}_t, t, s),
\end{aligned}$$

where $g(\mathbf{x}_t, t, s) := \mathbf{x}_t + \frac{t}{t-s}\int_t^s \frac{\mathbf{x}_u - \mathbb{E}[\mathbf{x}|\mathbf{x}_u]}{u}du$.

As the score, $\nabla \log p_t(\mathbf{x})$, is integrable, the Fundamental Theorem of Calculus applies, leading to

$$\begin{aligned}
\lim_{s\to t} g(\mathbf{x}_t, t, s) &= \mathbf{x}_t + t\lim_{s\to t}\frac{1}{t-s}\int_t^s \frac{\mathbf{x}_u - \mathbb{E}[\mathbf{x}_0|\mathbf{x}_u]}{u}du \\
&= \mathbf{x}_t - t\frac{\mathbf{x}_t - \mathbb{E}[\mathbf{x}_0|\mathbf{x}_t]}{t} \\
&= \mathbb{E}[\mathbf{x}_0|\mathbf{x}_t].
\end{aligned}$$

∎

## F.2  PROOF OF THEOREM 1

***Proof of Theorem 1.***  Define $\mathcal{T}_{t\to s}$ as the oracle transition mapping from $t$ to $s$ via the diffusion process Eq. (6). Let $\mathcal{T}_{t\to s}^{\boldsymbol{\theta}^*}(\cdot)$ represent the transition mapping from the optimal CTM, and $\mathcal{T}_{t\to s}^{\boldsymbol{\phi}}(\cdot)$ represent the transition mapping from the empirical probability flow ODE. Since all processes start at point $T$ with initial probability distribution $p_T$ and $\mathcal{T}_{t\to s}^{\boldsymbol{\theta}^*}(\cdot) = \mathcal{T}_{t\to s}^{\boldsymbol{\phi}}(\cdot)$, Theorem 2 in (Chen et al., 2022) and $\mathcal{T}_{T\to t}\sharp p_T = p_t$ from Proposition 9 tell us that for $t > s$

$$D_{TV}\left(\mathcal{T}_{t\to s}\sharp p_t, \mathcal{T}_{t\to s}^{\boldsymbol{\theta}^*}\sharp p_t\right) = D_{TV}\left(\mathcal{T}_{t\to s}\sharp p_t, \mathcal{T}_{t\to s}^{\boldsymbol{\phi}}\sharp p_t\right) = \mathcal{O}(t-s). \tag{9}$$

$$D_{TV}\left(\mathcal{T}_{t\to 0}\mathcal{T}_{\sqrt{1-\gamma^2}t\to t}\mathcal{T}_{T\to\sqrt{1-\gamma^2}t}\sharp p_T, \mathcal{T}_{t\to 0}^{\boldsymbol{\theta}^*}\mathcal{T}_{\sqrt{1-\gamma^2}t\to t}\mathcal{T}_{T\to\sqrt{1-\gamma^2}t}^{\boldsymbol{\theta}^*}\sharp p_T\right)$$

$$\overset{(a)}{\le} D_{TV}\left(\mathcal{T}_{t\to 0}\mathcal{T}_{\sqrt{1-\gamma^2}t\to t}\mathcal{T}_{T\to\sqrt{1-\gamma^2}t}\sharp p_T, \mathcal{T}_{t\to 0}^{\boldsymbol{\theta}^*}\mathcal{T}_{\sqrt{1-\gamma^2}t\to t}\mathcal{T}_{T\to\sqrt{1-\gamma^2}t}\sharp p_T\right)$$

$$+ D_{TV}\left(\mathcal{T}_{t\to 0}^{\boldsymbol{\theta}^*}\mathcal{T}_{\sqrt{1-\gamma^2}t\to t}\mathcal{T}_{T\to\sqrt{1-\gamma^2}t}\sharp p_T, \mathcal{T}_{t\to 0}^{\boldsymbol{\theta}^*}\mathcal{T}_{\sqrt{1-\gamma^2}t\to t}\mathcal{T}_{T\to\sqrt{1-\gamma^2}t}^{\boldsymbol{\theta}^*}\sharp p_T\right)$$

$$\overset{(b)}{=} D_{TV}\left(\mathcal{T}_{t\to 0}\mathcal{T}_{T\to t}\sharp p_T, \mathcal{T}_{t\to 0}^{\boldsymbol{\theta}^*}\mathcal{T}_{T\to t}\sharp p_T\right) + D_{TV}\left(\mathcal{T}_{T\to\sqrt{1-\gamma^2}t}\sharp p_T, \mathcal{T}_{T\to\sqrt{1-\gamma^2}t}^{\boldsymbol{\theta}^*}\sharp p_T\right)$$

$$\overset{(c)}{=} D_{TV}\left(\mathcal{T}_{t\to 0}\sharp p_t, \mathcal{T}_{t\to 0}^{\boldsymbol{\theta}^*}\sharp p_t\right) + D_{TV}\left(\mathcal{T}_{T\to\sqrt{1-\gamma^2}t}\sharp p_T, \mathcal{T}_{T\to\sqrt{1-\gamma^2}t}^{\boldsymbol{\theta}^*}\sharp p_T\right)$$

$$\overset{(d)}{=} \mathcal{O}(\sqrt{t}) + \mathcal{O}(\sqrt{T-\sqrt{1-\gamma^2}t}).$$

Here (a) is obtained from the triangular inequality, (b) and (c) are due to $\mathcal{T}_{\sqrt{1-\gamma^2}t\to t}\mathcal{T}_{T\to\sqrt{1-\gamma^2}t} = \mathcal{T}_{T\to t}$ and $\mathcal{T}_{T\to t}\sharp p_T = p_t$ from Proposition 9, and (d) comes from Eq. (9).

∎

### F.3 PROOF OF PROPOSITION 3

***Proof of Proposition 3.*** Consider a LPIPS-like metric, denoted as $d(\cdot,\cdot)$, determined by a feature extractor $\mathcal{F}$ of $p_{\text{data}}$. That is, $d(\mathbf{x},\mathbf{y}) = \|\mathcal{F}(\mathbf{x}) - \mathcal{F}(\mathbf{y})\|_q$ for $q \ge 1$. For simplicity of notation, we denote $\boldsymbol{\theta}_N$ as $\boldsymbol{\theta}$. Since $\mathcal{L}_{\text{CTM}}^N(\boldsymbol{\theta};\boldsymbol{\phi}) = 0$, it implies that for any $\mathbf{x}_{t_n}$, $n \in [\![1,N]\!]$, and $m \in [\![1,n]\!]$

$$\mathcal{F}\big(G_{\boldsymbol{\theta}}(G_{\boldsymbol{\theta}}(\mathbf{x}_{t_{n+1}}, t_{n+1}, t_m), t_m, 0)\big) = \mathcal{F}\big(G_{\boldsymbol{\theta}}(G_{\boldsymbol{\theta}}(\mathbf{x}_{t_n}^{\boldsymbol{\phi}}, t_n, t_m), t_m, 0)\big) \tag{10}$$

Denote

$$\mathbf{e}_{n,m} := \mathcal{F}\big(G_{\boldsymbol{\theta}}(G_{\boldsymbol{\theta}}(\mathbf{x}_{t_n}, t_n, t_m), t_m, 0)\big) - \mathcal{F}\big(G_{\boldsymbol{\theta}}(G(\mathbf{x}_{t_n}, t_n, t_m; \boldsymbol{\phi}), t_m, 0)\big).$$

Then due to Eq. (10) and $G$ is an ODE-trajectory function that $G(\mathbf{x}_{t_{n+1}}, t_{n+1}, t_m; \boldsymbol{\phi}) = G(\mathbf{x}_{t_n}, t_n, t_m; \boldsymbol{\phi})$, we have

$$\mathbf{e}_{n+1,m} = \mathcal{F}\big(G_{\boldsymbol{\theta}}(G_{\boldsymbol{\theta}}(\mathbf{x}_{t_{n+1}}, t_{n+1}, t_m), t_m, 0)\big) - \mathcal{F}\big(G_{\boldsymbol{\theta}}(G(\mathbf{x}_{t_{n+1}}, t_{n+1}, t_m; \boldsymbol{\phi}), t_m, 0)\big)$$

$$= \mathcal{F}\big(G_{\boldsymbol{\theta}}(G_{\boldsymbol{\theta}}(\mathbf{x}_{t_n}^{\boldsymbol{\phi}}, t_n, t_m), t_m, 0)\big) - \mathcal{F}\big(G_{\boldsymbol{\theta}}(G(\mathbf{x}_{t_n}, t_n, t_m; \boldsymbol{\phi}), t_m, 0)\big)$$

$$= \mathcal{F}\big(G_{\boldsymbol{\theta}}(G_{\boldsymbol{\theta}}(\mathbf{x}_{t_n}^{\boldsymbol{\phi}}, t_n, t_m), t_m, 0)\big) - \mathcal{F}\big(G_{\boldsymbol{\theta}}(G_{\boldsymbol{\theta}}(\mathbf{x}_{t_n}, t_n, t_m), t_m, 0)\big)$$

$$+ \mathcal{F}\big(G_{\boldsymbol{\theta}}(G_{\boldsymbol{\theta}}(\mathbf{x}_{t_n}, t_n, t_m), t_m, 0)\big) - \mathcal{F}\big(G_{\boldsymbol{\theta}}(G(\mathbf{x}_{t_n}, t_n, t_m; \boldsymbol{\phi}), t_m, 0)\big)$$

$$= \mathcal{F}\big(G_{\boldsymbol{\theta}}(G_{\boldsymbol{\theta}}(\mathbf{x}_{t_n}^{\boldsymbol{\phi}}, t_n, t_m), t_m, 0)\big) - \mathcal{F}\big(G_{\boldsymbol{\theta}}(G_{\boldsymbol{\theta}}(\mathbf{x}_{t_n}, t_n, t_m), t_m, 0)\big) + \mathbf{e}_{n,m}.$$

Therefore,

$$\|\mathbf{e}_{n+1,m}\|_q \le \left\|\mathcal{F}\big(G_{\boldsymbol{\theta}}(G_{\boldsymbol{\theta}}(\mathbf{x}_{t_n}^{\boldsymbol{\phi}}, t_n, t_m), t_m, 0)\big) - \mathcal{F}\big(G_{\boldsymbol{\theta}}(G_{\boldsymbol{\theta}}(\mathbf{x}_{t_n}, t_n, t_m), t_m, 0)\big)\right\|_q + \|\mathbf{e}_{n,m}\|_q$$

$$\le L_1 L_2^2 \left\|\mathbf{x}_{t_n}^{\boldsymbol{\phi}} - \mathbf{x}_{t_n}\right\|_q + \|\mathbf{e}_{n,m}\|_q$$

$$= \mathcal{O}((t_{n+1} - t_n)^{p+1}) + \|\mathbf{e}_{n,m}\|_q.$$

Notice that since $G_{\boldsymbol{\theta}}(\mathbf{x}_{t_m}, t_m, t_m) = \mathbf{x}_{t_m} = G(\mathbf{x}_{t_m}, t_m, t_m; \boldsymbol{\phi})$, $\mathbf{e}_{m,m} = \mathbf{0}$.

So we can obtain via induction that

$$\|\mathbf{e}_{n+1,m}\|_q \le \|\mathbf{e}_{m,m}\|_q + \sum_{k=m}^{n-1} \mathcal{O}((t_{k+1} - t_k)^{p+1})$$

$$= \sum_{k=m}^{n-1} \mathcal{O}((t_{k+1} - t_k)^{p+1})$$
$$\leq \mathcal{O}((\Delta_N t)^p)(t_n - t_m).$$

∎

Indeed, an analogue of Proposition 3 holds for time-conditional feature extractors.

Let $d_t(\cdot, \cdot)$ be a LPIPS-like metric determined by a time-conditional feature extractor $\mathcal{F}_t$. That is, $d_t(\mathbf{x}, \mathbf{y}) = \|\mathcal{F}_t(\mathbf{x}) - \mathcal{F}_t(\mathbf{y})\|_q$ for $q \geq 1$. We can similarly derive

$$\sup_{\mathbf{x} \in \mathbb{R}^D} d_{t_m}\big(G_{\boldsymbol{\theta}}(\mathbf{x}, t_n, t_m), G(\mathbf{x}, t_n, t_m; \boldsymbol{\phi})\big) = \mathcal{O}((\Delta_N t)^p)(t_n - t_m).$$

### F.4 PROOF OF PROPOSITION 5

***Proof of Proposition 5.*** We first prove that for any $t \in [0, T]$ and $s \leq t$, as $N \to \infty$,

$$\sup_{\mathbf{x} \in \mathbb{R}^D} \|G_{\boldsymbol{\theta}_N}(G_{\boldsymbol{\theta}_N}(\mathbf{x}, t, s), s, 0), G_{\boldsymbol{\theta}_N}(G(\mathbf{x}, t, s; \boldsymbol{\phi}), s, 0)\|_2 \to 0. \tag{11}$$

We may assume $\{t_n\}_{n=1}^N$ so that $t_m = s$, $t_n = t$, and $t_{m+1} \to s$, $t_{n+1} \to t$ as $\Delta_N t \to \infty$.

$$\sup_{\mathbf{x}} \|G_{\boldsymbol{\theta}_N}(G_{\boldsymbol{\theta}_N}(\mathbf{x}, t, s), s, 0), G_{\boldsymbol{\theta}_N}(G(\mathbf{x}, t, s; \boldsymbol{\phi}), s, 0)\|_2$$
$$\leq \sup_{\mathbf{x}} \|G_{\boldsymbol{\theta}_N}(G_{\boldsymbol{\theta}_N}(\mathbf{x}, t, s), s, 0), G_{\boldsymbol{\theta}_N}(G_{\boldsymbol{\theta}_N}(\mathbf{x}, t_{n+1}, t_{m+1}; \boldsymbol{\phi}), t_{m+1}, 0)\|_2$$
$$+ \sup_{\mathbf{x}} \|G_{\boldsymbol{\theta}_N}(G_{\boldsymbol{\theta}_N}(\mathbf{x}, t_{n+1}, t_{m+1}; \boldsymbol{\phi}), t_{m+1}, 0), G_{\boldsymbol{\theta}_N}(G(\mathbf{x}, t_{n+1}, t_{m+1}; \boldsymbol{\phi}), t_{m+1}, 0)\|_2$$
$$+ \sup_{\mathbf{x}} \|G_{\boldsymbol{\theta}_N}(G(\mathbf{x}, t_{n+1}, t_{m+1}; \boldsymbol{\phi}), t_{m+1}, 0), G_{\boldsymbol{\theta}_N}(G(\mathbf{x}, t, s; \boldsymbol{\phi}), s, 0)\|_2$$

Since both $G$ and $G_{\boldsymbol{\theta}_N}$ are uniform continuous on $\mathbb{R}^D \times [0, T] \times [0, T]$, together with Proposition 3, we obtain Eq. (11) as $\Delta_N t \to \infty$.

In particular, Eq. (11) implies that when $N \to \infty$

$$\sup_{\mathbf{x}} \|G_{\boldsymbol{\theta}_N}(G_{\boldsymbol{\theta}_N}(\mathbf{x}, T, 0), 0, 0) - G_{\boldsymbol{\theta}_N}(G(\mathbf{x}, T, 0; \boldsymbol{\phi}), 0, 0)\|_2$$
$$= \sup_{\mathbf{x}} \|G_{\boldsymbol{\theta}_N}(\mathbf{x}, T, 0) - G(\mathbf{x}, T, 0; \boldsymbol{\phi})\|_2 \to 0.$$

This implies that $p_{\boldsymbol{\theta}_N}(\cdot)$, the pushforward distribution of $p_T$ induced by $G_{\boldsymbol{\theta}_N}(\cdot, T, 0)$, converges in distribution to $p_{\boldsymbol{\phi}}(\cdot)$. Note that since $\{G_{\boldsymbol{\theta}_N}\}_N$ is uniform Lipschitz

$$\|G_{\boldsymbol{\theta}}(\mathbf{x}, t, s) - G_{\boldsymbol{\theta}}(\mathbf{x}', t, s)\|_2 \leq L \|\mathbf{x} - \mathbf{x}'\|_2, \quad \text{for all } \mathbf{x}, \mathbf{x}' \in \mathbb{R}^D, t, s \in [0, T], \text{ and } \boldsymbol{\theta},$$

$\{G_{\boldsymbol{\theta}_N}\}_N$ is asymptotically uniformly equicontinuous. Moreover, $\{G_{\boldsymbol{\theta}_N}\}_N$ is uniform bounded in $\boldsymbol{\theta}_N$. Therefore, the converse of Scheffé's theorem (Boos, 1985; Sweeting, 1986) implies that $\|p_{\boldsymbol{\theta}_N}(\cdot) - p_{\boldsymbol{\phi}}(\cdot)\|_\infty \to 0$ as $N \to \infty$. Similar argument can be adapted to prove $\|p_{\boldsymbol{\theta}_N}(\cdot) - p_{\text{data}}(\cdot)\|_\infty \to 0$ as $N \to \infty$ if the regression target $p_{\boldsymbol{\phi}}(\cdot)$ is replaced with $p_{\text{data}}(\cdot)$. ∎

### F.5 PROOF OF PROPOSITION 6

**Lemma 10.** *Let $f: \mathbb{R}^D \times [0, T] \to \mathbb{R}^D$ be a function which satisfies the following conditions:*

(a) *$f(\cdot, t)$ is Lipschitz for any $t \in [0, T]$: there is a function $L(t) \geq 0$ so that for any $t \in [0, T]$ and $\mathbf{x}, \mathbf{y} \in \mathbb{R}^D$*

$$\|f(\mathbf{x}, t) - f(\mathbf{y}, t)\| \leq L(t) \|\mathbf{x} - \mathbf{y}\|,$$

(b) *Linear growth in $\mathbf{x}$: there is a $L^1$- integrable function $M(t)$ so that for any $t \in [0, T]$ and $\mathbf{x} \in \mathbb{R}^D$*

$$\|f(\mathbf{x}, t)\| \leq M(t)(1 + \|\mathbf{x}\|).$$

*Consider the following ODE*

$$\mathbf{x}'(\tau) = f(\mathbf{x}(\tau), \tau) \quad on \ [0, T]. \tag{12}$$

*Fix a $t \in [0, T]$, the solution operator $\mathcal{T}$ of Eq. (12) with an initial condition $\mathbf{x}_t$ is defined as*

$$\mathcal{T}[\mathbf{x}_t](s) := \mathbf{x}_t + \int_t^s f(\mathbf{x}(\tau; \mathbf{x}_t), \tau) \, \mathrm{d}\tau, \quad s \in [t, T]. \tag{13}$$

*Here $\mathbf{x}(\tau; \mathbf{x}_t)$ denotes the solution at time $\tau$ starting from the initial value $\mathbf{x}_t$. Then $\mathcal{T}$ is an injective operator. Moreover, $\mathcal{T}[\cdot](s) \colon \mathbb{R}^D \to \mathbb{R}^D$ is bi-Lipschitz; that is, for any $\mathbf{x}_t, \hat{\mathbf{x}}_t \in \mathbb{R}^D$*

$$e^{-L(s-t)} \|\mathbf{x}_t - \hat{\mathbf{x}}_t\|_2 \le \|\mathcal{T}[\mathbf{x}_t](s) - \mathcal{T}[\hat{\mathbf{x}}_t](s)\|_2 \le e^{L(t-s)} \|\mathbf{x}_t - \hat{\mathbf{x}}_t\|_2. \tag{14}$$

*Here $L := \sup_{t \in [0, T]} L(t) < \infty$. In particular, if $\mathbf{x}_t \ne \hat{\mathbf{x}}_t$, $\mathcal{T}[\mathbf{x}_t](s) \ne \mathcal{T}[\hat{\mathbf{x}}_t](s)$ for all $s \in [t, T]$.*

***Proof of Lemma 10.*** Assumptions (a) and (b) ensure the solution operator in Eq. (13) is well-defined by applying Carathéodory-type global existence theorem (Reid, 1971). We denote $\mathcal{T}[\mathbf{x}_t](s)$ as $\mathbf{x}(s; \mathbf{x}_t)$. We need to prove that for any distinct initial values $\mathbf{x}_t$ and $\hat{\mathbf{x}}_t$ starting from $t$, $\mathcal{T}[\mathbf{x}_t] \not\equiv \mathcal{T}[\hat{\mathbf{x}}_t]$. Suppose on the contrary that there is an $s_0 \in [t, T]$ so that $\mathcal{T}[\mathbf{x}_t](s_0) = \mathcal{T}[\hat{\mathbf{x}}_t](s_0)$. For $s \in [t_0, s_0]$, consider $\mathbf{y}(s; \mathbf{x}_t) := \mathbf{x}(t + s_0 - s; \mathbf{x}_t)$ and $\mathbf{y}(s; \hat{\mathbf{x}}_t) := \mathbf{x}(t_0 + s_0 - s; \hat{\mathbf{x}}_t)$. Then both $\mathbf{y}(s; \mathbf{x}_t)$ and $\mathbf{y}(s; \hat{\mathbf{x}}_t)$ satisfy the following ODE

$$\begin{cases} \mathbf{y}'(s) = -f(\mathbf{y}(s), s), \quad s \in [t, s_0] \\ \mathbf{y}(t) = \mathcal{T}[\mathbf{x}_t](s_0) = \mathcal{T}[\hat{\mathbf{x}}_t](s_0) \end{cases} \tag{15}$$

Thus, the uniqueness theorem of solution to Eq. (15) leads to $\mathbf{y}(s_0; \mathbf{x}_t) = \mathbf{y}(s_0; \hat{\mathbf{x}}_t)$, which means $\mathbf{x}_t = \hat{\mathbf{x}}_t$. This contradicts to the assumption. Hence, $\mathcal{T}$ is injective.

Now we show that $\mathcal{T}[\cdot](s) \colon \mathbb{R}^D \to \mathbb{R}^D$ is bi-Lipschitz for any $s \in [t, T]$. For any $\mathbf{x}_t, \hat{\mathbf{x}}_t \in \mathbb{R}^D$,

$$\begin{aligned} \|\mathcal{T}[\mathbf{x}_t](s) - \mathcal{T}[\hat{\mathbf{x}}_t](s)\|_2 &= \|\mathbf{x}(s; \mathbf{x}_t) - \hat{\mathbf{x}}(s; \hat{\mathbf{x}}_t)\|_2 \\ &\le \|\mathbf{x}_t - \hat{\mathbf{x}}_t\|_2 + \int_t^s \|f(\mathbf{x}(\tau; \mathbf{x}_t), \tau) - f(\hat{\mathbf{x}}(\tau; \hat{\mathbf{x}}_t), \tau)\|_2 \, \mathrm{d}\tau \\ &\le \|\mathbf{x}_t - \hat{\mathbf{x}}_t\|_2 + L \int_t^s \|\mathbf{x}(\tau; \mathbf{x}_t) - \hat{\mathbf{x}}(\tau; \hat{\mathbf{x}}_t)\|_2 \, \mathrm{d}\tau. \end{aligned}$$

By applying Gröwnwall's lemma, we obtain

$$\|\mathcal{T}[\mathbf{x}_t](s) - \mathcal{T}[\hat{\mathbf{x}}_t](s)\|_2 = \|\mathbf{x}(s; \mathbf{x}_t) - \hat{\mathbf{x}}(s; \hat{\mathbf{x}}_t)\|_2 \le e^{L(s-t)} \|\mathbf{x}_t - \hat{\mathbf{x}}_t\|_2. \tag{16}$$

On the other hand, consider the reverse time ODE of Eq. (12) by setting $\tau = \tau(u) := t + s - u$, $\mathbf{y}(u) := \mathbf{x}(t + s - u)$, and $h(\mathbf{y}(u), u) := -f(\mathbf{y}(u), t + s - u)$, then $\mathbf{y}$ satisfies the following equation

$$\mathbf{y}'(u) = h(\mathbf{y}(u), u), \quad u \in [t, s]. \tag{17}$$

Similarly, we define the solution operator to Eq. (17) as

$$\mathcal{S}[\mathbf{y}_t](s) := \mathbf{y}_t + \int_t^s h(\mathbf{y}(u; \mathbf{y}_t), u) \, \mathrm{d}u. \tag{18}$$

Here $\mathbf{y}_t$ denotes the initial value of Eq. (17) and $\mathbf{y}(u; \mathbf{y}_t)$ is the solution starting from $\mathbf{y}_t$. Due to the Carathéodory-type global existence theorem, the operator $\mathcal{S}[\cdot](s)$ is well-defined and

$$\mathcal{S}[\mathbf{x}(s; \mathbf{x}_t)](s) = \mathbf{x}_t, \quad \mathcal{S}[\hat{\mathbf{x}}(s; \mathbf{x}_t)](s) = \hat{\mathbf{x}}_t.$$

For simplicity, let $\mathbf{y}_t := \mathbf{x}(s; \mathbf{x}_t)$ and $\hat{\mathbf{y}}_t := \hat{\mathbf{x}}(s; \mathbf{x}_t)$. Also, denote the solutions starting from initial values $\mathbf{y}_t$ and $\hat{\mathbf{y}}_t$ as $\mathbf{y}(u; \mathbf{y}_t)$ and $\hat{\mathbf{y}}(u; \hat{\mathbf{y}}_t)$, respectively. Therefore, using a similar argument, we obtain

$$\begin{aligned} \|\mathbf{x}_t - \hat{\mathbf{x}}_t\|_2 &= \|\mathcal{S}[\mathbf{y}_t](s) - \mathcal{S}[\hat{\mathbf{y}}_t](s)\|_2 \\ &\le \|\mathbf{x}(s; \mathbf{x}_t) - \hat{\mathbf{x}}(s; \mathbf{x}_t)\|_2 + \int_t^s \|h(\mathbf{y}(u; \mathbf{y}_t), u) - h(\hat{\mathbf{y}}(u; \hat{\mathbf{y}}_t), u)\|_2 \, \mathrm{d}u \\ &\le \|\mathbf{x}(s; \mathbf{x}_t) - \hat{\mathbf{x}}(s; \mathbf{x}_t)\|_2 + L \int_t^s \|\mathbf{y}(u; \mathbf{y}_t) - \hat{\mathbf{y}}(u; \hat{\mathbf{y}}_t)\|_2 \, \mathrm{d}u. \end{aligned}$$

$$= \left\| \mathcal{T}[\mathbf{x}_t](s) - \mathcal{T}[\hat{\mathbf{x}}_t](s) \right\|_2 + L \int_t^s \left\| \mathbf{y}(u; \mathbf{y}_t) - \hat{\mathbf{y}}(u; \hat{\mathbf{y}}_t) \right\|_2 \mathrm{d}u.$$

By applying Gröwnwall's lemma, we obtain

$$\left\| \mathbf{x}_t - \hat{\mathbf{x}}_t \right\|_2 \leq e^{L(s-t)} \left\| \mathcal{T}[\mathbf{x}_t](s) - \mathcal{T}[\hat{\mathbf{x}}_t](s) \right\|_2.$$

Therefore,

$$e^{-L(s-t)} \left\| \mathbf{x}_t - \hat{\mathbf{x}}_t \right\|_2 \leq \left\| \mathcal{T}[\mathbf{x}_t](s) - \mathcal{T}[\hat{\mathbf{x}}_t](s) \right\|_2.$$

∎

***Proof of Proposition 6.*** With the definition of $G(\mathbf{x}_t, t, s; \boldsymbol{\phi})$, we obtain

$$G(\mathbf{x}_t, t, s; \boldsymbol{\phi}) = \frac{s}{t}\mathbf{x}_t + (1 - \frac{s}{t})g(\mathbf{x}_t, t, s; \boldsymbol{\phi})$$

$$= \mathbf{x}_t + \int_t^s \frac{\mathbf{x}_u - D_{\boldsymbol{\phi}}(\mathbf{x}_u, u)}{u} \, \mathrm{d}u.$$

Here, $g(\mathbf{x}_t, t, s; \boldsymbol{\phi}) = \mathbf{x}_t + \frac{t}{t-s} \int_t^s \frac{\mathbf{x}_u - D_{\boldsymbol{\phi}}(\mathbf{x}_u, u)}{u} \, \mathrm{d}u$. Thus, the result follows by applying Lemma 10 to the integral form of $G(\mathbf{x}_t, t, s; \boldsymbol{\phi})$.

∎

### F.6 PROOF OF PROPOSITION 7

**Lemma 11.** *Let $X$ be a random vector on $\mathbb{R}^D$ and $h \colon \mathbb{R}^D \to \mathbb{R}^D$ be a bi-Lipschitz mapping with Lipschitz constant $L > 0$; namely, for any $\mathbf{x}, \mathbf{y} \in \mathbb{R}^D$*

$$L^{-1} \left\| \mathbf{x} - \mathbf{y} \right\|_2 \leq \left\| h(\mathbf{x}) - h(\mathbf{y}) \right\|_2 \leq L \left\| \mathbf{x} - \mathbf{y} \right\|_2.$$

*Then*

$$L^{-2} \mathit{Var}(X) \leq \mathit{Var}(h(X)) \leq L^2 \mathit{Var}(X).$$

***Proof of Lemma 11.*** Let $Y$ be an i.i.d. copy of $X$. Then $h(X)$ and $h(Y)$ are also independent. Thus, $\mathrm{cov}(X, Y) = 0$ and $\mathrm{cov}(h(X), h(Y)) = 0$.

$$2\mathrm{Var}\left(h(X)\right) = \mathrm{Var}\left(h(X) - h(Y)\right)$$

$$= \mathbb{E}\left[ (h(X) - h(Y))^2 \right] - \left( \mathbb{E}\left[ h(X) - h(Y) \right] \right)^2. \tag{19}$$

Since $h(X)$ and $h(Y)$ are identically distributed, $\mathbb{E}\left[ h(X) - h(Y) \right] = \mathbb{E}\left[ h(X) \right] - \mathbb{E}\left[ h(Y) \right] = 0$. Thus, by Lipschitzness of $h$

$$2\mathrm{Var}\left(h(X)\right) = \mathbb{E}\left[ (h(X) - h(Y))^2 \right] \tag{20}$$

$$\leq L^2 \mathbb{E}\left[ (X - Y)^2 \right]$$

$$= 2L^2 \mathrm{Var}\left(X\right).$$

The final equality follows the same reasoning as in Eq. (19). Likewise, we can apply the argument from Eq. (20) to show that

$$2\mathrm{Var}\left(h(X)\right) = \mathbb{E}\left[ (h(X) - h(Y))^2 \right]$$

$$\geq L^{-2} \mathbb{E}\left[ (X - Y)^2 \right]$$

$$= 2L^{-2} \mathrm{Var}\left(X\right).$$

Therefore, $L^{-2} \mathrm{Var}\left(X\right) \leq \mathrm{Var}\left(X\right) \leq L^2 \mathrm{Var}\left(X\right)$.

∎

***Proof of Proposition 7.*** For any $n \in \mathbb{N}$, since $G_{\boldsymbol{\theta}^*}(X_n, t_n, \sqrt{1-\gamma^2}t_{n+1})$ and $Z_{n+1}$ are independent,

$$\text{Var}\left(X_{n+1}\right) = \text{Var}\left(G_{\boldsymbol{\theta}^*}(X_n, t_n, \sqrt{1-\gamma^2}t_{n+1})\right) + \text{Var}\left(Z_{n+1}\right)$$

$$= \text{Var}\left(G_{\boldsymbol{\theta}^*}(X_n, t_n, \sqrt{1-\gamma^2}t_{n+1})\right) + \gamma^2\sigma^2(t_{n+1}). \tag{21}$$

Proposition 6 implies that $G_{\boldsymbol{\theta}^*}(\cdot, t_n, \sqrt{1-\gamma^2}t_{n+1})$ is bi-Lipschitz and that for any $\mathbf{x}, \mathbf{y}$

$$\zeta^{-1}(t_n, t_{n+1}, \gamma)\|\mathbf{x} - \mathbf{y}\|_2 \leq \left\|G_{\boldsymbol{\theta}^*}(\mathbf{x}, t_n, \sqrt{1-\gamma^2}t_{n+1}) - G_{\boldsymbol{\theta}^*}(\mathbf{y}, t_n, \sqrt{1-\gamma^2}t_{n+1})\right\|_2$$

$$\leq \zeta(t_n, t_{n+1}, \gamma)\|\mathbf{x} - \mathbf{y}\|_2, \tag{22}$$

where $\zeta(t_n, t_{n+1}, \gamma) = \exp\left(2L_{\boldsymbol{\phi}}(t_n - \sqrt{1-\gamma^2}t_{n+1})\right)$. Proposition 7 follows immediately from the inequalities (21) and (22). ∎

### F.7 PROOF OF PROPOSITION 9

***Proof of Proposition 9.*** $\{p_t\}_{t=0}^T$ is known to satisfy the Fokker-Planck equation (Øksendal, 2003) (under some technical regularity conditions). In addition, we can rewrite the Fokker-Planck equation of $\{p_t\}_{t=0}^T$ as the following equation (see Eq. (37) in (Song et al., 2020b))

$$\frac{\partial p_t}{\partial t} = -\text{div}\left(\mathbf{W}_t p_t\right), \quad \text{in } (0, T) \times \mathbb{R}^D \tag{23}$$

where $\mathbf{W}_t := -t\nabla \log p_t$.

Now consider the continuity equation for $\mu_t$ defined by $\mathbf{W}_t$

$$\frac{\partial \mu_t}{\partial t} = -\text{div}\left(\mathbf{W}_t \mu_t\right) \quad \text{in } (0, T) \times \mathbb{R}^D. \tag{24}$$

Since the score $\nabla \log p_t$ is of linear growth in $\mathbf{x}$ and upper bounded by a summable function in $t$, the vector field $\mathbf{W}_t := -t\nabla \log p_t \colon [0, T] \times \mathbb{R}^D \to \mathbb{R}^D$ satisfies that

$$\int_0^T \left(\sup_{\mathbf{x} \in K} \|\mathbf{W}_t(\mathbf{x})\|_2 + \text{Lip}(\mathbf{W}_t, K) \; \mathrm{d}t\right) < \infty,$$

for any compact set $K \subset \mathbb{R}^D$. Here $\text{Lip}(\mathbf{W}_t, K)$ denotes the Lipschitz constant of $\mathbf{W}_t$ on $K$.

Thus, Proposition 8.1.8 of (Ambrosio et al., 2005) implies that for $p_T$-a.e. $\mathbf{x}$, the following reverse time ODE (which is the Eq. (6)) admits a unique solution on $[0, T]$

$$\begin{cases} \frac{\mathrm{d}}{\mathrm{d}t}X_t(\mathbf{x}) = \mathbf{W}_t\left(X_t(\hat{\mathbf{x}})\right) \\ X_T(\hat{\mathbf{x}}) = \mathbf{x}. \end{cases} \tag{25}$$

Moreover, $\mu_t = X_t \sharp p_T$, for $t \in [0, T]$. By applying the uniqueness for the continuity equation (Proposition 8.1.7 of (Ambrosio et al., 2005)) and the uniqueness of Eq. (25), we have $p_t = \mu_t = X_t \sharp p_T = \mathcal{T}_{T \to t} \sharp p_T$ for $t \in [0, T]$. Again, since the uniqueness theorem with the given $p_T$, we obtain $p_s = \mathcal{T}_{t \to s} \sharp p_t$ for any $t \in [0, T]$ and $s \in [0, t]$. ∎

