# OpenReview forum: "Consistency Trajectory Models: Learning Probability Flow ODE Trajectory of Diffusion"
_ICLR.cc/2024/Conference — ICLR 2024 poster_

### Official Review · Reviewer_DUTv · 2023-10-22

**Soundness:** 3 good
**Presentation:** 3 good
**Contribution:** 3 good
**Rating:** 6
**Confidence:** 3

**Summary:**

This paper introduces the Consistency Trajectory Model (CTM), which extends the concept of Consistency Models (CM) and score-based diffusion models. CTM is a single neural network that can efficiently output scores in a single forward pass, allowing for flexible traversal between any initial and final time points in a diffusion process. CTM combines adversarial training and denoising score matching loss to achieve competitive performance in CIFAR-10 and ImageNet-64 datasets. Additionally, CTM accommodates various diffusion model inference techniques, including exact likelihood computation.

**Strengths:**

The paper introduces a novel way to distill dynamical generative models such as Diffusion Model. To the best of my knowledge, the proposed method is novel and yields appealing results in different datasets. The experimental details are provided clearly and the paper is well-written.

**Weaknesses:**

I have some uncertainties regarding potential weaknesses, but I believe it is valuable to bring this matter to the forefront for discussion.

1. The paper itself presents a novel concept, but the methodology employed appears to be excessively complex. The distillation process involves three networks, and various techniques, such as $\lambda_{GAN}$ for ensuring stable training, are necessary to attain the results reported in Figure 15. Nevertheless, it is worth noting that this complexity also contributes to the uniqueness and innovation of the paper.

2. One of the reasons for the popularity of diffusion models lies in their scalability with respect to data complexity. In Figure 15, it is evident that the distillation of diffusion models heavily depends on the underlying GAN training structure. This prompts the question of whether it might be more straightforward to solely employ GAN training for a single NFE generation. Essentially, this would involve passing through the computationally expensive pretrained Diffusion Model, followed by the resource-intensive distillation process (as CTM requires simulating trajectories), ultimately converging to the realization that the fundamental component driving performance improvement is the GAN training objective. Given the current state of the art where GANs can achieve superior results, it raises the question of why we necessitate a complex distillation methodology, notwithstanding the novelty of such an approach.

3. Sorry for my ignorance, but I think some concepts need more elaboration. For example,  why the adversarial training can lead to better performance than the teacher model? Why Taylor's expansion is the main reason for the discretization error?

4. To the fair comparison with extremely small NFE, the fair comparison should be Consistency Models instead of EDM.

**Questions:**

1. It would be great if the authors could provide further motivation for using such complicated training objective functions with pretrained model over just training GANs.

2. It would be helpful if the author could provide the wall-clock of the training procedure. It seems like the training will be even more expensive than the diffusion model or GAN.

3. More comparisons with CMs are needed.

---

> ### Author Response · Authors · 2023-11-18
> **Rebuttal by Authors**
>
> We genuinely value the constructive feedback provided by the reviewer. In response to these insightful comments, we present the following clarifications and adjustments. To reflect the reviwer's invaluable comments, we revised the manuscript based on our responses below.
>
> **Q1. [Methodology Excessively Complex]** The paper itself presents a novel concept, but the methodology employed appears to be excessively complex. The distillation process involves three networks, and various techniques, such as $\lambda_{GAN}$ for ensuring stable training, are necessary to attain the results reported in Figure 15. Nevertheless, it is worth noting that this complexity also contributes to the uniqueness and innovation of the paper.
>
> **Response to Q1.**
>
> Thank you for your valuable feedback. In response, we highlight CM's inherent speed-quality imbalance and then concentrate on elucidating the rationale behind CTM's design. The below arguments are integrated in our paper revision on Section 3.
> - **[Why Not CM?]** CM, consisting of 2 model components (teacher/student) and 1 distillation loss (local consistency), lacks a balance between speed and quality (the red curve of Figure 9-(a) of original version). Our Theorem 2 (of first draft) proves that this unbalance is inevitable due to the overlapping between jump (from $t$ to $0$) to jump (from $t'$ to $0$), and will last as a fundamental issue as long as we train the jump only to zero-time. Therefore, CTM has been introduced to resolve this fundamental problem of jump overlapping.
> - **[Why DSM Loss?]** A neural network, well trained for small jumps, will create high-quality sample for large-NFE. However, as the network gradient is proportional to jump step-size (Lemma 1 in the original manuscript), the neural jump for small step is challenging to train. This small-step jump precision can be boosted by applying DSM loss, serving as a regularizer for infinitesimally small jump in Figure 14-(b) of the original manuscript.
> - **[Why Soft Consistency Distillation Loss?]** CTM (soft consistency) constructs distillation target $x_{target}$ using teacher with multiple random steps of ODE solving, while CM (local consistency) relies on teacher for only 1 step. This difference in teacher utilization leads to a relatively weaker signal for CM's distillation compared to CTM, as shown in Figure 13 (of the initial submission). Additionally, we observed that training with local consistency is prone to unstable to train and shows occasional divergence, whereas such instability has never seen in our experiments with soft consistency.
> - **[Why GAN Loss?]** Now the question is whether we could also improve the quality of small-NFE samples. This challenge of long-jump training can be addressed by introducing GAN training, albeit with the addition of a new component, the discriminator. Specifically, the incorporation of GAN training enhances the precision of high-frequency (fine-details), as demonstrated in the comparison between Figure 11-(b) and 11-(c) of the first version. This finding aligns with the results from VQGAN [1]. For a detailed analysis, please refer to **Q3**.
> - **[Student Beats Teacher]** It is known that DSM loss is closely linked to KL divergence [2,3,4], and GAN loss serves as a proxy for $f$-divergence [5] or IPMs [6]. Consequently, our overall loss function can be interpreted as an approximation of $$L_{distill}(teacher,student)+D_{KL}(data,student)+D_{f}(data,student).$$ Indeed, training with our loss function, we firstly observe that the well-established phenomenon in classification distillation, "student can be as good as teacher," is transferable to generation tasks. Here, $D_{KL}(data,student)+D_{f}(data,student)$ plays a role similar to the cross entropy, a statistical divergence between student and the oracle classifier, in the classification distillation.
>
> [1] Esser, Patrick, Robin Rombach, and Bjorn Ommer. "Taming transformers for high-resolution image synthesis." Proceedings of the IEEE/CVF conference on computer vision and pattern recognition. 2021.
>
> [2] Song, Yang, et al. "Maximum likelihood training of score-based diffusion models." Advances in Neural Information Processing Systems 34 (2021): 1415-1428.
>
> [3] Kim, Dongjun, et al. "Soft Truncation: A Universal Training Technique of Score-based Diffusion Model for High Precision Score Estimation." International Conference on Machine Learning. PMLR, 2022.
>
> [4] Kingma, Diederik P., and Ruiqi Gao. "Understanding Diffusion Objectives as the ELBO with Simple Data Augmentation." Thirty-seventh Conference on Neural Information Processing Systems. 2023.
>
> [5] Nowozin, Sebastian, Botond Cseke, and Ryota Tomioka. "f-gan: Training generative neural samplers using variational divergence minimization." Advances in neural information processing systems 29 (2016).
>
> [6] Arjovsky, Martin, Soumith Chintala, and Léon Bottou. "Wasserstein generative adversarial networks." International conference on machine learning. PMLR, 2017.

---

> ### Author Response · Authors · 2023-11-18
> **Rebuttal by Authors**
>
> **Q2. [Why Not Solely Adversarial Loss?]** One of the reasons for the popularity of diffusion models lies in their scalability with respect to data complexity. In Figure 15, it is evident that the distillation of diffusion models heavily depends on the underlying GAN training structure. This prompts the question of whether it might be more straightforward to solely employ GAN training for a single NFE generation. Essentially, this would involve passing through the computationally expensive pretrained Diffusion Model, followed by the resource-intensive distillation process (as CTM requires simulating trajectories), ultimately converging to the realization that the fundamental component driving performance improvement is the GAN training objective. Given the current state of the art where GANs can achieve superior results, it raises the question of why we necessitate a complex distillation methodology, notwithstanding the novelty of such an approach.
>
> **Response to Q2.**
>
> We highly appreciate for the critical review and believe answering this feedback is important. As the reviewer points out, modern GAN models are highly compatible with diffusion models. In response to the reviewer's concerns, we have considered three factors: **Flexibility**, **Training**, and **Performance**.
> - **[Flexibility]** Recent diffusion and distillation models offer a high degree of flexibility, allowing for easy fine-tuning with a minimal number of parameters or, in some cases, can be utilized without any additional training for tailored datasets. An illustrative example is presented in [7], where a combination of LoRA [8] parameters from distillation models and fine-tuned diffusion models enables the generation of samples with specific styles, eliminating the need for extra training of distillation models. In contrast, the development of such application-oriented techniques to GAN models poses greater challenges as they need to retrain GAN models for their specific usage. Additionally, CTM facilitates the likelihood computation and supports score-based inference techniques, which are not applicable to GANs. Given the significant divergence in potential use cases, we advocate for training the model with a combined adversarial loss and distillation loss, as opposed to relying solely on the adversarial loss.
> - **[Training]** Training GAN models alone remains challenging. Although state-of-the-art GAN models work well with their proposed hyperparameters and network architectures, it requires expensive hyperparameter search if we retrain or fine-tune the generator with another dataset. To mitigate the instability, most modern GAN algorithms use progressive training across resolutions, whereas CTM training is end-to-end, meaning CTM training is resolution-agnostic. Additionally, in our experiments, CTM (even with adversarial loss) shows training stability and benign performance regardless of the hyperparameter choice because the existence of distillation loss and DSM loss significantly stabilize the training. Please refer to the paragraph **[Insight from VQGAN]** in the response of **Q3**.
>
> |Model|NFE|FID$\downarrow$|IS$\uparrow$|Recall$\uparrow$|
> |-|-|-|-|-|
> ||||||
> |*Validation data*||1.41|64.10|0.67|
> ||||||
> |**DM and GAN**|
> |ADM|250|2.07|-|0.63|
> |EDM|79|2.44|48.88|**0.67**|
> |BigGAN-deep|1|4.06|-|0.48|
> |StyleGAN-XL|1|2.09|**82.35**|0.52|
> |**Distillation Models (NFE 1)**|
> |PD|1|15.39|-|0.62|
> |BOOT|1|16.3|-|0.36|
> |CD|1|6.20|40.08|0.63|
> |CTM (*ours*)|1|1.92|70.38|0.57|
> |**Distillation Models (NFE 2)**|
> |PD|2|8.95|-|0.65|
> |CD|2|4.70|-|0.64|
> |CTM (*ours*)|2|**1.73**|64.29|0.57|
>
> - **[Performance]** We would like to compare samples from CTM and GAN in various metrics with the above updated table on ImageNet $64\times64$:
>     - **[Inception Score]** StyleGAN-XL generates samples showing excessive confidence, as assessed by the metric IS, in comparison to validation data. This means that StyleGAN-XL tends to generate samples with a higher likelihood of being classified for a specific class, even surpassing the probabilities of real-world validation data. In contrast, CTM demonstrates an intermediate IS, positioning itself between the diffusion models and the GAN models. Notably, it closely aligns with the IS of the validation data.
>     - **[Recall]** CTM outperforms GAN models in sample diversity. In fact, while CTM exhibits an intermediate level of recall, the diversity of its actual samples is comparable to that of EDM or CM, as visualized in Figure 16. Furthermore, Table 1 illustrates that CTM achieved state-of-the-art likelihood ($2.43$ bits/dim) and surpassed the teacher model (EDM) ($2.56$ bits/dim) that alleviates potential concerns regarding mode collapse in CTM.
>
> [7] Luo, Simian, et al. "LCM-LoRA: A Universal Stable-Diffusion Acceleration Module." arXiv preprint arXiv:2311.05556 (2023).
>
> [8] Hu, Edward J., et al. "LoRA: Low-Rank Adaptation of Large Language Models." International Conference on Learning Representations. 2021.

---

> ### Author Response · Authors · 2023-11-18
> **Rebuttal by Authors**
>
> **Q3. [Why Adversarial Training Leads Better Performance Than Teacher?]** Sorry for my ignorance, but I think some concepts need more elaboration. For example, why the adversarial training can lead to better performance than the teacher model?
>
> **Response to Q3.**
>
> We think your valuable question directly touches the essential aspects of our contribution. Let us consider the **Property of L2 Distance** and the **Insight from VQGAN**.
>
> - **[Property of L2 Distance]** Of the possible distances between $x_{target}$ and $x_{est}$ in the distillation loss, the Euclidean L2 distance is the most popular one. The L2 distance is decomposed into two components: the low-frequency part and the high-frequency part. The high-frequency part is relatively ignorable compared to the scale of the low-frequency part [9,10]. Therefore, analogous to the autoencoder, if we train the model with the L2 distance in pixel space, the generated samples become blurry as it only captures the low-frequency part in training.
>     This imperfect generation is largely mitigated but still remains when we use a perceptual loss, i.e., the feature distance. The feature distance calculates the L2 distance between two features $f_{target}=FE(x_{target})$ and $f_{est}=FE(x_{est})$, where $FE$ is a pre-trained feature extractor. Similar to the pixel distance, the high-frequency part of feature is incorrect after optimizing with the perceptual distillation loss. We tested CTM with different feature extractors [11,12,13], resulting in consistently inferior images across all distances for NFE 1 samples. Notably, the choice of the feature extractor did not yield discernible quality differences in the generated images in both human eyes and evaluation metrics.
> - **[Insight from VQGAN]** The adversarial loss can be understood as a distance on a (feature) space that is optimized to maximize the distance between the real data and the fake data. Consequently, if the high-frequency components are not adequately trained, the discriminator captures and optimizes these frequencies, enabling the successful training of the entire frequency spectrum through the adversarial approach. Notably, VQGAN [1] incorporates a combination of adversarial loss and reconstruction loss, yielding two significant advantages: 1) it enhances the stability of adversarial training; 2) it substantially improves the quality of reconstruction. In particular, the reconstruction FID of VQGAN approaches near zero. As the primary modeling goal of distillation models is the accurate reconstruction of teacher samples, we posit that the use of adversarial loss is pivotal in enhancing distillation performance.
>
> [9] Durall, Ricard, Margret Keuper, and Janis Keuper. "Watch your up-convolution: Cnn based generative deep neural networks are failing to reproduce spectral distributions." Proceedings of the IEEE/CVF conference on computer vision and pattern recognition. 2020.
>
> [10] Kim, Nahyun, et al. "Unsupervised Image Denoising with Frequency Domain Knowledge." The 32nd British Machine Vision Conference. British Machine Vision Conference, 2021.
>
> [11] Zhang, Richard, et al. "The unreasonable effectiveness of deep features as a perceptual metric." Proceedings of the IEEE conference on computer vision and pattern recognition. 2018.
>
> [12] Tan, Mingxing, and Quoc Le. "Efficientnet: Rethinking model scaling for convolutional neural networks." International conference on machine learning. PMLR, 2019.
>
> [13] Touvron, Hugo, et al. "Training data-efficient image transformers \& distillation through attention." International conference on machine learning. PMLR, 2021.

---

> ### Author Response · Authors · 2023-11-18
> **Rebuttal by Authors**
>
> **Q4. [Why Taylor's Expansion]** Why Taylor's expansion is the main reason for the discretization error?
>
> **Response to Q4.**
>
> Thank you for the question. We apologize that the original manuscript is confusing. For analytical simplicity, let us assume we generate samples with the DDIM sampler by using the oracle score function. The denoised sample from $t$ to $t-\Delta t$ is $x_{t-\Delta t}^{DDIM}=\big(1-\frac{\Delta t}{t}\big)x_{t}+\frac{\Delta t}{t}E[x\vert x_{t}]$. However, the Taylor expansion of the integration yields that the true trajectory sample is $x_{t-\Delta t}^{true}=\big(1-\frac{\Delta t}{t}\big)x_{t}+\frac{\Delta t}{t}\big(E[x\vert x_{t}]+O(\Delta t)\big)$. Therefore, the DDIM trajectory differs from the true trajectory by $\frac{\Delta t}{t}O(\Delta t )$, which is exactly the residual term of the Taylor expansion after 2nd order. Therefore, the discretization error stems from not estimating the residual term of the Taylor expansion. We appreciate for the reviewer to point out this confusion, and we revised the manuscript accordingly.
>
> **Q5. [Lack of Comparison with CM]** To the fair comparison with extremely small NFE, the fair comparison should be Consistency Models instead of EDM.
>
> **Response to Q5.**
>
> We appreciate the reviewer's insightful comments. The original submission predominantly compares CTM with EDM and StyleGAN-XL in Section 5.1, with limited emphasis on CM. We concur with the reviewer's suggestion to enhance the discussion on the CTM and CM comparison in small-NFE settings. Accordingly, we revised Section 5.1 to provide a more detailed exploration of the differences between CTM and CM. Additionally, we emphasized the numerical results from the ablation studies in Figures 13-15 of the original submission to offer a comprehensive comparison between CTM and CM in our revision.
>
> **Q6. [Further Motivation for Such Complicated Training Objectives]** It would be great if the authors could provide further motivation for using such complicated training objective functions with pretrained model over just training GANs.
>
> **Response to Q6.**
>
> In addition to the response to **Q1. [Methodology Excessively Complex]** and **Q2. [Why Not Solely Adversarial Loss?]**, we would like to mention that diffusion models are easy for controllable generation due to its explicit form of sample generation process, whereas GAN models should be considered case-by-case for controllable generation. Moreover, functionalities in diffusion models like classifier-free guidance, cost-quality trade-off, and exact likelihood evaluations, are infeasible in GAN models. Nonetheless, we consider the GAN loss pivotal for enhancing sample quality in generative models. Hence, rather than relying solely on adversarial loss, we incorporate GAN loss alongside the reconstruction-based loss, i.e., the distillation loss.
>
> **Q7. [Training Time]** It would be helpful if the author could provide the wall-clock of the training procedure. It seems like the training will be even more expensive than the diffusion model or GAN.
>
> **Response to Q7.**
>
> |                |Epochs (official report)       |GPU Days (calculated in identical machine)                         |
> |----------------|-------------------------------|-----------------------------|
> | | | |
> |EDM|1951            |896            |
> |CM          |959            |964            |
> |CTM          |48| 112 |
>
> We compare the training time of ImageNet $64\times64$ in the above Table. We calculate the number of epochs based on the official papers and re-calculate the GPU days based on our NVIDIA Ampere machine with 40Gb memory for fair comparison. According to the Table, CTM reaches to the convergence in nearly 1/20 of the iterations required by CM. In terms of wall-clock training time, CTM completes training in approximately 1/10 of the total time needed by CM. We include the above training time in our revision at the first paragraph of Section 5.1.

---

> > ### Comment · Reviewer_DUTv · 2023-11-21
> > **Thanks for the response**
> >
> > Thank you for the detailed response. I will increase the score accordingly.

---

> > > ### Author Response · Authors · 2023-11-21
> > > **Rebuttal by Authors**
> > >
> > > We are grateful for your encouraging feedback and would like to thank you sincerely. Your comments are of great value to us, and we will take them into account carefully to enhance the quality of our paper. We appreciate your time and effort in providing us with your valuable insights.

---

### Official Review · Reviewer_RDDt · 2023-10-30

**Soundness:** 3 good
**Presentation:** 4 excellent
**Contribution:** 3 good
**Rating:** 8
**Confidence:** 4

**Summary:**

This paper proposes the Consistency Trajectory Model (CTM) that predicts any final time from any initial condition along the probability flow ODE of diffusion models. It can sample high-quality samples in one step while keeping the ability to access the score. The authors also introduce $\gamma$-sampling method, which can consistently improve sample quality with more steps. The experiments show that CTM can achieve SOTA sampling results in one step.

**Strengths:**

1. The paper is well written, with valuable insights into the proposed method. It also clearly compares to related works, positioning it well in the literature.
2. The idea is natural and nicely connects the score-based and distillation methods.
3. SOTA FID results in one step while maintaining the flexibility of accessing the original score.
4. The provided ablation studies are helpful in understanding the effect of each design component.

**Weaknesses:**

1. While adversarial training greatly improves FID, it might lead to poor mode coverage, which is a known issue of GAN. Can you measure the mode coverage of CTM on ImageNet64? For example, reporting Recall in Table 2.
2. CTM loss brings a lot more additional training cost compared to other distillation methods. It takes at least 4 model forward passes of CTM, multiple steps of the ODE solver at each training step, the cost of computing DSM loss, and also the cost of training the discriminators. Can you provide a detailed description of how many model forward passes are used per training step? Can you also provide the total training time in Table 3 and compare it to the cost of training the teacher model?
3. Training CTM seems to require many tricks like the warm-up scheme for GAN loss and a particular time sampling scheme as described in appendix D.1. But there is little ablation study about them. Can you elaborate more on how you design these training strategies and how important they are?
4. Minors:
	1. Why is the retrained CM much worse than the official report?
	2. The official report of DFNO is 3.78.
	3. The FID of Rectified Flow 4.85 is from 2-Rectified Flow with distillation. It could be more concrete about which Rectified Flow model is compared in the table.

**Questions:**

Listed in the weaknesses.

---

> ### Author Response · Authors · 2023-11-18
> **Rebuttal by Authors**
>
> We would like to express our gratitude to the reviewer for the affirmative feedback. In response to these warm comments, we endeavor to address them by offering the following clarifications and adjustments. Also, in our paper revision, we faithfully reflected the reviewer's valuable feedback.
>
> **Q1. [Diversity Metrics]** While adversarial training greatly improves FID, it might lead to poor mode coverage, which is a known issue of GAN. Can you measure the mode coverage of CTM on ImageNet $64\times64$? For example, reporting Recall in Table 2.
>
> **Response to Q1.**
> |   Model             |NFE       |FID$\downarrow$ | IS$\uparrow$ | Recall$\uparrow$                         |
> |--------------|-|-|-------------------------------|-----------------------------|
> | | | | | |
> |*Validation data* | | 1.41 | 64.10 |0.67  |
> | | | | | |
> | **DM and GAN** |
> |ADM          |250 | 2.07 | - | 0.63 |
> |EDM          |79 | 2.44 | 48.88 | **0.67** |
> |BigGAN-deep|1|4.06|-|0.48|
> |StyleGAN-XL|1|2.09|**82.35**|0.52|
> | **Distillation Models (NFE 1)** |
> |PD|1|15.39|-|0.62|
> |BOOT|1|16.3|-|0.36|
> |CD|1|6.20|40.08|0.63|
> |CTM (*ours*)|1|1.92|70.38|0.57|
> | **Distillation Models (NFE 2)** |
> |PD|2|8.95|-|0.65|
> |CD|2|4.70|-|0.64|
> |CTM (*ours*)|2|**1.73**|64.29|0.57|
>
>
> We report the updated performance in the above Table, including recall, and contained this result and below arguments in our revised Section 5.1. According to the Table, CTM exhibits an intermediate level of recall, positioning itself between EDM and StyleGAN-XL. Nevertheless, we emphasize that this result was imperceptible through human judgment, as the diversity of CTM samples is comparable to that of EDM or CM, as illustrated by randomly generated samples in Figure 16. Indeed, in likelihood metric, CTM achieved state-of-the-art likelihood ($2.43$ bits/dim) that surpassed the teacher model (EDM) ($2.56$ bits/dim), that largely alleviates the issue of mode collapse.
>
> **Q2. [How Many Forward Passes Per Iteration?]** CTM loss brings a lot more additional training cost compared to other distillation methods. It takes at least 4 model forward passes of CTM, multiple steps of the ODE solver at each training step, the cost of computing DSM loss, and also the cost of training the discriminators. Can you provide a detailed description of how many model forward passes are used per training step? Can you also provide the total training time in Table 3 and compare it to the cost of training the teacher model?
>
> **Response to Q2.**
> |                |Epochs (official report)       |GPU Days (calculated in identical machine)     | #Forward Passes Per Iteration                    |
> |-|----------------|-------------------------------|-----------------------------|
> | | | | |
> |EDM|1951            |896            | 1 |
> |CM          |959            |964            | 5 |
> |CTM          |48| 112 | 9+n |
>
> We compare the training time of ImageNet $64\times64$ in the above Table. As the reviewer expected, CTM requires more model evaluations per iteration due to the multiple steps of the ODE solver. More precisely, CM computes $5$ forward passes with 1 for teacher (wrapped with `no_grad` in PyTorch), $2$ for student, and the other $2$ for LPIPS. CTM takes 9+n forward passes with n for the number of ODE steps (teacher wrapped with `no_grad` in PyTorch), $2$ for $s\rightarrow 0$ prediction $G_{sg(\theta)}(\cdot,s,0)$, $2$ for student ($G_{\theta}(\cdot,t,s)$ and $G_{sg(\theta)}(\cdot,u,s)$), $2$ for LPIPS, $1$ for DSM loss, and the other $2$ for discriminator. The number of ODE steps $n$ varies each iteration, depending on random $u$. For ImageNet experiment, $n$ is $10$ in average.
>
> We emphasize that CTM achieves significantly faster convergence in terms of wall-clock time compared to EDM and CM, despite requiring more function evaluations. To ensure a fair comparison, we calculate the number of epochs based on the official papers and recalculate the GPU days using our NVIDIA A100 machine with 40GB memory. As indicated in the above table, CTM reaches convergence in nearly $1/10$th of the total training time required by the teacher model (EDM) and the baseline (CM). We included the above argument in the first paragraph of our revised Section 5.1.

---

> ### Author Response · Authors · 2023-11-18
> **Rebuttal by Authors**
>
> **Q3. [How Important Does Tricks?]** Training CTM seems to require many tricks like the warm-up scheme for GAN loss and a particular time sampling scheme as described in appendix D.1. But there is little ablation study about them. Can you elaborate more on how you design these training strategies and how important they are?
>
>
>
>
> **Response to Q3.**
>
> We appreciate for the constructive comments. Below, we have considered five factors separately: **Warm-up**, **Time Sampling**, **GAN Details**, **EMA**, and **Adaptive Weights**. In our revision, we included below discussions in the revised manuscript more explicitly.
>
> - **[Warm-up]** We apply warm-up for GAN training mainly to stabilize CTM training at the initial phase. The insights of VQGAN is that it is better to incorporate the GAN loss on top of the reconstruction loss once the autoencoder captures the image semantic after sufficient iterations. In our experiments, we found that applying this warm-up to generator is more important than discriminator, and we did not warm-up the discriminator in our training.
> - **[Time Sampling]** An well-designed time sampling is critical for score training. When the forward-time SDE is represented by $dx_{t}=\sqrt{2t}dw_{t}$, the large-time score function closely resembles a contraction function towards the origin, exhibiting nearly proportional to $-x_{t}$. Consequently, in EDM, the learning target (oracle score) is predominantly linear for large-time, rendering frequent sampling on large-time unnecessary. In contrast, CTM learns complicated trajectory across the diffusion time $s<t$, demanding that the neural network deviates significantly from linearity when processing large-time inputs. In the process of learning this non-linear complexity for $s<t$, thus, the neural network easily forgets the linearity for $s=t$ on large-time. Without sampling more on large-time for better score matching, the FID significantly deteriorates with large-NFE sampling.
> - **[GAN Details]** We mostly follow the practice of StyleGAN-XL in our GAN training. In particular, except the fake data construction, our GAN training mirrors that of StyleGAN-XL, including upscaling to $224\times224$, using cross-channel mixing, and processing with multiple discriminators for different feature maps. While these training details could be substituted with other well-crafted components of state-of-the-art GAN models, we defer further exploration of alternative configurations to future study.
> - **[EMA]** When constructing $sg(\theta)$, we adopt the practice from CM by incorporating Exponential Moving Average (EMA) to enhance training stability. While CM used the EMA scale ($\mu$) of 0.0 on CIFAR-10 and 0.95 on ImageNet $64\times64$, we experiment with higher values of $\mu$, specifically 0.999 or 0.9999. This adjustment significantly contributes to stabilizing GAN training, and notably, in our experiments on CTM, we have observed minimal FID fluctuation.
> - **[Adaptive Weights]** In addition to warm-up and EMA, we apply the adaptive weighting for $\lambda_{DSM}$ and $\lambda_{GAN}$ to scale each component (the distillation loss, the DSM loss, and the GAN loss) at identical level. The adaptive weighting computes $\lambda_{DSM}=stopgrad(\frac{\Vert \nabla_{\theta_{L}}L_{CTM}(\theta;\phi) \Vert}{\Vert \nabla_{\theta_{L}}L_{DSM}(\theta) \Vert})$ and $\lambda_{GAN}=stopgrad(\frac{\Vert \nabla_{\theta_{L}}L_{CTM}(\theta;\phi) \Vert}{\Vert \nabla_{\theta_{L}}L_{GAN}(\theta,\eta) \Vert})$, where $\theta_{L}$ is the last layer of the UNet output block. This adaptive weight prevents one component to dominate other factors. For instance, without the adaptive weight, after sufficient iterations, the GAN loss suddenly breaks the stability and ruins CTM training.
>
>
> **Q4. [Why Not Regenerable?]** Why is the retrained CM much worse than the official report?
>
> **Response to Q4.**
>
> The authors of CM provided distinct codebases for CIFAR-10 (implemented in Jax) and ImageNet (implemented in PyTorch). Our retrained CM is grounded in the official PyTorch code (designed for ImageNet), with all hyperparameters calibrated to align with the released Jax code for regenerating CM on CIFAR-10.
>
> While the reasons for the performance disparity between PyTorch and Jax remain unclear, we advocate for a unified codebase to propel the advancement of the distillation community. In alignment with this vision, we have opted to develop our code using PyTorch and will subsequently release a comprehensive, unified codebase that spans across various datasets.
>
> **Q5. [More Careful Performance Citation Needed]** The official report of DFNO is 3.78. The FID of Rectified Flow 4.85 is from 2-Rectified Flow with distillation. It could be more concrete about which Rectified Flow model is compared in the table.
>
> **Response to Q5.**
>
> We sincerely appreciate for detailed review and the reviewer's devotion to the community. We modified the performances of DFNO and Rectified Flow in our revision.

---

> > ### Comment · Reviewer_RDDt · 2023-11-20
> >
> > Thanks for your detailed response. My concerns are addressed. I have read and considered all the author's responses and other reviewer's reviews. I think it's a good paper and will vote for acceptance.

---

> > > ### Author Response · Authors · 2023-11-21
> > > **Rebuttal by Authors**
> > >
> > > We express our sincere appreciation for the reviewer's comments and positive feedback! We'll consider your review carefully and make any necessary revisions to enhance the quality of our paper revision. Thank you.

---

### Official Review · Reviewer_csVy · 2023-10-30

**Soundness:** 4 excellent
**Presentation:** 4 excellent
**Contribution:** 2 fair
**Rating:** 6
**Confidence:** 4

**Summary:**

The work proposes a new work that is able to train generative models from scratch and distill a faster generative model from teacher models. At its core, CMT tries to learn an ODE transition function, which is achieved by minimizing a combination of soft-matching loss, and DSM loss. Moreover, the authors suggest additional GAN to treat numerical accuracy for perceptual quality. The work achieved SOTA performance with the help of a pre-train teacher model and additional gan discriminator.

**Strengths:**

- SOTA performance
- Theorem 2 and related analysis and Classifier-Rejection Sampling is interesting and non-trivial. They are new and novel to me.

**Weaknesses:**

Many of the losses and concepts are not entirely new, and there are some existing works that share similar ideas. Since this work achieves state-of-the-art (SOTA) performance, I encourage the authors to include more experimental details (see below) and empirical analyses in the main paper.

Some recent SOTA distillation or generative works are difficult to reproduce, such as the Guided Distilled Stable Diffusion (Meng et al.). The authors also encountered difficulties in reproducing CD, even when the source code is available. I would like the authors to discuss the challenges and unstable factors associated with reproducing CD, and whether similar factors will affect CMT. Have the authors encountered any stability issues?

The authors mentioned that the weights of GAN are tuned similarly to VQGAN. Can the authors provide concrete implementation details?

**Questions:**

See above

---

> ### Author Response · Authors · 2023-11-18
> **Rebuttal by Authors**
>
> We sincerely value the thorough feedback provided by the reviewer. In response to these helpful comments, we aim to address them by offering the following clarifications.
>
> **Q1. [Interesting Contributions]** Theorem 2 and related analysis and Classifier-Rejection Sampling is interesting and non-trivial. They are new and novel to me.
>
> **Response to Q1.**
>
> We are grateful to the reviewer for the positive feedback and recognition of the intriguing nature of our analysis.
>
> - **[Theorem 2 and Related $\gamma$-sampling]** Theorem 2 clarifies the underlying rationale for the deterioration of CM's multistep sampling with an increase in NFE, and it underscores the inevitable estimation error resulting from any time overlap in the sampling process. Moreover, in the general framework of our innovative $\gamma$-sampling, we note that a decrease in the $\gamma$ value corresponds to a gradual reduction in time overlap. Consequently, the estimation error of $\gamma$-sampling is minimized with $\gamma=0$ as evidenced in both theory (Theorem 2) and experiments (Figure 8).
>
>     In connection with $\gamma$-sampling, it is noteworthy to highlight another intriguing property. Fundamentally, there is no Ito’s SDE that aligns with our $\gamma$-sampling in terms of  sample paths. Therefore, our $\gamma$-sampling introduces an entirely new family of sampling technique in diffusion models. A detailed explanation is provided in Appendix C.2.
>
> - **[Classifier-rejection Sampling]** We also acknowledge the reviewer's recognition of the non-trivial nature of classifier-rejection sampling. Indeed, classifier-rejection sampling can be applied in various contexts. For instance, we observed improved sample quality when rejecting samples at intermediate times rather than at zero time.  A potential refinement of this algorithm includes the following steps: 1) synthesizing a diffused sample with a single-step jump; 2) determining sample acceptance in the diffused space based on classifier probability; 3) if rejected, diffusing the sample with sufficient noise to eliminate undesirable semantic aspects; 4) denoising the sample back to time zero with a single jump. In our experiments with the CelebA 64x64 dataset, this classifier-rejection sampling reduced the FID from 1.9 to 1.2.
>
> **Q2. [Idea Not New]** Many of the losses and concepts are not entirely new, and there are some existing works that share similar ideas.
>
> **Response to Q2.**
>
> We deeply appreciate to the reviewer's devotion to the community. While some aspects of the methodology draw inspiration from existing concepts, we would like to emphasize our novel contributions, including:
> - A single model attaining all the advantages of score-based (e.g., likelihood computation) and distillation models (e.g., fast sampling);
> - $\gamma$-sampling for flexible sampling;
> - Soft matching for efficient and stabilized training;
> - The use of auxiliary losses for better student training.
>
> We further elaborate on our discussion below.
> - **[A Unified Single Model]** Through our innovative $g$-modeling (in Lemma 1 of the original manuscript), CTM can leverage well-established score-based inference techniques (including exact likelihood computation, ODE/SDE-based sampling, classifier-free sampling), and also few NFE sampling characteristic of distillation models with *a single model*.
> - **[$\gamma$-Sampling]** Contrastive of CM's multistep sampler that shows unclear NFE-FID curve (the red line of Figure 9-(a) in the original submission), our $\gamma$-sampling with $\gamma=0$ provides a clear trade-off between NFEs and sample quality, as illustrated in the blue line of Figure 9-(a).
> - **[Soft Consistency]** CTM (soft consistency loss) constructs the distillation target $x_{target}$ using the teacher model with multiple (random number of) steps of ODE solving, while CM (local consistency loss) relies on the teacher for only 1 step. This difference in teacher utilization leads to a relatively weaker learning signal for CM's distillation compared to CTM, as shown in Figure 13 of the first draft. Additionally, we observed that training with local consistency is prone to instability and occasional divergence, whereas such training instability has never been observed in our training with soft consistency.
> - **[Good Student]** It is known that DSM loss is closely linked to KL divergence [1,2,3], and GAN loss serves as a proxy for $f$-divergence [4] (or IPMs [5,6]). Consequently, our overall loss function can be interpreted as an approximation of $$L_{distill}(teacher,student)+D_{KL}(data,student)+D_{f}(data,student).$$ Indeed, training with our loss function, we firstly observed that the well-established phenomenon in classification distillation, "student can be as good as teacher," is transferable to generation tasks. Here, $D_{KL}(data,student)+D_{f}(data,student)$ plays a role similar to the cross entropy, a statistical divergence between student and the oracle classifier, in the classification distillation.

---

> > ### Comment · Reviewer_csVy · 2023-11-20
> >
> > do authors miss the reference for [1,2,3,4,5,6] in the response?

---

> > > ### Author Response · Authors · 2023-11-21
> > > **Rebuttal by Authors**
> > >
> > > **do authors miss the reference for [1,2,3,4,5,6] in the response?**
> > >
> > > $\rightarrow$ We apologize for the missing references [1,2,3,4,5,6]. Below we list the missing references. Thank you for bringing this to our attention.
> > >
> > > [1] Song, Yang, et al. "Maximum likelihood training of score-based diffusion models." Advances in Neural Information Processing Systems 34 (2021): 1415-1428.
> > >
> > > [2] Kim, Dongjun, et al. "Soft Truncation: A Universal Training Technique of Score-based Diffusion Model for High Precision Score Estimation." International Conference on Machine Learning. PMLR, 2022.
> > >
> > > [3] Kingma, Diederik P., and Ruiqi Gao. "Understanding Diffusion Objectives as the ELBO with Simple Data Augmentation." Thirty-seventh Conference on Neural Information Processing Systems. 2023.
> > >
> > > [4] Nowozin, Sebastian, Botond Cseke, and Ryota Tomioka. "f-gan: Training generative neural samplers using variational divergence minimization." Advances in neural information processing systems 29 (2016).
> > >
> > > [5] Arjovsky, Martin, Soumith Chintala, and Léon Bottou. "Wasserstein generative adversarial networks." International conference on machine learning. PMLR, 2017.
> > >
> > > [6] Sriperumbudur, Bharath K., et al. "Hilbert space embeddings and metrics on probability measures." The Journal of Machine Learning Research 11 (2010): 1517-1561.

---

> ### Author Response · Authors · 2023-11-18
> **Rebuttal by Authors**
>
> **Q3. [Include More Experimental Details]** Since this work achieves state-of-the-art (SOTA) performance, I encourage the authors to include more experimental details (see below) and empirical analyses in the main paper.
>
> **Response to Q3.**
>
> We acknowledge the reviewer's comment regarding the initial submission's insufficient description of the experimental setup. We carefully proofread the manuscript and made a major revision in the Experimental Section to enhance clarity and improve the understanding of the experiments. Please refer Table 1 and Section 5.1 for detailed explanations. Also, we are planning to release our code upon the acceptance.
>
> **Q4. [Discuss Challenges and Unstable Factors]** I would like the authors to discuss the challenges and unstable factors associated with reproducing CD, and whether similar factors will affect CMT. Have the authors encountered any stability issues?
>
> **Response to Q4.**
>
> Upon retraining CM using its official PyTorch code, we found that CM training occasionally diverge and fail to stabilize thereafter. This training instability occurred randomly, irrespective of hyperparameter settings. Our investigation revealed that the primary cause of training divergence was the local matching (consistency distillation) loss of CM. We successfully mitigated training instability by modifying the loss function to our proposed *soft matching*. This adjustment from local consistency to soft consistency is empirically demonstrated to be a critical factor for achieving stable training. Following the reviewer's suggestion, we included and discussed the various factors for training stability in our manuscript revision at Sections 3.1, 3.2, 3.3, 5.1, and 5.2.
>
> **Q5. [Implementation Details]** The authors mentioned that the weights of GAN are tuned similarly to VQGAN. Can the authors provide concrete implementation details?
>
> **Response to Q5.**
>
> In alignment with VQGAN, we adopt adaptive weighting denoted as  $\lambda_{GAN}=stopgrad(\frac{\Vert \nabla_{\theta_{L}}L_{CTM}(\theta;\phi) \Vert}{\Vert \nabla_{\theta_{L}}L_{GAN}(\theta,\eta) \Vert})$, where $\theta_{L}$ is the last layer of the UNet output block. The stop gradient technique is employed to render $\lambda_{GAN}$ independent of $\theta$ during gradient computation. This adaptive weighting mechanism contributes to the further stabilization of training. We included this adaptive weighting trick with details in the last paragraph of our revised Section 3.3.

---

> > ### Comment · Reviewer_csVy · 2023-11-20
> >
> > Thanks for your response.
> >
> > Based on https://arxiv.org/abs/2203.06026, the gan loss may be helpful for getting a good fid while does not improve image quality in general. For Figure 13, can authors report quantitative results without gan loss in a table format?

---

> > > ### Author Response · Authors · 2023-11-21
> > > **Rebuttal by Authors**
> > >
> > > **Based on https://arxiv.org/abs/2203.06026, the gan loss may be helpful for getting a good fid while does not improve image quality in general.**
> > >
> > > $\rightarrow$ We appreciate the reviewer for highlighting this paper discussing the discrepancy between humans' perceptual judgment and FID scores. In response, we have included a new figure (Figure   15 in Appendix C.4) in the revised manuscript that compares samples generated by CTM (NFE 1) with and without GAN loss. The figure illustrates that samples utilizing the auxiliary GAN loss exhibit enhanced fine details, effectively addressing high-frequency alisasing artifacts. Moreover, improvements in overall shapes (butterfly/background) and features (brightness/contrast/saturation) are evident in these samples. Although https://arxiv.org/abs/2203.06026 discusses the possibility that FID improvement may not necessarily correlate with an actual enhancement in human perceptual judgement, our observations in Figure 15 demonstrates that, for CTM, the enhancement achieved through GAN is perceptually discernible in human judgment.

---

> > > ### Author Response · Authors · 2023-11-21
> > > **Rebuttal by Authors**
> > >
> > > **For Figure 13, can authors report quantitative results without gan loss in a table format?**
> > >
> > > $\rightarrow$ Thank you for your suggestion. Following the helpful comment, we added a new table that reports and compares the quantitative numbers with/without GAN loss in Table 4 of Appendix C.4. Also, we provide below table that shows the results of the FID curve (NFE=1 results) by iterations.
> > >
> > > | Model (NFE 1)\Iterations | 11K | 21K | 31K | 41K | 100K |
> > > |-|-|-|-|-|-|
> > > | | | | | | |
> > > | CTM + DSM | 7.31 | 6.42 | 5.97 | 5.78 | 5.28 |
> > > | CTM + DSM + GAN | | 3.63 | 2.09 | 2.04 | 2.01 |

---

> > > > ### Author Response · Authors · 2023-11-23
> > > > **Looking forward to your response**
> > > >
> > > > Dear Reviewer 1 (csVy),
> > > >
> > > > We sincerely appreciate your devotion and constructive comments to our work. As the reviewer-author discussion period left 1 hour, we would like to ask if there is any other concerns and questions that we could discuss more for better revision.
> > > >
> > > > Warm regards,
> > > >
> > > > Authors 7687

---

### Official Review · Reviewer_6usj · 2023-11-10

**Soundness:** 3 good
**Presentation:** 3 good
**Contribution:** 3 good
**Rating:** 6
**Confidence:** 4

**Summary:**

The authors extend the recently introduced Consistency Models framework. Consistency Models were proposed as a distillation method to accelerate the sampling of diffusion models. They are trained to predict the output of the Probability Flow ODE trajectory. The authors of this work extend this method by training a model that can predict any intermediate point of the Probability Flow ODE trajectory. This has two benefits: i) the trained model can recover the conditional expectation (and the score) when the starting and the final time of the Probability Flow ODE are chosen to be close, ii) the framework provides a natural way to perform multistep sampling. The authors demonstrate successfully that their method works by achieving a new SOTA on CIFAR-10 and ImageNet.

**Strengths:**

Overall, I think this is a great submission.

- The topic of the paper is timely. Consistency Models have been recently proposed and they offer a promising solution for one-step generation with diffusion models.

- The proposed method is simple, intuitive, and novel.

- This framework provides a natural way to perform multistep sampling. It mitigates a significant limitation of Consistency Models which is that the performance was not increasing significantly even if more computational budget was available.
- The method has great empirical success -- it achieves a new SOTA for single-step generation in ImageNet and CIFAR-10.
- The comparison with the baselines is thorough.
- The presentation of the method is clear. I particularly liked Figure 2.

**Weaknesses:**

I don't see any major issues, but there are a couple of small concerns.
- The presentation of the paper could be improved a little. Figure 1 is not very helpful I think -- I would advocate making Figure 2 the main Figure of the paper. There are some typos that could be fixed with more careful reading. For example, there is a parenthesis missing in the citation at the top of page 3.
- In the definition of the $g$ function (Lemma 1), I think the integral bounds $s, t$ have to change order (or a minus sign needs to be added in front of the integral).
- Some of the mathematical details are unnecessarily complicated or a little sloppy. For example, the fact that $G$ can be written in terms of $g$ is trivial and hence it should not be part of the lemma. What Lemma proves is that at the limit of $s\to t$, $g$ becomes the conditional expectation. This can be easily seen by approximating the integrand as being constant in $[t, s]$ -- similar to how the first-order ODE solvers operate.
- Following this point, I think being able to approximate the conditional expectation with the same network is not really important. For this method to work, we need to start from a pre-trained score model anyway. Additionally, the proposed method for learning this doesn't actually work because of numerical instabilities, and hence the authors resort to adding an additional loss term.
- Section 3.2 is a little bit confusing. The notation added is too much and makes it hard to follow what's going on. I think (?) that the main point is that another network is used for the ODE solving and the student network learns these predictions. If that's the case, this needs to be a little bit more clear.

**Questions:**

See weaknesses above. Also, why would the student outperform the teacher? And why this method gives better results that Consistency Models when s=0?

---

> ### Author Response · Authors · 2023-11-18
> **Rebuttal by Authors**
>
> We truly appreciate the reviewer of the constructive feedback. In light of these insightful comments, we would like to address them by providing the following clarifications and adjustments. Also, to faithfully reflect the reviewer's valuable comments, we revised our manuscripts (particularly on Section 3), accordingly.
>
> **Q1. [Presentation and Proofreading]** The presentation of the paper could be improved a little. Figure 1 is not very helpful I think -- I would advocate making Figure 2 the main Figure of the paper. There are some typos that could be fixed with more careful reading. For example, there is a parenthesis missing in the citation at the top of page 3.
>
> **Response to Q1.**
>
> We acknowledge the reviewer's comment regarding the significance of figures in our manuscript. In response to this feedback, we propose relocating Figure 1 to the Appendix and featuring Figure 2 as the primary thumbnail. Also, we have rechecked the manuscript thoroughly and revised typos including the citation of page 3 in our revision. We appreciate for the reviewer of the careful review.
>
> Here, we wish to provide a brief contextual explanation for Figure 1 (of the original manuscript), and (if encouraged) reflect this discussion in the manuscript revision. One of the key motivations driving this research is the creation of a novel family of multi-purpose models that extend beyond traditional score-based models (requiring a minimum of $10$ NFE for sampling) and distillation models (whose performance wasn't consistently enhanced with large NFE). In larger-scale learning scenarios, such as dealing with diverse data modalities, training and saving both score-based and distillation models separately may not be cost-effective. Additionally, practitioners face the dilemma of choosing between a score-based model and a distillation model for their application, unable to fully leverage the merits of both. However, CTM, accommodating training from scratch, distillation, and score-based approaches, offers a unified and versatile model. This single model seamlessly integrates all the advantages of separate models, catering to diverse applications and providing a more economical and practical solution.

---

> ### Author Response · Authors · 2023-11-18
> **Rebuttal by Authors**
>
> **Q2. [Integral Order]** In the definition of the $g$ function (Lemma 1), I think the integral bounds $s,t$ have to change order (or a minus sign needs to be added in front of the integral).
>
> **Response to Q2.**
>
> We thank the reviewer for the detailed feedback. In response to the reviewer's comment, we carefully recheck our derivations with particular emphasis on the time direction and integration order. Based on our careful examination, to the best of our knowledge, we confirm that there are no mathematical flaws in our original manuscript. For the benefit of clarity and transparency, we present a comprehensive account of the derivation below.
>
> The reverse-time SDE is
> $$dx_{T-\tau}=g^{2}(T-\tau)\nabla\log{p_{T-\tau}(x_{T-\tau})}d t+g(T-\tau)dw_{\tau},
> $$ where the forward-time SDE is given by $$dx_{\tau}=g(\tau)dw_{\tau}.$$
> The reverse-time PF-ODE thus becomes $$dx_{T-\tau}=\frac{1}{2}g^{2}(T-\tau)\nabla\log{p_{T-\tau}(x_{T-\tau})}d \tau.$$
> Therefore, by integrating from $T-t$ to $T-s$ ($s<t$), we obtain $$x_{s}-x_{t}=\int_{T-t}^{T-s}\frac{1}{2}g^{2}(T-\tau)\nabla\log{p_{T-\tau}(x_{T-\tau})}d \tau.$$
> By change-of-variable with $u=T-\tau$, the equation is derived to be $$x_{s}=x_{t}-\int_{t}^{s}\frac{1}{2}g^{2}(u)\nabla\log{p_{u}(x_{u})}d u.$$
> With $g(u)=\sqrt{2u}$ and the Tweedie's formula $\nabla\log{p_{u}(x_{u})}=\frac{E[x\vert x_{u}]-x_{u}}{u^{2}}$, we derive Eq. (2) in our paper: $$x_{s}=G(x_{t},t,s)=x_{t}+\int_{t}^{s}\frac{x_{u}-E[x\vert x_{u}]}{u}d u.$$
> Now, we derive the following equations:
> $$
> G(x_{t},t,s)=\frac{s}{t}x_t+ \Big(1-\frac{s}{t}\Big) x_t+ \int_{t}^{s} \frac{x_{u}-E[x\vert x_{u}]}{u} d u
> =\frac{s}{t}x_t+ \Big(1-\frac{s}{t}\Big) \bigg[ x_t + \frac{1}{(1-\frac{s}{t})} \int_{t}^{s} \frac{x_{u}-E[x\vert x_{u}]}{u} d u \bigg] $$
> $$=\frac{s}{t}x_{t}+\Big(1-\frac{s}{t}\Big) \bigg[x_{t}+\frac{t}{t-s}\int_{t}^{s}\frac{x_{u}-E[x\vert x_{u}]}{u}d u \bigg]
> =\frac{s}{t}x_{t}+\Big(1-\frac{s}{t}\Big)g(x_{t},t,s),
> $$
> where $g(x_{t},t,s):=x_{t}+\frac{t}{t-s}\int_{t}^{s}\frac{x_{u}-E[x\vert x_{u}]}{u}d u$.

---

> ### Author Response · Authors · 2023-11-18
> **Rebuttal by Authors**
>
> **Q3. [Unnecessarily Complicated Mathematical Details]** Some of the mathematical details are unnecessarily complicated or a little sloppy. For example, the fact that $G$ can be written in terms of $g$ is trivial and hence it should not be part of the lemma. What Lemma proves is that at the limit of $s\rightarrow t$, $g$ becomes the conditional expectation. This can be easily seen by approximating the integrand as being constant in $[t,s]$-- similar to how the first-order ODE solvers operate.
>
> **Response to Q3.**
>
> As the reviewer points out, that $g$ becomes the conditional expectation could be easily seen by approximating the integrand as being constant in $[t,s]$. Following the reviewer's comment, we revised the manuscript to eliminate Lemma 1 in Section 3.1 for easy understanding in our revision.
>
>
> **Q4. [Approximating Conditional Expectation]** Following this point, I think being able to approximate the conditional expectation with the same network is not really important.
>
> **Response to Q4.**
>
> Thank you for the insightful review. We agree with the reviewer's observation that once an any-to-any jump $G(\mathbf{x}_{t},t,s)$ is approximated, the necessity for infinitesimal jumps is eliminated. Below, given that we agree with the reviewer, we would like to provide the detailed motivation of our model design and discuss that the infinitesimal jump is gained as a side effect of our model design. We have revised Section 3.1 in line of below response of reviewer's question.
>
>
> - **[Constrained to Unconstrained]** Directly approximating $G$ using a neural network is challenging due to the constrained optimization problem posed by the initial condition  $G_{\theta}(x_{t},t,t)=x_{t}$: $$\min_{\theta}L(\theta)\quad\text{subject to}\quad G_{\theta}(x_{t},t,t)=x_{t}.$$ By setting $c_{skip}(t,s)=\frac{s}{t}$ in $G_{\theta}(x_{t},t,s)=c_{skip}(t,s)x_{t}+c_{out}(t,s)NN_{\theta}(x_{t},t,s)$, the initial condition $G_{\theta}(x_{t},t,t)=x_{t}$ is satisfied for free, transforming the constrained problem into an unconstrained optimization task $\min_{\theta}L(\theta)$.
> - **[$c_{out}$ for Stable Training]** In our design, $G_{\theta}=c_{skip}(t,s)x_{t}+c_{out}(t,s)NN_{\theta}$, $c_{out}(t,s)$ acts as a time-weighting factor for $NN_{\theta}$ and influences the loss gradient. We experimented with $c_{out}(t,s)\equiv 1$, indicating equal contribution of various time, but this resulted in extremely unstable training. Therefore, inspired by the Euler solver (Appendix C.1), we adopted $c_{out}(t,s)=1-\frac{s}{t}$ as a mathematically oriented weighting function, which contributed to stabilizing the training.
> - **[Denoiser As Bonus]** The conditional expectation (i.e., denoiser function), as a natural consequence of our design, is now approximated by $NN_{\theta}$ as $s\rightarrow t$. This inherent capability allows CTM to naturally generalize to both diffusion models ($s=t$) and distillation models ($s<t$). After examining various candidates for $c_{skip}$ and $c_{out}$, we found that the Euler-solver motivated design of $c_{skip}(t,s)=\frac{s}{t}$ and $c_{out}(t,s)=1-\frac{s}{t}$ demonstrated to be the most stable, to the best of our knowledge. Please refer to Appendix C.1 for a detailed analysis in this regard.
>
> **Q5. [Pre-trained Score Model Needed?]** For this method to work, we need to start from a pre-trained score model anyway.
>
> **Response to Q5.**
>
> We appreciate the reviewer for the well-founded comment. To answer, we kindly remind the reviewer of the last paragraph in Section 5, named with **Training Without Pre-trained DM**. Our finding is that there is no training instability when we randomly initialize the model. Rather, from the DSM loss $L_{DSM}$ in our model training loss, our model eventually approximate the score function well (without the need of pre-trained DM) and this property builds the positive feedback loop with CTM loss $L_{CTM}$: a better score approximation leads a better long jump estimation and vice versa as training evolves.
>
> **Q6. [Numerical Instability]** Additionally, the proposed method for learning this doesn't actually work because of numerical instabilities.
>
> **Response to Q6.**
>
> We presume the reviewer thinks the proposed method suffers from numerical instability by $c_{out}(t,s)=1-\frac{s}{t}$. However, indeed, the opposite happened in experiments. If $c_{out}(t,s)\equiv 1$, the training always diverged and was not stabilized with any tricks that we tried, including time reweighting or importance sampling. In contrast, $c_{out}(t,s)=1-\frac{s}{t}$ significantly stabilize the training.

---

> ### Author Response · Authors · 2023-11-18
> **Rebuttal by Authors**
>
> **Q7. [Additional Loss]** The authors resort to adding an additional loss term.
>
> **Response to Q7.**
>
> To balance the speed-quality trade-off (as depicted by the blue curve in Figure 9-(a)), accurate training of both small-step and large-step jumps is crucial. However, the reviewer rightly points out that if $c_{out}(t,s)=1-\frac{s}{t}$, the small-step neural jump is inaccurately trained due to the multiplication by the factor $1-\frac{s}{t}$, which converges to zero as $s\rightarrow t$. The DSM loss serves as a regularizer for the accurate training of small-step jump because the DSM loss learns the infinitesimally small neural jump.
>
> **Q8. [Confusing Section]** Section 3.2 is a little bit confusing. The notation added is too much and makes it hard to follow what's going on. I think (?) that the main point is that another network is used for the ODE solving and the student network learns these predictions. If that's the case, this needs to be a little bit more clear.
>
> **Response to Q8.**
>
> We agree that the notation could be difficult to understand. Hence, in the revision of Section 3.2, we aim to streamline equations, focusing on providing more comprehensive yet simplified explanations of formulas and notations. We appreciate the reviewer to point out this.
>
> **Q9. [Why Student Beats Teacher?]** Why would the student outperform the teacher?
>
> **Response to Q9.**
>
> Thank you for the insightful review. We believe that the reviewer's question touches the essence of our main contribution. In CTM, we extend a principle ("student beats teacher") of classification distillation community to generation distillation. This involves introducing direct learning signals from auxiliary losses, such as $L_{DSM}(data,student)$ and $L_{GAN}(data,student)$, to enhance student learning. Detailed explanation is provided below as we revisit knowledge distillation for classification models.
>
> - **[Student Learning in Classification Distillation]** How to obtain a good student is vastly investigated in the knowledge distillation community of classification tasks. It is widely acknowledged that the student classifier frequently performs equivalent to the teacher classifier [1]. A crucial factor contributing to this success is the direct training signal derived from the true label. More precisely, the student loss $L_{distill}(teacher,student)+L_{cls}(data,student)$ is the combination of a distillation loss $L_{distill}$ and a classifier loss $L_{cls}$ using the true data label, which provides a high-quality signal to stduent with the data label.
> - **[Student Learning in Generation Distillation]** On the contrary, within the generative community, distillation models exhibit inferior sample quality compared to the teacher, primarily because model optimization relies solely on the distillation loss $L_{distill}$. In our approach, we extend the principles of classification distillation to our model by introducing direct signals from $L_{DSM}(data,student)$ (an upper bound of KL divergence [2]) and $L_{GAN}(data,student)$ (a proxy of $f$-divergence [3]/IPMs [4])  to facilitate student learning. More precisely, $L_{DSM}$ leads the stduent to learn more accurate score, resulting in the improvement of large-NFE sample quality. On the other hand, $L_{GAN}$ substantially improves the quality of few-step generation. Our empirical findings consistently mirror the outcomes observed in classification distillation, illustrating that our model (student) surpasses the teacher (diffusion) through the direct learning signal provided by data.
>
> We reflected the above argument in the last paragraph of Introduction and Section 3.3.
>
> [1] Gou, Jianping, et al. "Knowledge distillation: A survey." International Journal of Computer Vision 129 (2021): 1789-1819.
>
> [2] Song, Yang, et al. “Maximum likelihood training of score-based diffusion models.” Advances in Neural Information Processing Systems 34 (2021): 1415-1428.
>
> [3] Nowozin, Sebastian, Botond Cseke, and Ryota Tomioka. "f-gan: Training generative neural samplers using variational divergence minimization." Advances in neural information processing systems 29 (2016).
>
> [4] Arjovsky, Martin, Soumith Chintala, and Léon Bottou. "Wasserstein generative adversarial networks." International conference on machine learning. PMLR, 2017.

---

> ### Author Response · Authors · 2023-11-18
> **Rebuttal by Authors**
>
> **Q10. [Why CTM Better than CM?]** Why this method gives better results that Consistency Models when s=0?
>
> **Response to Q10.**
>
> The reviewer's question pertains to understanding why CTM achieves a more accurate estimation of $G(\mathbf{x}_{t},t,0)$ compared to CM. Building upon the insights provided in **Q9. [Why Student Beats Teacher?]** that discusses supplementary components such as DSM/GAN losses, this thread further emphasizes the differences in distillation losses between CTM and CM. We also reflect the below arguments in the first paragraph of Section 5.2 and the entire new section of Appendix C.3.
>
> Our focus centers on 1-step generation $G_{\theta}(x_{T},T,0)$, where we examine the distillation mechanisms employed in CM (using *local consistency*) and CTM (using *soft consistency*), respectively. To align with the implementation, we adopt a discrete time setup.
>
> - **[Local Consistency (Implicit Information from Teacher)]**  Let us assume that at some training iteration the maximum time $T$ is sampled as a random time $t$. Then CM matches the long jump provided by $G_{\theta}(x_{T},T,0)$ and $G_{sg(\theta)}(x_{T \rightarrow T-\Delta t}^{\phi},T-\Delta t,0\big)$. Here, $x_{T \rightarrow T-\Delta t}^{\phi}$ denotes a 1-step denoised sample of the *pre-trained teacher* (with parameters $\phi$), obtained using an ODE solver, starting from $x_T$ at time $T$ and terminating at $T-\Delta t$. Hence, the student model $G_{\theta}(x_{T},T,0)$ distills only the teacher information within the interval $[T-\Delta t,T]$ and may lack precision for the trajectory within $[0,T-\Delta t]$. The transfer of teacher information for the interval $[0,T-\Delta t]$ may occur in another iteration with a random time $t\le T-\Delta t$. In this case, the student model $G_{\theta}(x_{t},t,0)$ distills teacher information solely within the interval $[t-\Delta t,t]$, where the network is trained with $x_{t}$ as the input.
>
>     However, for 1-step generation, $G_{\theta}(x_{T},T,0)$ still lacks perfect knowledge of the teacher's information within $[t-\Delta t,t]$. This is because, when the student network input is $x_{T}$, the teacher's information for the interval $[t-\Delta t,t]$ with $t\le T-\Delta t$ has not been explicitly provided, as the student was trained with the input $x_t$ to distill information within $[t-\Delta t,t]$. Given the non-overlapping intervals with distilled information from local consistency, the student neural network must extrapolate and attempt to connect the scattered teacher information. Consequently, this implicit signal provided by teacher results in slow convergence and inferior performance. We reference Figure 13 in the manuscript, wherein it illustrates the inferior performance of local consistency (blue curve).
> - **[Soft Consistency (Explicit Information from Teacher)]** At the opposite end of local consistency is *global consistency*, where the teacher prediction is constructed using an ODE solver to cover the entire interval $[0, T]$ (or $[0, t]$ for a random time $t$). In this case, the student model can explicitly extract information from the teacher. However, this approach is resource-intensive (3$\times$ slower than local consistency on CIFAR-10) during training due to the ODE solving calls on the entire interval at each iteration.
>
>     In contrast, CTM constructs the teacher prediction with our innovative technique, *soft matching* through the use of an ODE solver, spanning from $T$ to a random $u$. Importantly, $u$ is not limited to $T-\Delta t$ but can take any value in the range of $[0, T]$. As a result, the teacher information has the opportunity to be distilled and transmitted over a broader range of $[u, T]$. More precisely, if a random $u$ is sampled with $u\le t-\Delta t$, the range $[u,T]$ containes the interval $[t-\Delta t,t]$, and the student network directly distills teacher information of the interval $[t-\Delta t,t]$. As $u$ is arbitrary, the student of CTM with input $\mathbf{x}_{T}$ will ultimately receive explicit information from the teacher for any intermediate timesteps. This renders CTM superior to CM for 1-step generation, as evidenced in Figure 13 (green curve), while maintaining training efficiency (2$\times$ faster than global consistency on CIFAR-10).

---

> ### Author Response · Authors · 2023-11-22
> **Looking forward to your response**
>
> Thank you for your thorough and insightful review. As the rebuttal period is coming to its end, we are reaching out to see if you have any additional questions or concerns that we can address. Thank you for your valuable feedback, and we sincerely anticipate your response.

---

> > ### Author Response · Authors · 2023-11-23
> > **Looking forward to your response**
> >
> > Dear Reviewer 1 (6usj),
> >
> > We sincerely appreciate your devotion and constructive comments to our work. As the reviewer-author discussion period left 1 hour, we would like to ask if there is any other concerns and questions that we could discuss more for better revision.
> >
> > Warm regards,
> >
> > Authors 7687

---

### Author Response · Authors · 2023-11-20
**A kind reminder for further discussion.**

We express sincere gratitude to the reviewers for their constructive comments, which we consider crucial for improving our work. A gentle reminder to reviewers to review our replies; we are eager to address any further concerns or questions and engage in an active discussion. Thank you again for your attention and consideration.

---

### Meta-Review · Area_Chair_vr24 · 2023-12-14

**Metareview:**

**Summary**

This paper proposes the consistency trajectory model (CTM) by extending the idea of the consistency model (CM) (Song et al., 2023). Whereas the CM reduces the required number of function evaluations by constructing a model which directly estimates the endpoint $x_0$ of a sampling trajectory of a diffusion model from the point $x_t$ on the trajectory at time $t\in[0,T]$, the CTM constructs a model $G(x_t,t,s)$ which estimates the point $x_s$ at time $s\in[0,t]$ on a trajectory from the point $x_t$ at time $t$ on the trajectory. The CTM corresponds to the CM if one lets $s=0$, and to the conventional score-based model if one lets $s=t$. Three proposed loss functions, the CTM loss $ℒ_{\rm CTM}$ for soft consistency matching, the DSM loss $ℒ_{\rm DSM}$ for improving jump precision when $s\approx t$, and the adversarial loss $ℒ_{\rm GAN}$ for better reconstruction quality and for stabilizing training, are combined to form the loss function for training (Sections 3.2 and 3.3). A new sampling, named $\gamma$-sampling, is also proposed, which can be regarded as a one-parameter extension of the distillation sampling ($\gamma=0$), the multistep sampling introduced in Song et al. (2023) ($\gamma=1$), and the EDM stochastic sampler ($\gamma\in(0,1)$).

**Strengths**

The rating/confidence of the four reviewers were 6/4, 6/4, 8/4, 6/3, all of which were above the acceptance threshold.
The contribution of this paper is not a mere extension of the CM to estimation of midpoints on the solution trajectories but includes proposal of a new training loss combining three loss functions, as well as the $\gamma$-sampling.
Discussion between the authors and the reviewers has been reflected in the revised version of the paper, with improved presentation.

**Weaknesses**

Upon reading of this paper by myself, I noticed the following minor points that would need revision:
- Page 1, line 36: NFE increase(s)
- Page 1, line 37: Theorem 1 (in this paper) explains
- Page 2, line 4: The abbreviation DSM should be explained at its first appearance here.
- Page 2, line 17: is established (as)
- Page 3: Figure 2 should be cited in the main text.
- Page 3, line 41: We refer (to)
- Page 4, line 20: coincides (to → with)
- Page 5, line 12: I did not understand the meaning of the subscripts of the expectation $𝔼_{t,x_0,x_t|x_0}$. It should be written as  $𝔼_{t,x_0,x_t}$ (i.e., the expectation is to be taken with respect to the joint distribution of $t,x_0,x_t$), or decomposed into two steps as $𝔼_{t,x_0}[𝔼_{x_t|x_0}[\cdots]]$ if the authors want to make the conditioning on $x_0$ explicit.
- Page 5, line 18: Is $x_{\rm est}$ meant to be $x_{\rm est}(x_t,t,0)$?
- Page 6, lines 30-31: The observation that CM's multistep sampler, originally proposed in Song et al. (2023) with the aim of improving sample quality with a larger NFE, may degrade sample quality would be worth mentioning.
- Page 7, line 4: 2-step(s); Denote $p_{\theta^*,2}$ as → Let $p_{\theta^*,2}$ denote
- Page 7, line 6: The definition of $D_{TV}$ should be provided.
- Page 7, lines 3, 7: N-steps → N steps
- Page 8, line 6: comparably diverse to (that → those) of EDM or CM.
- Page 8, Tables 2, 3:
  - What were the standard errors, and how significant were the best values?
  - What are the underlined numbers representing?
- Page 9, line 13: lead (to)
- Page 9, lines 30-31: for both small and large NFE sample quality → for both small- and large-NFE sample quality

**Justification For Why Not Higher Score:**

Although a number of novel technical ideas are involved, heavy dependence of the contribution on the CM (Song et al., 2023) would make it harder to attract attention of wider audience.

**Justification For Why Not Lower Score:**

Extending the consistency model is a timely research subject. The ideas leading to the proposal are easy to understand, and good performance of the proposal has been demonstrated via the experiments. All the reviewers rated this paper above the acceptance threshold.

---

### Decision · Program_Chairs · 2024-01-16

Accept (poster)